# Polygenic adaptation after a sudden change in environment

**Laura Katharine Hayward[1,2]\*, Guy Sella[3,4]\***

[1]Department of Mathematics, Columbia University, New York, United States; [2]Institute of Science and Technology, Maria Gugging, Austria; [3]Department of Biological Sciences, Columbia University, New York, United States; [4]Program for Mathematical Genomics, Columbia University, New York, United States

**Abstract** Polygenic adaptation is thought to be ubiquitous, yet remains poorly understood. Here, we model this process analytically, in the plausible setting of a highly polygenic, quantitative trait that experiences a sudden shift in the fitness optimum. We show how the mean phenotype changes over time, depending on the effect sizes of loci that contribute to variance in the trait, and characterize the allele dynamics at these loci. Notably, we describe the two phases of the allele dynamics: The first is a rapid phase, in which directional selection introduces small frequency differences between alleles whose effects are aligned with or opposed to the shift, ultimately leading to small differences in their probability of fixation during a second, longer phase, governed by stabilizing selection. As we discuss, key results should hold in more general settings and have important implications for efforts to identify the genetic basis of adaptation in humans and other species.

## Editor's evaluation

This paper is an impressive look at an important problem: understanding the genetic underpinnings of evolution acting on a quantitative trait. The authors analytically study the response to an abrupt shift in phenotypic optimum, in terms of both phenotype and genetic basis (how various alleles/ loci contribute to this response). The basic assumptions are classic, but the methods and findings are new (especially the finite population effects) and well supported by clear analytical approximations and extensive simulation checks. The main finding is that the relative contribution of large vs moderate effect alleles changes substantially and predictably over a long-time period after the shift, even though the phenotypic changes are already undetectable over this period.

**\*For correspondence:** lauhayward@gmail.com (LKH); gs2747@columbia.edu (GS)

**Competing interest:** The authors declare that no competing interests exist.

## Introduction

Many traits under natural selection are quantitative, highly heritable, and genetically complex, meaning that they take on continuous values, that a substantial fraction of the population variation in their values arises from genetic differences among individuals, and that this variation arises from small contributions at many segregating loci. It therefore stands to reason that the responses to changing selective pressures often involve adaptive changes in such traits, accomplished through changes to allele frequencies at the many loci that affect them. In other words, we should expect polygenic adaptation in complex, quantitative traits to be ubiquitous. This view traces back to the dawn of population and quantitative genetics (*Wright, 1931*; *Fisher, 1958*) and is supported by many lines of evidence (*Walsh and Lynch, 2018*; *Sella and Barton, 2019*).

Notably, it is supported by studies of the response to directional, artificial selection on many traits in plants and animals in agriculture and in evolution experiments (*Walsh and Lynch, 2018*; *Sella and Barton, 2019*). In these settings, selected traits typically exhibit amazingly rapid and sustained

adaptive changes (*Weber and Diggins, 1990*; *Barton and Keightley, 2002*; *Hill, 2016*), which are readily explained by models in which the change is driven by small shifts in allele frequencies at numerous loci (*Weber and Diggins, 1990*; *Hill and Kirkpatrick, 2010*), and inconsistent with models with few alleles of large effect (*Barton and Keightley, 2002*; *Zhang and Hill, 2005*). The potential importance of polygenic adaptation has also been highlighted by more recent efforts to elucidate the genetic basis of adaptation in humans. In the first decade after genome-wide polymorphism datasets became available, this quest was largely predicated on the monogenic model of a hard selective sweep (*Voight et al., 2006*), in which adaptation proceeds by the fixation of new or initially rare beneficial mutations of large effect (e.g. *Smith and Haigh, 1974*; *Kaplan et al., 1989*). Subsequent analyses, however, echoed studies of artificial selection in indicating that hard sweeps were rare, at least over the past ~500,000 years of human evolution (*Coop et al., 2009*; *Hernandez et al., 2011*). Yet humans plausibly adapted in myriad ways during this time period, and they definitely experienced substantial changes in selection pressures, notably during more recent expansions across the globe. These considerations refocused the quest for the genetic basis of human adaptation on polygenic adaptation (*Pritchard et al., 2010*; *Pritchard and Di Rienzo, 2010*).

Findings from genome wide association studies (GWASs) in humans have been central to this research program. Statistical analyses of GWASs indicate that in humans, heritable variation in complex traits is highly polygenic (*Loh et al., 2015*; *Shi et al., 2016*; *Boyle et al., 2017*). For example, for many traits, estimates of the heritability contributed by chromosomes are approximately proportional to their length (*Shi et al., 2016*), suggesting that the contributing variants are numerous and roughly uniformly distributed across the genome. Such findings reinforced the view that adaptive changes to quantitative traits are likely to often be highly polygenic, but also implied that their identification would be difficult, as the changes to allele frequencies at individual loci may be minute. To overcome this limitation, recent studies pooled signatures of frequency changes over the hundreds to thousands of alleles that were found to be associated with an increase (or decrease) in a given trait (*Turchin et al., 2012*; *Berg and Coop, 2014*; *Robinson et al., 2015*; *Field et al., 2016*; *Berg et al., 2019b*; *Edge and Coop, 2019*; *Speidel et al., 2019*). Initial studies suggest that polygenic adaptation has affected multiple human traits, but these conclusions have been called into question with the realization that the results are highly sensitive to systematic biases in GWASs, most notably due to population structure confounding (*Berg et al., 2019a*; *Sohail et al., 2019*).

Given that polygenic adaptation is plausibly ubiquitous, yet likely hard to identify, there is a clear need for a deep understanding of its behavior in populations and footprints in data. To date, theoretical work has primarily focused on two scenarios. The first is motivated by the observed responses to sustained artificial selection, modeled either as truncation selection (*Robertson, 1960*) or as stabilizing selection, with the optimal phenotype moving at a constant rate in a given direction (e.g. *Bürger and Lynch, 1995*; *Bürger, 1999*; *Kopp and Hermisson, 2009*; *Matuszewski et al., 2015*; *Jain and Devi, 2018*). In natural populations, however, quantitative traits are unlikely to be subject to long-term continuous change in one direction. Instead, considerable evidence indicates that they are often subject to long-term stabilizing selection (*Sella and Barton, 2019*), with intermittent shifts of the optimum in different directions. The second scenario therefore assumes that a sudden change in the environment induces an instantaneous shift in the optimum of a trait under stabilizing selection (*Lande, 1976*; *Barton and de Vladar, 2009*; *de Vladar and Barton, 2014*; *Jain and Stephan, 2015*; *Bod'ová et al., 2016*; *Jain and Stephan, 2017b*; *Stetter et al., 2018*; *Thornton, 2019*). Although more elaborate scenarios (where, for example, the optimum and/or strength of stabilizing selection vary frequently) are also possible, this simple scenario provides a sensible starting point for thinking about polygenic adaptation in nature, and is our focus here.

Although there has been considerable work on the adaptive response to an instantaneous change in optimal phenotype, our understanding of this process is still limited. Seminal work by *Lande, 1976* described the change in the phenotypic mean assuming that phenotypes are normally distributed in the population and that the phenotypic variance remains constant over time. *Barton and Turelli, 1986b* derived recursions for the expected change to higher moments of the phenotypic distribution, and showed that when phenotypic variation arises from alleles with large effect sizes, which are strongly selected and rare, the response to selection introduces skew in the phenotypic distribution that can substantially affect the change in the phenotypic mean. Their recursions, however, are not

generally tractable, and their analyses do not extend to the phenotypic response in more realistic cases, in which phenotypic variation arises from alleles with a wide range of effect sizes. Moreover, with GWASs now enabling us, at least in principle, to learn about the genetic basis of the phenotypic response, we would like to understand the allele dynamics that underlie it.

Several studies have tackled this problem using simulations (e.g. *Stetter et al., 2018*; *Thornton, 2019*). Although illustrative of the dynamics, it is unclear how to generalize their results, given (necessarily) arbitrary choices about multiple parameters and the complexity of these dynamics. In turn, elegant analytical work by *de Vladar and Barton, 2014* and extensions by *Jain and Stephan, 2017a*; *Jain and Stephan, 2017b* afford a general understanding of the allele dynamics in models with an infinite population size. These dynamics, however, are shaped by features of mutation-selection balance that are specific to infinite populations. Notably, they strongly depend on the frequency of alleles prior to the shift in optimum following deterministically from their effect size, and on the critical effect size at which this frequency transitions from being dominated by selection to being dominated by mutation. In real (finite) populations, the frequencies of alleles whose selection effects are sufficiently small to be dominated by mutation will be shaped by genetic drift; more generally, variation in allele frequencies due to genetic drift will crucially affect the allele response to selection (see below). Thus, we still lack a solid understanding of the allele dynamic underlying polygenic adaptation in natural populations, notably in humans.

Here, we follow previous work in considering the phenotypic and allelic responses of highly polygenic traits after a sudden change in optimal phenotype, but we do so in finite populations and employ a combination of analytic and simulation approaches to characterize how the responses vary across a broad range of evolutionary parameters.

## Model

We build upon the standard model for the evolution of a highly polygenic, quantitative trait subject to stabilizing selection (*Wright, 1935b*; *Robertson, 1956b*; *Turelli, 1984*; *Keightley and Hill, 1988*; *Johnson and Barton, 2005*; *Simons et al., 2018*; *Sella and Barton, 2019*). An individual's phenotype is represented by the value of a continuous trait, which follows from its genotype by the standard additive model (*Falconer, 1996*; *Lynch and Walsh, 1998*). Namely, we assume that the number of genomic sites affecting the trait (i.e. the target size) is very large, $L \gg 1$, and that an individual's phenotype is given by

$$z = \sum_{l=1}^{L} (a_l + a'_l) + \epsilon, \tag{1}$$

where the first term is the genetic contribution, with $a_l$ and $a'_l$ denoting the phenotypic effects of the alleles inherited from the parents at site $l$, and $\epsilon \sim N(0, V_E)$ is the environmental contribution.

Stabilizing selection is introduced by assuming that fitness declines with distance from the optimal trait value positioned at the origin ($z = 0$). Specifically, we assume a Gaussian fitness function:

$$W(z) = \text{Exp} \left[ -z^2 / (2V_S) \right], \tag{2}$$

where $V_S^{-1}$ measures the strength of selection. The specific form of the fitness function is unlikely to affect our results under parameter ranges of interest (see below), however. Additionally, since the additive environmental contribution to the phenotype can be absorbed into $V_S$ (by replacing it by $V'_S = V_S + V_E$; e.g., *Turelli, 1984*; *Bürger, 2000*), we consider only the genetic contribution.

The population dynamics follow the standard model of a diploid, panmictic population of constant size $N$, with non-overlapping generations. In each generation, parents are randomly chosen to reproduce with probabilities proportional to their fitness (i.e. Wright-Fisher sampling with fertility selection), followed by mutation, free recombination (i.e. no linkage) and Mendelian segregation. We assume that the mutational input per site per generation is sufficiently small such that segregating sites are rarely more than bi-allelic (i.e. that $\theta = 4Nu \ll 1$, where $u$ is the mutation rate per site per generation). We therefore employ the infinite sites approximation, in which the number of mutations per gamete per generation follows a Poisson distribution with mean $U = Lu$. The effect sizes of mutations, $\pm a$, are drawn from a symmetric distribution, that is, with equal probability of increasing or decreasing the trait value; we therefore specify this distribution in terms of the distribution of allele magnitudes, $g(a)$. *Appendix 1—table 1* provides a summary of our notation.

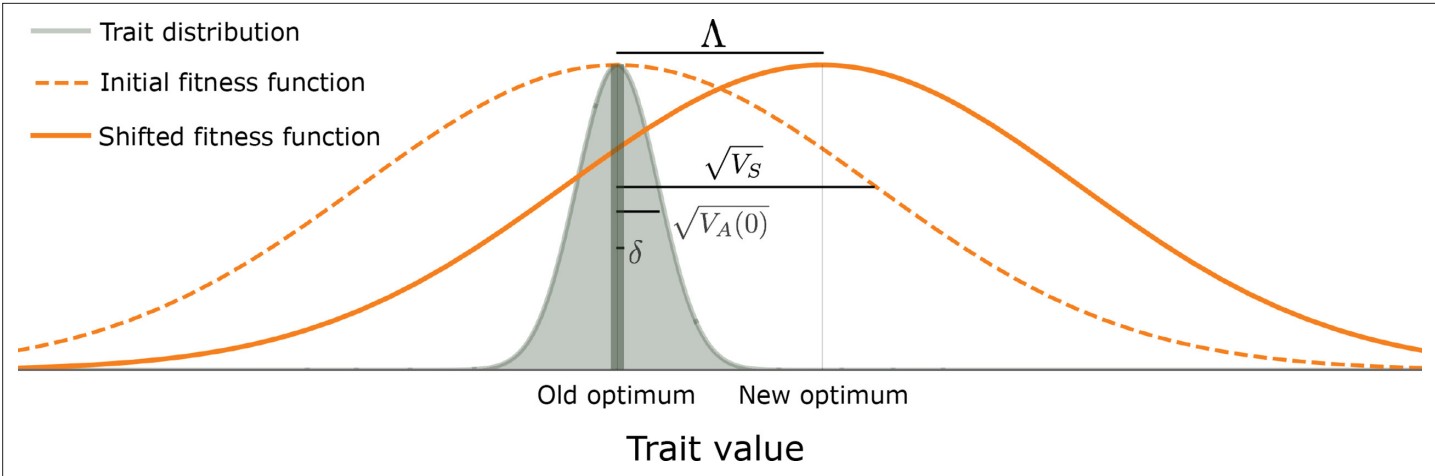

**Figure 1.** The evolutionary scenario. Before the shift in optimum, phenotypes are distributed symmetrically, with a mean that is very close to the old optimum and a standard deviation that is substantially smaller than the width of the fitness function ($V_A(0) \ll V_S$). We consider the response to an instantaneous shift in optimum, for the case where the magnitude of the shift is smaller than the width of the fitness function ($\Lambda \lesssim \sqrt{V_S}$). See text for further details.

## Evolutionary scenario and parameter ranges

We consider that at the outset (i.e. before the shift in optimal phenotype), the population has attained mutation-selection-drift balance. We follow previous work modeling this balance in making several plausible assumptions about parameter ranges (e.g. *Simons et al., 2018*), which ensure that genetic variation in the trait is highly polygenic, and subject to effective but not catastrophically strong selection. First, we assume that the per generation, population scaled mutational input is sufficiently large to guarantee high polygenicity (specifically, that $\sqrt{2NU} \gg 1$). Second, we assume that the expected number of mutations affecting the trait per generation, per gamete, is small (specifically, that $U = Lu \leq 0.02$), such that the loss in mean population fitness (i.e. the genetic load) is not too large. As an example, for this assumption to be violated in humans, the mutational target size, $L$, would have to exceed ~1.5 Mb (assuming that $u \approx 1.25 \cdot 10^{-8}$ per bp per generation; *Kong et al., 2012*; *Besenbacher et al., 2016*). We note that we expect our results to hold for substantially greater values of $U$ in extensions of our model in which genetic variation in the trait under consideration has pleiotropic effects on other selected traits. Third, we make the standard assumption that the selection coefficients of all alleles satisfy $s \ll 1$, which implies that $s_e = a^2/V_S \ll 1$ (subscript $e$ for equilibrium; see below and *Wright, 1931*; *Wright, 1935a*; *Turelli, 1984*). Fourth, we assume that a substantial proportion of mutations are not effectively neutral, i.e., have $S = 2Ns_e \gtrsim 1$ (by "$\gtrsim$"/"$\lesssim$" we mean greater/smaller than or on the same order as). This last assumption is supported by empirical estimates of persistence time of mutations contributing to quantitative genetic variation for a variety of traits and taxa (*Walsh and Lynch, 2018*; *Sella and Barton, 2019*) and by inferences based on human GWASs (*Simons et al., 2018*; *Zeng et al., 2018*; *O'Connor et al., 2019*; *Zeng et al., 2021*), which indicate that quantitative genetic variation is not predominantly neutral. Under these assumptions, the phenotypic distribution at mutation-selection-drift balance is symmetric and tightly centered on the optimal phenotype (*Figure 1*). Specifically, the mean phenotype exhibits tiny, rapid fluctuations around the optimal phenotype with variance $\delta^2 = V_S/(2N)$ (*Simons et al., 2018*); the phenotypic standard deviation is considerably greater than these fluctuations, i.e., $\sqrt{V_A} \gg \delta$ (Section 3.2 of *Appendix 3*), but it is substantially smaller than the width of the fitness function, that is, $V_S \gg V_A$ (*Simons et al., 2018* and Section 3.2 of *Appendix 3*).

We then consider the response to an instantaneous shift of $\Lambda$ in optimal phenotype at time $t = 0$ (*Figure 1*), ensuring that the shift is substantial yet not immensely large when compared to the genetic variance in the trait and in terms of the reduction in mean population fitness. To these ends, we first assume that the shift in optimum is greater than the equilibrium fluctuations in mean phenotype, i.e., that $\Lambda > \delta$. Second, we assume that the shift is smaller than, or on the order of, the width of the fitness function ($\Lambda \lesssim \sqrt{V_S}$) and that the vast majority of mutations move the phenotype by much less than the distance over which fitness declines, $\sqrt{V_S}$ (i.e. that $a \ll \sqrt{V_S}$). These two assumptions guarantee that the maximal directional selection coefficient of alleles, attained immediately after the shift, satisfies $s_d = 2 \cdot (\Lambda \cdot a)/V_S \ll 1$ (see below and

*Wright, 1931*; *Barton and Turelli, 1986b*). Our current condition on the magnitudes of mutations (i.e. that $a \ll \sqrt{V_S}$) is stronger than our earlier one at equilibrium (i.e., that $s_e = a^2/V_S \ll 1$), but is still not particularly restrictive, as it allows for the selection coefficients of alleles at equilibrium, $s_e$, to be as large as 1%. As a concrete example, with a population size of $N = 10^4$, both this condition and the condition that a substantial proportion of mutations not be effectively neutral ($S = 2Ns_e \gtrsim 1$) are satisfied if selection coefficients of alleles at equilibrium are exponentially distributed with mean between $10^{-5}$ and $10^{-3}$. Third, we assume that adaptation to the new optimum requires only a small average frequency change per segregating site, which translates into the requirement that $\Lambda/\sqrt{V_A(0)} \lesssim 1/2 \cdot \sqrt{2NU}$ (see Section 3.2 of *Appendix 3*). Both this and the previous bound on the shift size (that $\Lambda \lesssim \sqrt{V_S}$) are again not particularly restrictive, in that they allow for shifts of several equilibrium phenotypic standard deviations (given that $\sqrt{2NU} \gg 1$ and $V_S \gg V_A(0)$ for this and the former bounds, respectively). Our assumptions on parameter values are summarized in *Appendix 1—table 2*.

## Choice of units

When we study the allelic response, we use units based on the dynamics at mutation-selection-drift balance (i.e. before the shift in optimum). Using arbitrary units, and denoting corresponding parameter values with tildes, the population-scaled selection coefficient at mutation-selection-drift balance is $S = 2Ns_e = 2N\tilde{a}^2/\tilde{V}_S = \tilde{a}^2/\tilde{\delta}^2$. We measure the trait in units of $\delta$, the typical deviation of the population mean from the optimum. In these units, the magnitude of effect size $a = \sqrt{S}$, the stabilizing selection parameter $V_S = 2N$, and the contribution of a segregating allele to variance is $v^*(a,x) = 2a^2x(1-x)$. As shown below, stating our results in these terms makes their form invariant with respect to the population size, $N$, and the strength of stabilizing selection, $V_S^{-1}$.

## Simulations and resources

We compare our analytical results to three kinds of simulations, which differ in their simplifying assumptions and computational tractability (see Section 2 of *Appendix 3* for further detail). The first realizes the full model described above; it is run for a burn-in period of $10N$ generations before the shift and for a period of $12N$ generations after, to attain equilibrium both before and after. The second traces *all alleles* rather than individuals, assuming linkage equilibrium (rather than free recombination). Changes to allele frequencies every generation are modeled according to the Wright-Fisher process. These second simulations are also run with a burn-in period of $10N$ generations before the shift, then for $12N$ generations after. The third kind of simulation traces the trajectory of a *single allele* segregating at the time of the shift. To that end: (i) given the effect size of the allele, we sample initial minor allele frequencies from the closed form, equilibrium distributions (*Equation A3-27* in *Appendix 3*), using importance sampling based on the density of variance contributed by different minor allele frequencies (Section 3.1 of *Appendix 3*); and (ii) the trajectory of the mean phenotype of the population over time, on which the allele dynamics depend, is given as input, based on either an analytical approximation (see below) or on an average over simulations of the second kind. The *single allele* simulation is run until the focal allele fixes or goes extinct. Documented code for simulations can be found at https://github.com/sellalab/PolygenicAdaptation1D (copy archived at swh:1:rev:35d0857272a3929bad9fad0856e90c24e032b5ff; *Hayward and Sella, 2022*).

In the main text, we use the simulations that afford the highest resolution in comparisons with analytical predictions, whereas in Section 2.2 of *Appendix 3* we validate our main results against simulations that realize the full model (at a lower resolution). Specifically, we compare most of the predictions about the allele dynamics with the results of *single allele* simulations, running 250,000 replicas for any given allele effect size and optimum shift size (see parameter choices below). The *single allele* simulations do not describe phenotypic change or the trajectories of mutations that arise after the shift in optimum, however. We therefore compare the predictions for these processes with the results of the *all alleles* simulations; in these simulations, we run 2500 replicas with any given set of parameters.

The simulations used in the main text correspond to the two qualitative phenotypic responses described below, which we refer to as the Lande and non-Lande (see *Results*). Specifically, we use the following simulation parameter values:

- In all simulations, we take a populations size of $N = 10^4$ and a shift size of $\Lambda = 2\sqrt{V_A(0)}$ or $4\sqrt{V_A(0)}$. Since we work in units of $\delta$, the typical deviation of the population mean from the optimum at equilibrium, we take $V_S = 2N$.

- The *single allele* simulations always assume the Lande phenotypic response, which is determined by the initial genetic variance; we take an initial variance such that $\sqrt{V_A(0)} = 17 \cdot \delta$.
- The *all alleles* simulations are specified by the mutation rate, $U$, and the distribution of allele effect sizes squared, for which we use an exponential distribution (measuring the trait in units of $\delta$). We use the following parameter values:

**Lande case:** $U = 0.03$ and $E(a^2) = E(S) = 1$.
**Non-Lande case:** $U = 0.01$ and $E(a^2) = E(S) = 16$.
These parameter choices yield the same genetic variance at equilibrium (before the shift) in both cases; specifically, $\sqrt{V_A(0)} = 29 \cdot \delta$.

## Results

### Phenotypic response

We first consider how the population's mean phenotype approaches the new optimum. In Section 1.2 of *Appendix 3*, we express the mean distance from the new optimum, $D(t)$, as a sum over alleles' contributions. We show that under our assumptions, the expected, per generation change in this distance is well approximated by

$$E\left(\Delta D(t)\right) \approx -V_A(t)/V_S \cdot D(t) + (1 - D^2(t)/V_S) \cdot \mu_3(t)/(2V_S), \tag{3}$$

where $V_A(t)$ and $\mu_3(t)$ denote the 2nd and 3rd central moments of the phenotypic distribution. The 1st term on the right-hand side reflects selection to reduce the distance between the mean phenotype and the new optimum, which is proportional to this distance and to the additive genetic variance (*Lande, 1976*). The 2nd term reflects the effect of stabilizing selection on an asymmetric (skewed) phenotypic distribution. In particular, when the mean phenotype is near the optimum (and this 2nd term is approximately $\mu_3(t)/(2V_S)$), stabilizing selection pushes the mean phenotype in the direction opposite to the thicker tail of the phenotype distribution (see Section 1.2 of *Appendix 3* for further discussion of *Equation 3*). Similar expressions were derived by *Barton and Turelli, 1986b* under the rare-alleles approximation and by *Bürger, 1991* under the assumption of a parabolic fitness function.

We rely on *Equation 3* to describe the phenotypic response to selection. This response takes a simple form in the infinitesimal limit (*Fisher, 1918*) in which genetic variation at equilibrium arises from infinitely many segregating alleles with infinitesimally small effect sizes (see Section 8 of *Appendix 3* for details). In this limit, the equilibrium phenotypic distribution is Normal and it remains Normal with the same variance after the shift, because the change in mean phenotype is achieved by infinitesimally small changes to allele frequencies at infinitely many loci, with no change to the frequency distribution (*Lande, 1976*, and Section 8 of *Appendix 3*). Under these assumptions, *Equation 3* reduces to

$$E\left(\Delta D(t)\right) = -V_A(0)/V_S \cdot D(t), \tag{4}$$

which (in continuous time) is solved by

$$D_L(t) = \Lambda \cdot \mathrm{Exp}\left[-V_A(0)/V_S \cdot t\right]. \tag{5}$$

This solution was first derived by *Lande, 1976*, and in what follows, we refer to it as Lande's solution or approximation. When genetic variance is dominated by loci with small and intermediate effect sizes (as defined below), the trait is highly polygenic, and the shift in optimum is not too large relative to the phenotypic standard deviation, changes to the 2nd and 3rd central moments of the phenotypic distribution are small and the expected phenotypic response is well approximated by Lande's solution (*Figure 2A* and *Appendix 3—figures 26* and *27*).

More generally, given our assumptions that polygenicity is high and that the shift is not too large (see Section 6 of *Appendix 3*), the deviations from Lande's approximation are usually small and their magnitude is determined by the distribution of allele effect sizes (see section on *The allelic response in the equilibration phase* and Section 6 of *Appendix 3*). Specifically, changes to the 2nd and 3rd central moments of the phenotypic distribution are greater when alleles with large effects contribute markedly to genetic variance (*Figure 2C*, *Appendix 3—figures 26* and *27*, and *Barton and Turelli, 1986b*). For some intuition, consider a pair of minor alleles with the same initial frequency and magnitude of effect, where the effect of one is aligned with the shift and the effect of the other opposes it. After the shift, directional selection increases the frequency of the aligned allele relative to that of the opposing one.

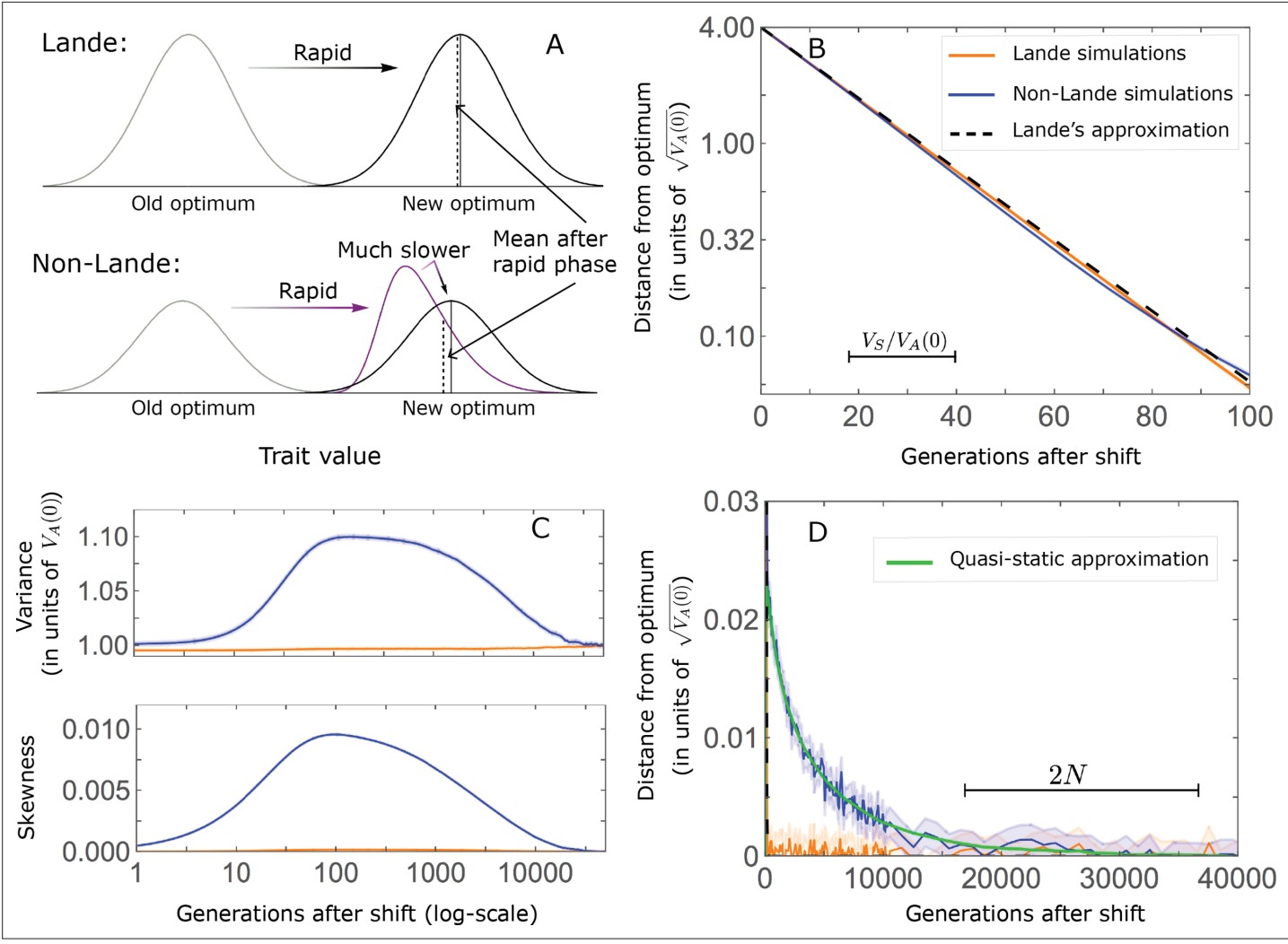

**Figure 2.** The phenotypic response to a shift in optimal phenotype. (**A**) Cartoon of the two kinds of phenotypic response: (i) the Lande approximation, in which the mean approaches the new optimum exponentially with time and the phenotypic distribution maintains its shape; (ii) substantial deviations from Lande's approximation, in which the mean approaches the new optimum rapidly at first, but during this time the phenotypic distribution becomes skewed, causing the mean's approach to slow down dramatically, to a rate that is dictated by the decay of the 3rd central moment. (**B**) In both the Lande and non-Lande cases, the mean phenotype initially approaches the new optimum rapidly. This approach is described by Lande's approximation, and is thus almost identical in the two cases (which is why only the Lande curve is visible). The simulation results were generated using the *all alleles* simulation with a shift of $\Lambda = 4 \cdot \sqrt{V_A(0)}$, as detailed in *Simulations and resources*. For each quantity described in **B-D**, we show the simulations' mean ±1.96 SE (solid lines and shaded regions, respectively). (**C**) In the non-Lande case, the phenotypic variance and skewness increase during the rapid phase and then take a very long time to decay to their values at equilibrium. (**D**) Over the longer-term, the approach to the optimum in the non-Lande case almost grinds to a halt, where its rate can be described by the quasi-static approximation (*Equation 6*). While the non-Lande response differs from Lande's approximation, the difference is small: the maximal deviation in mean phenotype is $\sim 0.06 \cdot \sqrt{V_A(0)}$, the variance increases by ~ 10% and the maximal skewness is tiny (less than 0.01).

The frequency increase of the aligned allele increases variance more than the frequency decrease of the opposing allele decreases it, resulting in a net increase to variance (*Figure 2C*; *Barton and Turelli, 1986b*; *de Vladar and Barton, 2014*; *Jain and Stephan, 2017a*). The relative changes in frequency and thus the net increase in variance are greater for alleles with larger effects. Next consider the 3rd central moment. At equilibrium, the contributions of alleles with opposing effects to the 3rd central moment cancel out. After the shift, the frequency increase of aligned alleles relative to opposing ones introduces a non-zero 3rd central moment (*Figure 2C*). Large effect alleles contribute substantially more to this 3rd central moment, plausibly because their individual contribution to the 3rd central moment at equilibrium is greater (see Section 3.3 of *Appendix 3* and *Appendix 3—figure 7B*) and because they exhibit larger relative changes in frequency after the shift (see section on *The allelic*

*response in the rapid phase* and Section 4 of *Appendix 3*). The same reasoning suggests that very large shifts in optima or low polygenicity could also lead to substantial changes to the 2nd and 3rd central moments of the phenotypic distribution (Section 6 of *Appendix 3* and **Barton and Turelli, 1986b**), but these cases violate our assumptions and are beyond the scope of this manuscript.

The increase in 2nd and 3rd central moments after the shift result in a phenotypic dynamic with two distinct phases. First, immediately after the shift, the mean phenotype rapidly approaches the new optimum, akin to the exponential approach in Lande's approximation. In this case, however, genetic variance increases and thus the exponential rate of approach may increase, making the expected approach even faster (*Equation 3*). Shortly thereafter, when the mean phenotype nears the optimum, the decreasing distance and increasing 3rd central moment reach the point at which

$$D(t) \approx \mu_3(t)/(2V_A(t)). \tag{6}$$

The two terms on the right-hand side of *Equation 3* then approximately cancel out, and the dynamic enters a second, prolonged phase, in which the approach to the optimum nearly grinds to a halt (*Figure 2D*). During this phase, the expected change in mean phenotype can be described in terms of a quasi-static approximation given by *Equation 6* (*Figure 2D* and *Appendix 3—figure 25*). The rate of approaching the optimum is then largely determined by the rate at which the 3rd central moment decays. This roughly corresponds to the rate at which the allele frequency distribution equilibrates and mutation-selection-drift balance is restored around the new optimum (see section on *Other properties of the equilibration process*).

## Allele dynamics

We now turn to the allele dynamics that underlie the phenotypic response. These dynamics can be described in terms of the first two moments of change in frequency in a single generation (**Ewens, 2004**, Chapter 4). For an allele with effect size $\pm a$ and frequency $x$, we calculate the moments by averaging the fitness of the three genotypes over genetic backgrounds (Section 1 of *Appendix 3*). Under our assumptions, the moments are well approximated by

$$E(\Delta x) \approx (\pm a \cdot D(t)/V_S) \cdot x(1-x) - (a^2/V_S) \cdot (1 - D^2(t)/V_S) \cdot x(1-x)(1/2-x) \tag{7}$$

and

$$V(\Delta x) \approx x(1-x)/(2N), \tag{8}$$

which is the standard drift term. Similar expressions for the first moment trace back to **Wright, 1935b** and have been used previously to study the response to selection on quantitative traits (**Barton, 1986a**; **Bürger, 1991**; **Charlesworth, 2013**; **de Vladar and Barton, 2014**).

The two terms in the first moment reflect different modes of selection: directional and stabilizing, respectively. The first term arises from directional selection on the trait and takes a semi-dominant (additive) form with selection coefficient $s_d = \pm 2a \cdot D(t)/V_S$. Its effect is to increase the frequency of alleles whose effects are aligned with the shift (and vice versa) and its strength weakens as the distance to the new optimum, $D$, decreases. The second term arises from stabilizing selection on the trait and takes an under-dominant form with selection coefficient $s_e = a^2/V_S \cdot (1 - D^2(t)/V_S)$. Its effect is to decrease an allele's contribution to phenotypic variance, $2a^2x(1-x)$, by reducing minor allele frequency (MAF); it becomes weaker as the MAF approaches 1/2.

The relative importance of the two modes of selection varies as the mean distance to the new optimum, $D$, decreases. We therefore divide the allelic response into two phases: a *rapid phase*, immediately after the shift, in which the mean distance to the new optimum is substantial and changes rapidly, and a subsequent, prolonged *equilibration phase*, in which the mean distance is small and changes slowly (**Jain and Stephan, 2017a**). We define the end of the rapid phase as the time, $t_1$, at which Lande's approximation for the distance to the optimum $D_L(t_1)$ equals the typical deviation of the population mean from the optimum at equilibrium $\delta = \sqrt{V_S/2N}$, i.e.,

$$t_1 \equiv (V_S/V_A(0)) \cdot \text{Ln}\left[\Lambda/\delta\right] \sim (1/U) \cdot \text{Ln}\left[\Lambda/\delta\right] \tag{9}$$

(in Section 3.2 of *Appendix 3* we show that $V_S/V_A(0) \sim 1/U$). This definition is somewhat arbitrary, as the transition between phases is gradual, but it roughly captures the change in allele dynamics

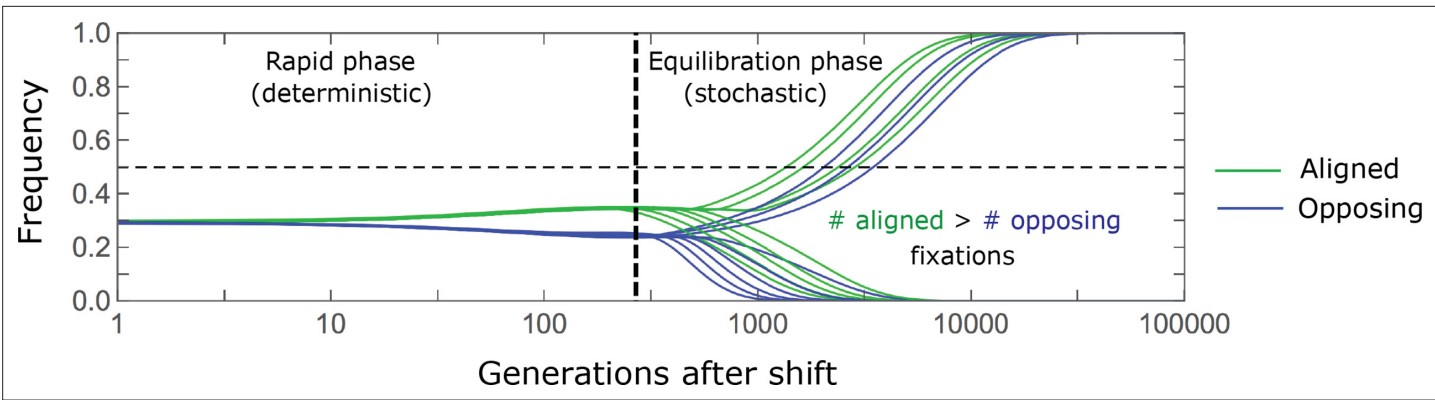

**Figure 3.** A cartoon of allele dynamics. We divide the allele dynamics into rapid and equilibration phases, based on the rate of phenotypic change, and consider the trajectories of alleles with opposing effects of the same magnitude, which start at the same initial minor frequency. During the rapid phase, alleles whose effects align with the shift slightly increase in frequency relative to those with opposing effects. During the equilibration phase, this frequency difference can increase further and eventually leads aligned alleles to fix with slightly greater probabilities than opposing ones.

(*Appendix 3—figure 25*). Moreover, our analysis is insensitive to this particular choice (we only use it in comparing analytic and simulation results for the rapid phase).

The change in mean phenotype during the rapid phase is driven by the differential effect of directional selection on minor alleles whose effects are aligned and opposed to the shift in optimum (*Figure 3*). Considering a pair of minor alleles with opposing effects of the same magnitude and the same initial frequency, selection increases the frequency of the aligned allele relative to the opposing one. By the end of the rapid phase, the frequency differences across all aligned and opposing alleles drive the mean phenotype close to the new optimum (*Figure 2A*). Deviations from Lande's approximation manifest as prolonged, weak directional selection during the equilibration phase, which further increases the expected frequency difference between aligned and opposing alleles. However, given that we are considering a highly polygenic trait, the expected frequency difference between a pair of opposing alleles will be small. This small difference causes aligned alleles to have a slightly greater probability of eventually fixing during the equilibration phase (*Figure 3*). Over a period on the order of $2N$ generations (see below), the frequency differences between aligned and opposing alleles are replaced by a slight excess of fixed differences between them, and the equilibrium genetic architecture is restored around the new optimum. In the following sections, we describe these processes quantitatively: Specifically, we ask how the relative contribution of alleles to phenotypic change during the two phases depends on their effect size and initial frequency.

### The allelic response in the rapid phase

We can describe changes to allele frequencies during the rapid phase with a simple deterministic approximation. The duration of the rapid phase is much shorter than the time scale over which genetic drift has a substantial effect ($t_1 \sim 1/U \ll 2N$ generations; see *Equation 9*), allowing us to rely only on the first moment of change in allele frequency (*Equation 7*). Additionally, deviations of the distance $D(t)$ from Lande's approximation during this phase have negligible effects (*Figure 2B* and *Appendix 3—figure 8D–F*), allowing us to assume that $D(t) = D_L(t)$ (*Equation 5*). Lastly, when relative frequency changes are small, we can substitute the frequency in the first moment by its initial value. With these simplifications, we can integrate the first moment over time to obtain an explicit linear approximation for frequency changes.

Consider a pair of minor alleles with opposing effects of size $\pm a$ and initial frequency $x_0$ before the shift in optimum. Using our linear approximation, we find that the frequency difference between them at the end of the rapid phase is

$$\Delta x_{t_1}^*(a, x_0) \quad = x_{t_1}(a, x_0) - x_{t_1}(-a, x_0) \approx 2 \cdot (a/V_S) \cdot x_0(1 - x_0) \int_0^{t_1} D_L(t)dt \qquad (10)$$
$$= (\Lambda - D_L(t_1)) \cdot 2ax_0(1 - x_0)/V_A(0).$$

The contribution of the pair to the change in mean phenotype is

$$\Delta z_{t_1}^*(a, x_0) = 2a \cdot \Delta x_{t_1}^*(a, x_0) \approx (\Lambda - D_L(t_1)) \cdot 2 \cdot v^*(a, x_0)/V_A(0), \qquad (11)$$

where $v^*(a, x_0) = 2a^2 x_0(1 - x_0)$ is the contribution to variance of an allele with magnitude $a$ and frequency $x_0$. Thus, the pair's contribution to phenotypic change is proportional to its contribution to phenotypic variance before the shift in optimum.

The expected contribution of alleles with a given magnitude and initial frequency is therefore proportional to their expected contribution to phenotypic variance at equilibrium, before the shift occurs. We focus on the contribution of alleles divided by the mutation rate at which they are introduced into the population, that is the 'contribution per unit mutational input'. To this end, we measure the trait value in units of $\delta = \sqrt{V_S/(2N)}$ and express allele magnitudes in terms of the scaled selection coefficients at equilibrium (when $D = 0$); in these units $S = 2N s_e = a^2$ (see *Choice of units*). Expressing our results in this form makes them invariant with respect to changing the population size, $N$, stabilizing selection parameter, $V_S$, mutational input per generation, $2NU$, and distribution of magnitudes, $g(a)$. In these terms, the expected contribution of alleles with given magnitude and initial MAF to phenotypic change is

$$\Delta z_{t_1}(a, x_0) \approx (\Lambda - D_L(t_1)) \cdot v(a, x_0)/V_A(0), \tag{12}$$

and the marginal contribution of alleles with a given magnitude is

$$\Delta z_{t_1}(a) = \int_0^{1/2} \Delta z_{t_1}(a, x)dx \approx (\Lambda - D_L(t_1)) \cdot v(a)/V_A(0), \tag{13}$$

where $v(a, x_0) \approx 4a^2 \cdot \mathrm{Exp}\left[-a^2 x_0(1 - x_0)\right]$ and $v(a) \approx 4a^2 \cdot \int_0^{1/2} \mathrm{Exp}\left[-a^2 x(1 - x)\right] dx = 4a \cdot D_+(a/2)$ are the corresponding densities of variance per unit mutational input at equilibrium, and $D_+$ is the Dawson function (Section 3.2 of *Appendix 3*). The expected absolute contributions follow from multiplying these expressions by the mutational input per generation, $2NU \cdot g(a)$. Specifically, as we would expect, the total change in mean phenotype during the rapid phase is $2NU \cdot \Delta z_{t_1} = 2NU \cdot \int_0^\infty \Delta z_{t_1}(a) \cdot g(a)da \approx \Lambda - D_L(t_1)$, as $V_A(0) = 2NU \cdot \int_0^\infty v(a) \cdot g(a)da$.

The relative contribution of alleles with a given magnitude and initial MAF to phenotypic change follows from their expected contribution to variance at equilibrium (*Equations 12 and 13*, and *Figure 4*). The properties of $v(a)$ imply that (*Figure 4A*): (i) the relative contribution of alleles with small effect sizes ($a^2 \ll 1$) scale linearly with $S = a^2$ ($v(a) \approx 2a^2$, measured in units of $\delta^2$); (ii) the contribution of alleles with moderate and large effect sizes (roughly $S = a^2 > 3$) are much greater, and fairly insensitive to the effect size (with $v(a) \approx 4$); and (iii) the contribution is maximized for $a^2 \approx 10$ ($v(\sqrt{10}) \approx 5.2$) (see *Simons et al., 2018*, for intuition about these properties). While large and moderate effect alleles make similar contributions to phenotypic change, MAFs of large effect alleles before the shift are much lower than MAFs of moderate ones (*Figure 4B*), because they are subject to stronger stabilizing selection. The expected frequency difference between pairs of opposing alleles is greatest for moderate effect sizes (*Figure 4C*), because it is proportional to $E(2ax_0(1 - x_0)) \propto v(a)/a$ (*Equation 10*), and $v(a)$ is similar for moderate and large effect sizes. Additional properties of the allelic response during the rapid phase are presented in Section 4 of *Appendix 3*.

When the polygenicity is low and/or the shift in optimum or effect sizes are large our linear approximation becomes less accurate (*Figure 4A*). Specifically, minor alleles exhibit large relative changes in frequency such that substituting the initial MAF for the frequency in *Equation 7* for the 1st moment is inaccurate. In Section 4.1.2 of *Appendix 3* we derive a nonlinear approximation that is more accurate in these cases (*Figure 4A* and *Appendix 3—figures 9* and *10*). Nonetheless, the qualitative behaviors we outlined remain intact.

## The allelic response in the equilibration phase

Over the long run, the small frequency differences between opposite alleles that accrue during the rapid phase translate into small differences in their fixation probabilities (*Figure 3*). In the non-Lande case, prolonged weak directional selection during the equilibration phase amplifies these differences in fixation probabilities. We approximate fixation probabilities in two steps. First, we model the effect of directional selection on frequency as an instantaneous, deterministic pulse. Second, we apply the diffusion approximation for the fixation probability (*Ewens, 2004*, Chapter 4), assuming stationary stabilizing selection ($D = 0$), genetic drift, and the initial frequency after the pulse. We further assume that the relative changes in allele frequencies due to directional selection are small, such that we can use approximations that are linear in this change; but in Section 5 of *Appendix 3*, we derive nonlinear approximations that relax this assumption.

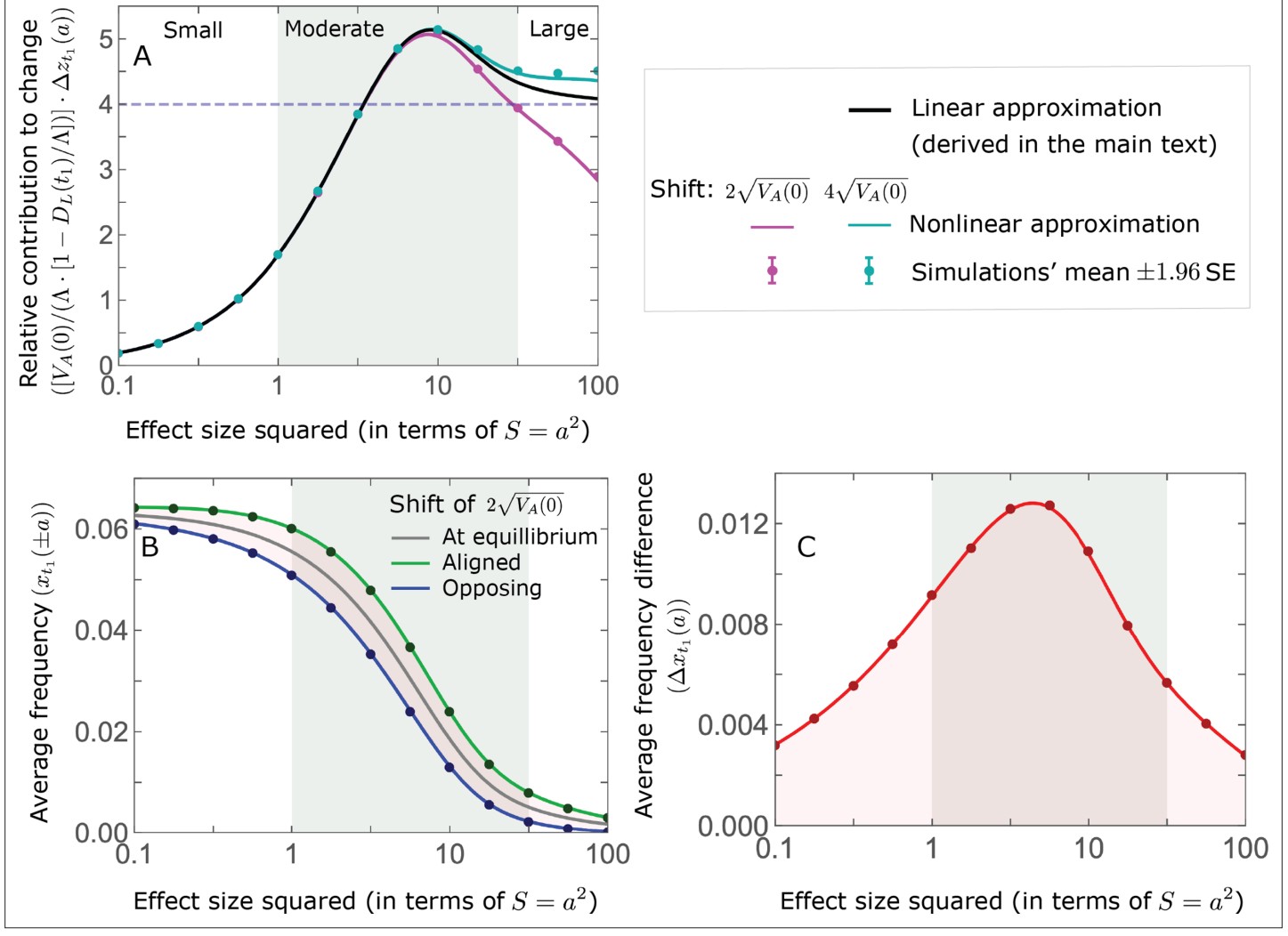

**Figure 4.** The allelic response during the rapid phase. (**A**) Alleles with moderate and large magnitudes make the greatest contribution to phenotypic change (per unit mutational input). The results of our linear approximation (derived in the main text) are compared with a more accurate nonlinear one (derived in Section 4.1.2 of *Appendix 3*) and with simulations. (**B**) The average MAF of aligned and opposing alleles at the end of the rapid phase decreases with effect size squared. (**C**) The expected frequency difference between pairs of opposing alleles is greatest for moderate effect sizes. In B and C, the results of the nonlinear approximation are compared with simulations. The simulation results in all panels were were generated using the *single allele* simulation, as detailed in *Simulations and resources*; error bars are not visible because they are smaller than the points.

## The Lande case

When Lande's approximation is accurate, directional selection is non-negligible only briefly after the shift. This justifies approximating its effects as if they were caused by an instantaneous pulse. It also suggests that mutations that arise after the shift in optimum contribute negligibly to phenotypic change, because when directional selection is non-negligible, few of them arise and their fixation probabilities are tiny (given that they start from an initial frequency of $1/2N$).

Consider a pair of opposite minor alleles, with magnitude $a$ and initial frequency $x_0$. Analogously to our derivations for the rapid phase (*Equations 10*, *11* and *12*) by modeling the effects of directional selection on their frequencies as an instantaneous pulse, and assuming that these effects are small, we find that the resulting frequency differences between them is approximated by

$$\Delta x_d^*(a, x_0) \equiv x_d(a, x_0) - x_d(-a, x_0) \approx 2ax_0(1 - x_0) \cdot \int_0^\infty (D_L(\tau)/V_S)d\tau$$

$$= 2ax_0(1 - x_0) \cdot \Lambda/V_A(0).$$

(14)

Consequently, the pair's expected contribution to phenotypic change is approximated by

$$\Delta z_d^*(a, x_0) \equiv 2a \cdot \Delta x_d^*(a, x_0) \approx 2\Lambda \cdot v^*(a, x)/V_A t(0), \tag{15}$$

and the contribution of such pairs per unit mutational input is approximated by

$$\Delta z_d^L(a, x_0) \approx \Lambda \cdot v(a, x)/V_A(0), \tag{16}$$

where, as before, $v^*(a, x) = 2a^2 x(1 - x)$ is an allele's contribution to genetic variance and $v(a, x)$ is the density of variance per unit mutational input at equilibrium, and we use the superscript $L$ to denote that this applies to the Lande case.

We approximate a pair's expected long-term, fixed contribution to phenotypic change by calculating the difference in fixation probabilities of the opposite alleles given their frequency after the pulse, again assuming that the effects of the pulse are small. Namely,

$$
\begin{aligned}
\Delta z_\infty^*(a, x_0) &\approx 2a \left( \pi(a, x_d(a, x_0)) - \pi(a, x_d(-a, x_0)) \right) \\
&\approx 2a \cdot \frac{\partial \pi}{\partial x}(a, x_0) \cdot \Delta x_d^*(a, x_0) = \frac{\partial \pi}{\partial x}(a, x_0) \cdot \Delta z_d^*(a, x_0).
\end{aligned}
\tag{17}
$$

where $\pi(a, x)$ denotes the fixation probability of an allele with magnitude $a$ and initial frequency $x$ under stationary stabilizing selection and drift. In Section 3 of *Appendix 3* we derive the diffusion approximation for $\pi(a, x)$ and show that

$$\frac{\partial \pi}{\partial x}(a, x_0) = 2f(a)/v(a, x_0), \tag{18}$$

where $f(a) \equiv 2a^3 \cdot \mathrm{Exp}\left[-a^2/4\right] / \left(\sqrt{\pi} \cdot \mathrm{Erf}\left[a/2\right]\right)$. From *Equation 14-18*, we find that the expected fixed contribution per unit mutational input of pairs of alleles is

$$\Delta z_\infty^L(a, x_0) \approx \frac{\partial \pi}{\partial x}(a, x_0) \cdot \Delta z_d^L(a, x_0) \approx 2\Lambda \cdot f(a)/V_A(0). \tag{19}$$

Note that this expression does not depend on the initial frequency! The expected marginal contribution of alleles with a given magnitude follows and is

$$\Delta z_\infty^L(a) = \int_0^{1/2} \Delta z_\infty(a, x) dx \approx \Lambda \cdot f(a)/V_A(0). \tag{20}$$

Hence, the function $f$ approximates how the relative long-term contribution of alleles depends on their magnitudes (*Figure 5A*).

We expect the total long-term allelic contribution to equal the shift in optimum, $\Lambda$. In our linear Lande approximation, the total contribution is

$$
\begin{aligned}
2NU \cdot \Delta z_\infty^L &= 2NU \cdot \int_0^\infty \Delta z_\infty^L(a) \cdot g(a) da \approx \Lambda \cdot \frac{2NU \cdot \int_0^\infty f(a) \cdot g(a) da}{V_A(0)} \\
&= \Lambda \cdot \frac{\int_0^\infty f(a) \cdot g(a) da}{\int_0^\infty v(a) \cdot g(a) da} = \frac{\Lambda}{1 + C},
\end{aligned}
\tag{21}
$$

where we use the fact that $V_A(0) = 2NU \cdot \int_0^\infty v(a) \cdot g(a) da$ and define

$$C \equiv \frac{\int_0^\infty v(a) \cdot g(a) da}{\int_0^\infty f(a) \cdot g(a) da} - 1. \tag{22}$$

$C$ measures the extent to which our approximation underestimates the long-term phenotypic change. We note that $C > 0$ for any distribution of allele magnitudes, because $v(a) > f(a)$ for any magnitude $a$ (*Figure 5A*), and further that $C \ll 1$ is a necessary condition for our approximation to be accurate. Given that $v(a)$ is appreciably greater than $f(a)$ only for $a^2 \gtrsim 4$ (*Figure 5A*), this condition implies that for our approximation to be accurate, the vast majority of the genetic variance at equilibrium must arise from alleles with $a^2 < 4$.

When alleles with larger effects contribute substantially to the variance at equilibrium (and $C$ is appreciable), Lande's approximation becomes inaccurate. The prevalence of large effect alleles leads to a quasi-static decay of the mean distance from the new optimum, $D$, during the equilibration

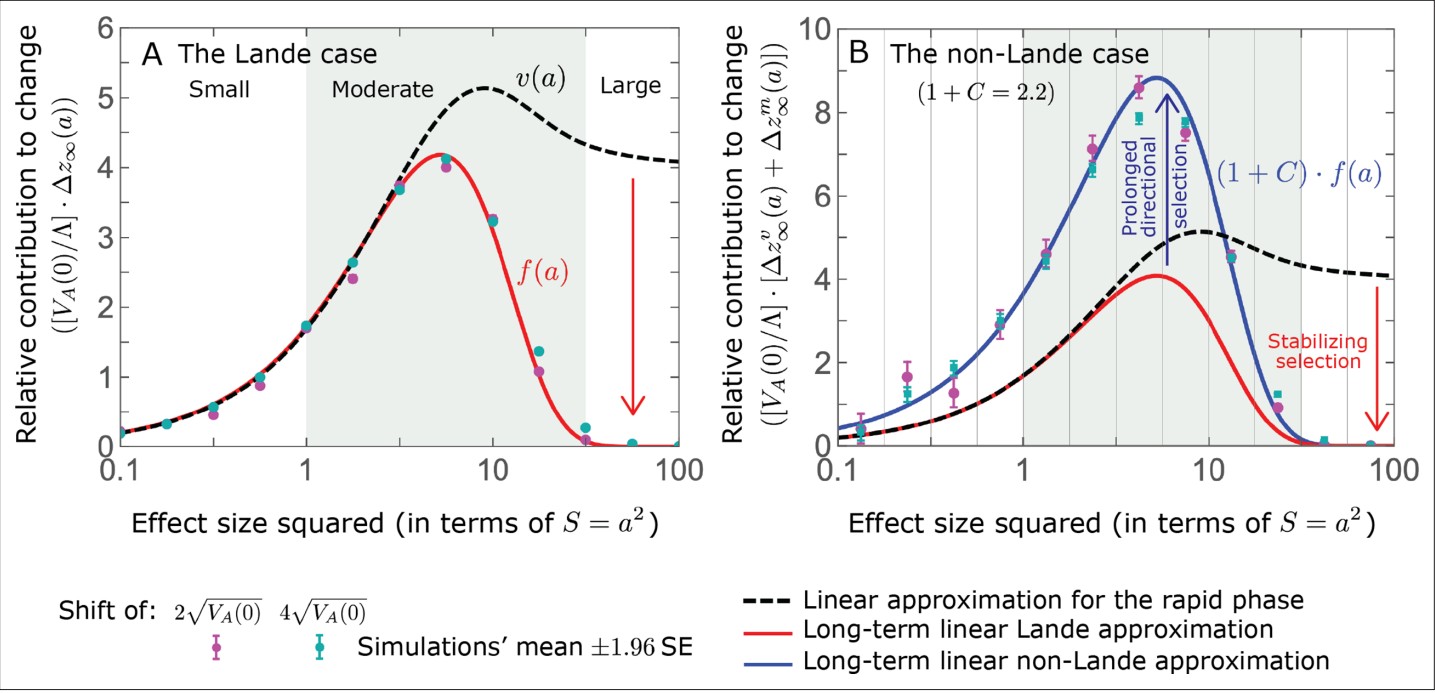

**Figure 5.** The long-term (fixed) allelic contribution to phenotypic adaptation. We show the relative contribution of alleles as a function of effect size squared, based on the linear approximations and on simulations with the two shift sizes specified in the caption. (**A**) The Lande case. The theoretical prediction is described by the function $f(a)$ (*Equations 13* and *20*), and simulation results were generated using the *single allele* simulation, as detailed in *Simulations and resources*; error bars are not visible because they are smaller than the points. Our prediction for the long-term contribution (corresponding to $f(a)$) is always below the prediction for the rapid phase (corresponding to $v(a)$). The difference becomes substantial for $a^2 \gtrsim 4$, implying that the linear Lande approximation underestimates the fixed contribution when large effect alleles contribute markedly to the genetic variance at equilibrium. (**B**) The non-Lande case. Here, we assume an exponential distribution of effect sizes squared with $E(a^2) = 16$, which yields an amplification factor of $1 + C \approx 2.2$ (*Equation 22*). The theoretical prediction for the joint contribution of standing variation and new mutations is described by the function $(1 + C) \cdot f(a)\Lambda/V_A(0)$ (*Equation 28*). Simulation results were generated using the *all alleles* simulation for the non-Lande case (see section on *Simulations and resources*). Specifically, we calculated the relative contribution of alleles in each effect size bin (between the gray gridlines), by dividing the contribution of all fixations in the bin by the mutation rate per generation corresponding to that bin. In both Lande and non-Lande cases, long-term stabilizing selection diminishes the contribution of alleles with large effects (red arrows). In the non-Lande case, long-term, weak directional selection greatly amplifies the contribution of alleles with small and moderate effects (blue arrow). See *Appendix 3—figures 12–19* for other attributes of the long-term allelic response and for the nonlinear approximations.

phase (*Figure 2D* and the section on the *Phenotypic response*). For the distance during the equilibration phase, and therefore for the deviations from Lande's phenotypic approximation to be substantial, would require that $C \gg 1$ (see Section 6 of *Appendix 3*), which implies that the vast majority (e.g. > 90%) of the genetic variance at equilibrium arises from alleles with large effect sizes, say with $a^2 \gg 4$ (see *Figure 5A* and *Equation 22*). Even a small distance $D$ during the equilibration phase, however, would result in prolonged, weak directional selection that could markedly amplify the difference in fixation probabilities between opposite alleles. Our linear Lande approximation does not account for this amplification, and it could therefore greatly underestimate the total long-term allelic contribution.

## The non-Lande case

We can, however, extend our approximation to account for the amplification in the non-Lande case. To this end, we modify our instantaneous pulse approximation for a pair of opposite alleles (*Equation 14*) to

$$\Delta x_d^*(a, x_0) \equiv x_d(a, x_0) - x_d(-a, x_0) \approx 2ax_0 \left(1 - x_0\right) \cdot (1 + A) \cdot \Lambda/V_A(0), \tag{23}$$

where the factor $A > 0$ accounts for the greater effect of directional selection relative to the Lande case (with the caveat that the justification for the instantaneous pulse approximation is less obvious

in the non-Lande case, given prolonged, weak directional selection; see Section 5.2 of *Appendix 3*). Following the same steps as taken in the Lande case, we then find that the expected long-term (fixed) contribution per unit mutational input of pairs of alleles with a given magnitude and initial MAF is

$$\Delta z_\infty\left(a, x_0\right) \approx 2 \cdot (1 + A) \cdot \Lambda \cdot f(a)/V_A(0) \tag{24}$$

and that their expected marginal contribution for a given magnitude is

$$\Delta z_\infty^v\left(a\right) \approx (1 + A) \cdot \Lambda \cdot f(a)/V_A(0), \tag{25}$$

where the superscript $v$ denotes that these contributions originate from variation that segregated before the shift in optimum.

In the non-Lande case, the fixation of mutations that arise after the shift in optimum can also contribute substantially (*Appendix 3—figure 21B*), because prolonged, weak directional selection can produce a substantial difference in the numbers of fixations of mutations with opposite effects. In Section 5.2.2 of *Appendix 3*, we follow the same approach that applied for standing variation to show that the relative long-term contribution of new mutations with a given magnitude can be approximated by

$$\Delta z_\infty^m(a) \approx B \cdot \Lambda \cdot f\left(a\right) /V_A(0), \tag{26}$$

where the factor $B > 0$ does not dependent on the magnitude (Section 5.2.2 of *Appendix 3*). Thus, similar to what we found in the Lande case, the function $f$ approximates how the relative long-term contribution of alleles depends on their magnitudes, but here it applies to both standing variation and new mutations (*Figure 5B*).

To gain further understanding of the non-Lande case, we consider the joint contribution of standing variation and new mutations. Equating the total contribution with the shift in optimum we find that

$$2NU \cdot \Delta z_\infty = 2NU \cdot \left(\Delta z_\infty^v + \Delta z_\infty^m\right) \approx \frac{1 + A + B}{1 + C} \cdot \Lambda = \Lambda, \tag{27}$$

with $C$ defined in *Equation 22*. This implies that $A + B = C$ and that the proportional contributions of standing variation and new mutations are $(1 + A)/(1 + C)$ and $B/(1 + C)$, respectively. It also implies that the contribution per unit mutational input of alleles with a given magnitude $a$ is

$$\Delta z_\infty(a) = \Delta z_\infty^v(a) + \Delta z_\infty^m(a) \approx (1 + C) \cdot \Delta z_\infty^L(a) = (1 + C) \cdot \Lambda \cdot f\left(a\right) /V_A(0). \tag{28}$$

Thus, in the linear non-Lande approximation, prolonged weak directional selection amplifies the relative contribution of alleles of any given magnitude by the same factor of $(1 + C)$ (*Figure 5B* and *Appendix 3—figure 19*). $C$ is therefore an allelic measure of the deviation from Lande's approximation (see *Appendix 3—figures 20* and *21*), and, intriguingly, it depends only on the distribution of mutation magnitudes (*Equation 22*).

In Section 5 of *Appendix 3*, we show that the linear approximations are accurate when $a \cdot (1 + A) \cdot \Lambda/V_A(0) \ll 1$ (with $A = 0$ in the Lande case) and we derive nonlinear approximations that are more accurate when this condition is violated (*Figure 4A* and *Appendix 3—figures 13* and *16*). When polygenicity is low, the shift in optimum is large, or effect sizes are large, directional selection causes large relative changes in MAFs, such that the use of the initial MAF in the instantaneous pulse approximation (i.e. in *Equations 14* and *23*) and the Taylor approximation of fixation probabilities (*Equation 17*) become inaccurate. Even in these cases, however, the linear approximations capture the salient features of the long-term allelic contribution to phenotypic adaptation (*Figure 5* and *Appendix 3—figures 13–19*).

## Turnover in the genetic basis of adaptation

Notably, our linear approximations capture the dramatic turnover in the genetic basis of adaptation during the equilibration phase (*Figure 6*). In the long run, the short-term contribution of large effect alleles ($S = a^2 \gtrsim 30$) is almost entirely wiped out, and is supplanted by the contribution of moderate effect alleles ($a^2 \approx 5$) (*Figure 6A* and *Appendix 3—figures 15* and *19*). Moreover, for any given magnitude, the proportional long-term contribution of minor alleles that segregated at low frequencies before the shift is diminished relative to their short-term contribution, all the more so for large effect sizes (*Figure 6B* and *Appendix 3—figure 14*). For instance, for an effect size $a^2 = 35$, minor alleles with initial frequencies below

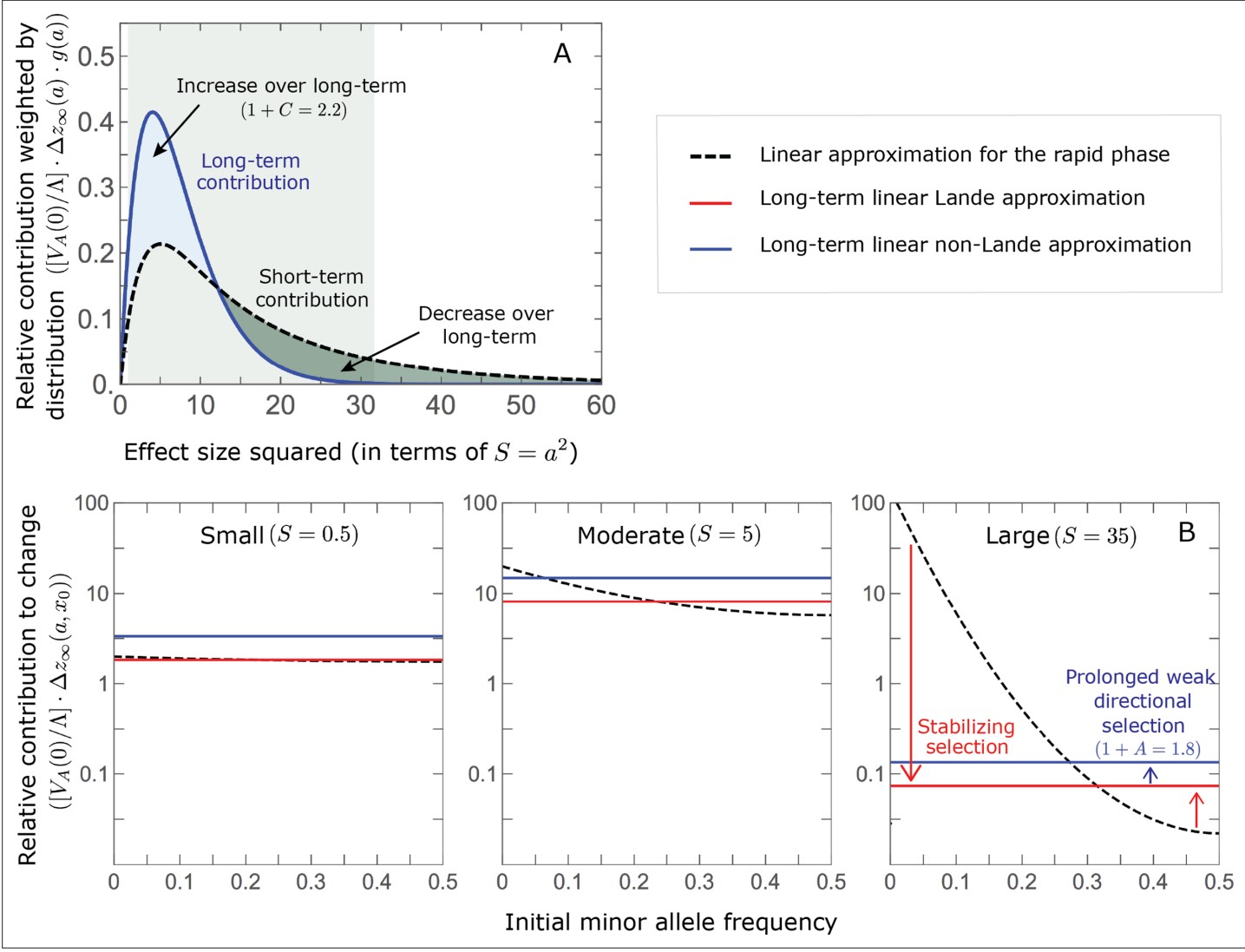

**Figure 6.** The genetic basis of adaptation turns over during the equilibration phase. (**A**) The short-term contribution of large effect alleles is supplanted by the contribution of moderate effect alleles. As an illustration, we show the results of the linear approximation for the non-Lande case in *Figure 5B* (*Equations 13* and *28*). Specifically, we weight the short- and long-term relative contributions by the density of effect size squared (given $E(a^2) = 16$) and use a linear (rather than log) scale for the effect sizes squared. This way we can see that the decrease in the contribution of large effect alleles (shaded dark gray area) equals the increase in the contribution of moderate effect alleles (shaded blue area). (**B**) The proportional long-term contribution of alleles that segregated at low MAFs before the shift is diminished relative to their short-term contribution, an effect most pronounced for large effect sizes. As an illustration, we show the linear approximations for the contribution of alleles with a given effect size as a function of their initial MAF (*Equations 12*, *19* and *24*) for the same non-Lande case as in **A**, with a shift of $\Lambda = 2\sqrt{V_A(0)}$. To this end, we estimate the amplification factor for standing variation $(1 + A)$ using *all alleles* simulations for the non-Lande case (see section on *Simulations and resources* and Section 5.2 of *Appendix 3*). In both the Lande and non-Lande cases, long-term stabilizing selection diminishes the contribution of alleles with lower initial MAF and amplifies the contribution of alleles with higher initial MAF (red arrows). In the non-Lande case, prolonged, weak directional selection amplifies the contribution of alleles, regardless of their initial MAF (blue arrow).

0.05 account for more than 99% of the short-term contribution but for only ~10% of the (much smaller) long-term contribution (*Figure 6B* and *Appendix 3—figure 14*).

We can understand this turnover by considering the effects of stabilizing selection during the equilibration phase (*Appendix 2—figure 2*). As noted, stabilizing selection on the trait induces selection against minor alleles, which weakens as MAF increases and vanishes at MAF = 1/2. Now consider how it affects a pair of alleles with opposite, moderate or large effect. If their initial frequencies are very low, both alleles will have low MAFs at the end of the rapid phase. Consequently, they will both be strongly selected against during the equilibration phase and will almost certainly go extinct (*Appendix 2—figure 2*). In the long

run, their expected contribution to phenotypic adaptation is therefore diminished. In contrast, if the alleles' initial MAF is sufficiently high, the relative increase in the aligned allele's frequency by the end of the rapid phase causes it to be subject to substantially weaker selection than is the opposing allele. In the extreme in which the aligned allele has exceeded frequency 1/2, the direction of selection on it is even reversed (*Appendix 2—figure 2*). In such cases, the pair's expected contribution to phenotypic adaptation will be amplified. This reasoning suggests that, for a given magnitude, there is a critical initial MAF such that the long-term contribution of alleles that start above it is amplified and the contribution of those that start below it is diminished (*Figure 6B* and *Appendix 2—figure 3*). This critical frequency is lower in the non-Lande case, because prolonged, weak directional selection amplifies the long-term contribution from alleles with any initial MAF.

The turnover among alleles with different magnitudes can be explained in similar terms. Alleles with large magnitudes almost always start from low MAF, because they are subject to strong stabilizing

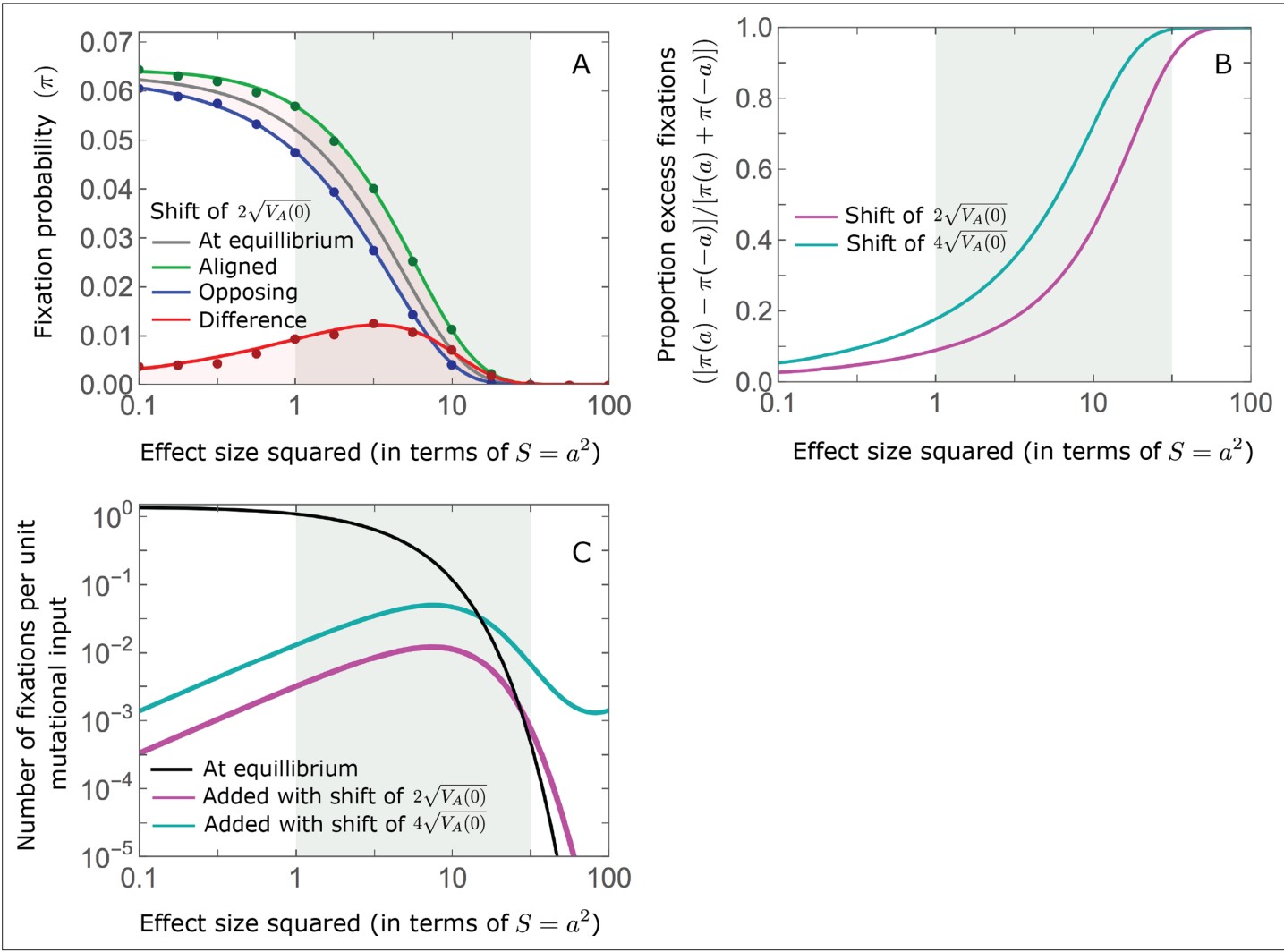

**Figure 7.** While long-term phenotypic adaptation arises from an excess in fixations of aligned relative to opposing alleles, this excess and its effect on the total number of fixations is typically small. (**A**) The fixation probabilities of aligned and opposing alleles segregating before the shift, as a function of their effect size squared. Simulation results were generated using the *single alleles* simulation, as detailed in *Simulations and resources*; error bars are not visible because they are smaller than the points. Analytic predictions in all panels were calculated using the nonlinear Lande approximation derived in Section 5.3 of *Appendix 3*. For large effect sizes, the fixation probabilities become vanishingly small, whereas for small and moderate effect sizes, the difference in the fixation probabilities of alleles with opposing effects is small. (**B**) The relative excess of fixations of aligned alleles as a function of effect size squared. (**C**) Polygenic adaptation typically adds a small number of fixations relative to the number at equilibrium. For large effect sizes, the relative increase in number is large, but their absolute number and the corresponding contribution to phenotypic adaptation are extremely small.

selection before the shift (*Figure 4B*). Consequently, they are highly unlikely to exceed the critical initial frequency and their expected long-term contribution to phenotypic adaptation is diminished (*Figures 5A* and *6A*). However, changes to the frequencies of these alleles skew the phenotypic distribution, leading to prolonged, weak directional selection that amplifies the long-term contribution of small and moderate effect alleles (*Figure 6B*). In our linear approximation, this amplification will occur for allele magnitudes that satisfy $(1 + C) \cdot f(a) \gtrless v(a)$ (*Figure 5B*). These considerations explain why the contributions of alleles with small and moderate effects supplant those of alleles with large effects (*Figures 5A* and *6A*). They also highlight that deviations from Lande's approximation are critical to understanding the allelic response, even when they have small phenotypic effects.

## Other properties of the equilibration process

While long-term phenotypic adaptation arises from an excess in fixations of aligned relative to opposing alleles, this excess and the increase in the total number of fixations are typically small relative to the number of fixations at equilibrium (*Figure 7*). The relative excess of aligned fixations (defined as $(\pi(a) - \pi(-a))/(\pi(a) + \pi(-a))$) and the relative increase in total fixations (compared to the expected number at equilibrium) both decrease with increased polygenicity and increase with the shift in optimum and allele magnitude (*Figure 7B and C*). For sufficiently large effect sizes, practically all fixations are caused by the shift and are of aligned alleles (*Figure 7B and C*). However, with the exception of extreme cases in which the contribution of alleles of small and moderate effects to genetic variance is negligible, the number of fixations of such large effect alleles and their contribution to phenotypic change will be small (*Figures 5* and *7*). Typically, most fixations and contribution to phenotypic change will arise from alleles with small and moderate effects (*Figures 5* and *7*), for which the proportional excess of aligned and total fixations is modest (*Figure 7B and C*).

In the long run, these fixations move the mean phenotype all the way to the new optimum, and genetic variation around the new optimum returns to equilibrium. A proxy for the approach to equilibrium is the 'fixed distance from the optimum', defined as the phenotypic distance of an individual that is homozygous for the ancestral allele at every segregating site; at equilibrium, we expect the fixed distance to be 0. Our simulations suggest that, under a broad range of parameter values, the change in fixed distance after the shift is well approximated by an exponential decay with a rate of $1/(2N)$ per generation (*Figure 8* and *Appendix 3—figure 28*). This approximation is remarkably accurate in the Lande case. In the non-Lande case, the decay is initially slower than the approximation suggests, possibly because the long-term contribution of new mutations (as opposed to standing variation)

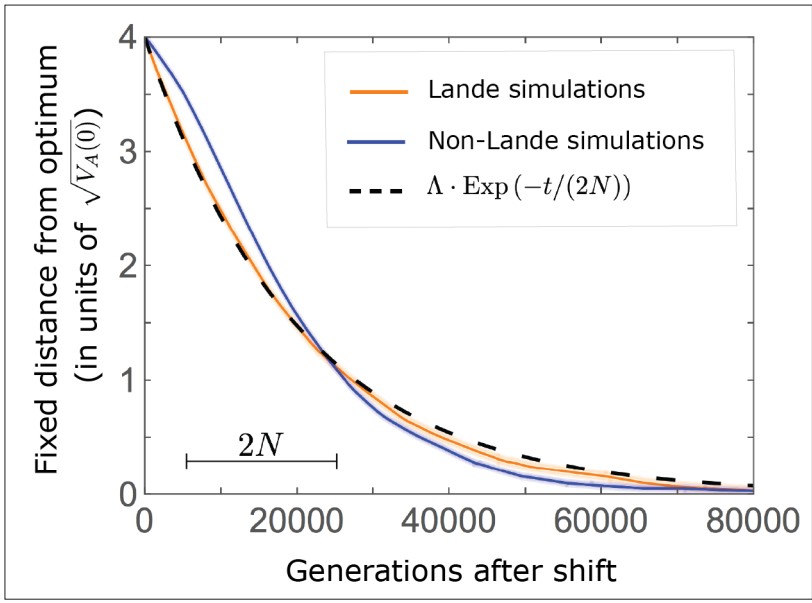

**Figure 8.** Equilibrium around the new optimum is restored on a time scale of $2N$ generations after the shift. The change in fixed distance after the shift estimated using *all alleles* simulations for the Lande and non-Lande cases (see section on *Simulations and resources*) is compared with an exponential decay with a rate of $1/(2N)$ per generation. See *Appendix 3—figure 28* for similar comparisons using a broad range of model parameter values.

takes longer to amass. In both cases, the return to equilibrium occurs on a time scale of $2N$ generations after the shift (*Figure 8* and *Appendix 3—figure 28*).

## Discussion

Here, we investigated the phenotypic and genetic adaptive response to selection on a highly polygenic quantitative trait in a simple yet highly relevant setting, in which a sudden change in environment shifts the trait's optimal value. The phenotypic response to selection was previously studied by *Lande, 1976*. Assuming that phenotypes are normally distributed in the population, he predicted that after the shift the population's mean phenotype will approach the new optimum exponentially, at a rate that is proportional to the additive genetic variance in the trait. The Normality assumption, however, was only shown to hold in an infinitesimal limit, in which the effect sizes of individual loci are infinitesimally small (see *Turelli, 2017* and Section 8 of *Appendix 3*). We found that when the trait is sufficiently polygenic and the shift in optimum is not too large (relative to genetic variance in the trait), Lande's prediction is accurate so long as the genetic variance is dominated by loci with small and moderate effect sizes, which are defined based on the selection acting on them before the shift. When these conditions are violated, most notably when loci with large effects contribute markedly to genetic variance, the initial, rapid change in mean phenotype is followed by a pronounced quasi-static phase, governed by changes to the 3rd central moment of the phenotypic distribution, in which the mean phenotype takes much longer to catch up to the new optimum.

We also characterized the genetic basis of these adaptive phenotypic changes. The closest previous work assumed an infinite population size (*de Vladar and Barton, 2014*; *Jain and Stephan, 2015*; *Jain and Stephan, 2017a*; *Jain and Stephan, 2017b*). As we show, relaxing this assumption leads to entirely different behavior. In infinite populations, small effect alleles, whose equilibrium frequencies are dominated by mutation and are held at 1/2 before the shift, make the greatest contribution to phenotypic change after the shift (see Introduction). In contrast, in most if not all real (finite) populations (with $N_e \ll 1/u$, where $u$ is the mutation rate per site per generation), the frequencies of such small effect alleles are dominated by genetic drift rather than mutation. More generally, variation in allele frequencies due to genetic drift, which is absent in infinite populations, critically affects the allelic response to selection.

To study the allelic response, we divided it into two periods: a rapid phase, immediately after the shift, and a subsequent, prolonged equilibration phase. During the rapid phase, the population's mean distance to the optimum is substantial and changes rapidly. Directional selection on the trait increases the frequency of minor alleles whose effects are aligned with the shift relative to minor alleles with opposing effects (given the same magnitude and initial frequency). By the end of the rapid phase, the cumulative effect of these frequency differences pushes the mean phenotype close to the new optimum, but because this effect is spread over myriad alleles, the frequency difference between any individual pair of opposing alleles is fairly small. Specifically, we found that an allele's contribution to phenotypic change is proportional to its contribution to phenotypic variance before the shift, implying that alleles with moderate and large effect sizes make the greatest per site contributions to phenotypic change, while alleles with moderate effect sizes experience the greatest frequency changes. The expected frequency differences between opposing alleles is amplified by prolonged, weak directional selection during the subsequent equilibration phase, and this amplification is pronounced when the phenotypic approach to the new optimum deviates markedly from Lande's approximation.

Over the long run, stabilizing selection on the trait and genetic drift transform these small frequency differences into a small excess of fixed aligned alleles relative to opposing ones, and cumulatively this excess moves the population mean all the way to the new optimum. This transformation process involves a massive turnover in the properties of the contributing alleles. Notably, the transient contributions of large effect alleles are supplanted by contributions of fixed moderate, and to a lesser extent, small effect alleles. In the non-Lande cases, the fixation of mutations that arise after the shift in optimum can also contribute substantially to long-term phenotypic adaptation. These processes take on the order of $2N_e$ generations, after which the equilibrium architecture of genetic variation around the new optimum is restored.

Our finding that large effect alleles almost never sweep to fixation appears at odds with the results of previous studies of similar models. These discrepancies are largely explained by earlier papers considering settings that violate our assumptions, notably about evolutionary parameter ranges. For instance, some studies assume that large effect alleles segregate at high frequencies before the shift in optimum (e.g.

*Christodoulaki et al., 2019*), which is presumably uncommon in natural populations and in any case, violates our assumption that the population is at mutation-selection-drift equilibrium before the shift. Other models implicitly consider quantitative traits of intermediate genetic complexity; while such traits likely exist, there are to our knowledge few well-established examples. Notably, *Thornton, 2019* observes sweeps in cases in which the trait is not highly polygenic (violating our assumption that $\sqrt{2NU} \gg 1$). Relatedly, *Chevin and Hospital, 2008* observe sweeps in cases in which a rare mutation of large effect contributes substantially to genetic variance, which violates our assumptions that genetic variation is highly polygenic and is not predominantly effectively neutral (i.e. that alleles with $S \gtrsim 1$ contribute substantially). Although it remains to be seen, we believe that this architecture is much less common, given mounting evidence, reviewed in the *Introduction*, which suggests that traits are often highly polygenic, and other considerations, notably estimates of persistence time (*Walsh and Lynch, 2018*; *Sella and Barton, 2019*) and inferences based on human GWASs (*Simons et al., 2018*; *Zeng et al., 2018*), which indicate that quantitative genetic variation is not predominantly neutral.

Lastly, *Stetter et al., 2018* considered a huge shift in the optimal trait value (e.g. of ~90 phenotypic standard deviations), resulting in a massive drop in fitness (violating our assumption that $\Lambda \lesssim \sqrt{V_S}$)— although shifts in optimum need not be that large to result in the fixations of some large effect alleles. While there are many examples of rapid and large environmental fluctuations, e.g., seasonal fluctuations or shifting weather systems, they occur on a much shorter time scale than fixation (although they might have some effect on genetic architecture; see below). In turn, little is known about the magnitude of shifts in optimal trait values over the time scales of large effect, beneficial fixations. While it seems plausible that moderate shifts, which fall within our assumed parameter ranges, are common, we cannot rule out that larger shifts are common as well. The response to such larger shifts is not covered by our analysis and clearly warrants further study.

Other factors that we have not considered may also affect polygenic adaptation. Most notable among them is pleiotropy. Given that quantitative genetic variation affecting one trait often affects many other traits (*Bulik-Sullivan et al., 2015*; *Pickrell et al., 2016*; *Boyle et al., 2017*; *Liu et al., 2019*; *Sella and Barton, 2019*), alleles that would have been positively selected because of their effect on the trait under directional selection may be selected against because of their adverse effects on other traits. Moreover, pleiotropy is known to affect the genetic architecture of a given trait at equilibrium (*Simons et al., 2018*), which we have shown to shape the allelic response to selection on that trait. Pleiotropy is therefore likely to affect which alleles contribute to phenotypic change at the different phases of polygenic adaptation (see *Otto, 2004*, for related considerations for genetically simple traits). Linkage disequilibrium (LD) may have an effect as well, perhaps most notably for minor alleles with large effects, which start at low frequencies and experience strong directional selection during the rapid phase. Before the shift, large effect alleles located in genomic regions with low recombination and high functional density are more likely to be in LD with, for example, alleles with countervailing effects on the focal trait (*Lande, 1975*) or deleterious effects on other traits. If this were the case, then directional selection during the rapid phase might be effectively weaker, because it would act on extended haplotypes rather than on individual alleles.

In addition, the demography of a population, notably its size, and the selection pressures on quantitative traits are likely to change over a shorter time scale than it takes the genetic architecture of complex traits to equilibrate. When these changes occur over the $\sim 2N_e$ generations preceding a shift in optimal trait value, they may affect the genetic architecture of the trait and consequently its response to selection. Changes in population size influence the number of segregating sites affecting a trait and the distribution of their frequencies and contributions to variance, with more recent population sizes affecting strongly selected variation more than weakly selected variation (*Lohmueller, 2014*; *Simons et al., 2014*; *Simons and Sella, 2016*; *Sella and Barton, 2019*). The effects of varying selection will depend on the attributes of this variation in ways that await further study.

While the effects of all of these factors on the response to a shift in optimum warrant investigation, we expect the response to follow from the principles we outlined. Notably, we expect the short-term contribution of alleles to phenotypic change to be proportional to their contribution to variance before the shift, and their long-term contribution to arise from differences between the fixation probabilities of alleles with opposite effects, caused by the opposing effects of directional selection on

their frequencies. Thus, while all these factors are likely to affect the response, we expect the main features of the dynamics we portrayed to remain largely intact. These features include the role of the 3rd central moment of the phenotypic distribution in slowing down phenotypic adaptation near the new optimum; the transient contribution of large effect alleles to phenotypic adaptation; and the long-term importance of alleles with moderate effects.

As polygenic adaptation in quantitative traits is likely ubiquitous, our conclusions have potentially important implications. One is that, contrary to adaptation mediated by selective sweeps of initially rare, large effect, beneficial alleles (*Smith and Haigh, 1974*; *Kaplan et al., 1989*; *Braverman et al., 1995*; *Hermisson and Pennings, 2005*; *Coop and Ralph, 2012*; *Berg and Coop, 2015*), polygenic adaptation might have minor effects on patterns of neutral diversity at any given point in time (but may affect temporal diversity patterns; *Buffalo and Coop, 2019*; *Buffalo and Coop, 2020*). The effects of selected alleles on neutral diversity at linked loci follow from their trajectories (*Barton, 2000*). Our results indicate that directional selection on a highly polygenic trait introduces only small changes to allele frequencies at individual loci, which amount to minor perturbations to the allele trajectories expected under stabilizing selection at equilibrium (also see *Chevin and Hospital, 2008*; *Thornton, 2019*). Indeed, alleles with large effects exhibit only small, transient changes. For those with more moderate effects, there is a modest, long-term excess of fixations of those alleles whose effects are aligned with the shift relative to those whose effects are opposed, accompanied by a small increase in the total number of fixations (*Figure 7*). The trajectories of the alleles that fix are largely driven by weak stabilizing selection and tend to be drawn out (*Figure 8*). Thus, our results indicate that the effects of polygenic adaptation on neutral diversity should be minor (other than perhaps for massive shifts in optimal trait values, as noted above).

In contrast, long-term stabilizing selection on quantitative traits likely has substantial effects on neutral diversity patterns. Specifically, selection against minor alleles induced by stabilizing selection may well be a major source of background selection and is expected to affect neutral diversity patterns in ways that are similar to those of background (purifying) selection from other selective origins (*Charlesworth et al., 1993*; *Hudson and Kaplan, 1995*; *Santiago and Caballero, 1998*; *McVean and Charlesworth, 2000*).

Another implication of our results pertains to the search for the genetic basis of human adaptation, as well as adaptation in other species. Efforts to uncover the identity of individual adaptive genetic changes on the human lineage were guided by the notion that their identity would offer insight into what 'made us human'. Under the plausible assumption that many adaptive changes on the human lineage arose from selection on complex, quantitative traits, this approach may not be as informative as it appears (*Pritchard et al., 2010*; *Boyle et al., 2017*). Our results indicate that after a shift in the optimal trait value, the number of fixations of alleles whose effects are aligned with the shift are typically nearly equal to the number of alleles that are opposed (*Figure 7A*). Moreover, the alleles that fix are a largely random draw from the vastly greater number of alleles that affect the trait, both in the sense of being those that happened to segregate at high MAFs at the onset of selection and because of the stochasticity of fixation. Thus, in this plausible scenario, it becomes meaningless to say that any given fixation was adaptive, and arguably uninteresting to focus on the particular subset of alleles that happened to reach fixation. In contrast, identifying the traits that experienced adaptive changes promises to provide important insights. Recent efforts to do so pool the signatures of frequency changes over many loci that were found to be associated with a given trait in GWAS (*Turchin et al., 2012*; *Berg and Coop, 2014*; *Robinson et al., 2015*; *Field et al., 2016*; *Berg et al., 2019b*; *Edge and Coop, 2019*; *Speidel et al., 2019*), an exciting approach that has also proven to be technically challenging (*Berg et al., 2019a*; *Sohail et al., 2019*). A better understanding of the process of polygenic adaptation should help to guide such efforts.

## Acknowledgements

We thank Guy Amster, Jeremy Berg, Nick Barton, Yuval Simons and Molly Przeworski for many helpful discussions, and Jeremy Berg, Graham Coop, Joachim Hermisson, Guillaume Martin, Will Milligan, Peter Ralph, Yuval Simons, Leo Speidel and Molly Przeworski for comments on the manuscript.

## Additional information

### Funding

| Funder | Grant reference number | Author |
|---|---|---|
| National Institutes of Health | GM115889 | Laura Katharine Hayward Guy Sella |
| National Institutes of Health | GM121372 | Laura Katharine Hayward |

The funders had no role in study design, data collection and interpretation, or the decision to submit the work for publication.

### Author contributions

Laura Katharine Hayward, Conceptualization, Formal analysis, Validation, Investigation, Methodology, Writing – original draft, Writing – review and editing; Guy Sella, Conceptualization, Supervision, Funding acquisition, Writing – original draft, Writing – review and editing

### Author ORCIDs

Laura Katharine Hayward ![ORCID] http://orcid.org/0000-0003-4445-8067
Guy Sella ![ORCID] http://orcid.org/0000-0002-5239-7930

### Decision letter and Author response

Decision letter https://doi.org/10.7554/eLife.66697.sa1
Author response https://doi.org/10.7554/eLife.66697.sa2

## Additional files

### Supplementary files

• Transparent reporting form

### Data availability

No new data was collected for this study. Data for this study were generated by computer simulations run by the authors. These simulations output summaries of several quantities of interest, as well as the standard error of these quantities. Source data files with the results of these simulations have been provided for Figures 2B–C, 4, 5, 7A and 8.

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

# Appendix 1

## Additional tables

**Appendix 1—table 1.** Summary of notation.

| Symbol | Definition |
|---|---|
| $N$ | Population size |
| $U$ | Expected number of mutations per gamete per generation affecting the trait |
| $V_S$ | Width of the Gaussian fitness function ($-1/V_S$ is the quadratic selection gradient) |
| $\delta$ | Typical magnitude of fluctuations around the optimum at equilibrium ($= \sqrt{V_S/(2N)}$) |
| $a$ | The magnitude of an allele's effect on the trait |
| $S$ | An allele's scaled selection coefficient at equilibrium ($= a^2$ in units of $\delta^2$) |
| $g(a)$ | The mutational distribution of phenotypic magnitudes |
| $\Lambda$ | The size of the shift in optimum |
| $t$ | Time after shift in optimum |
| $V_A(t)$ | The additive genetic variance at time $t$ |
| $\mu_3(t)$ | The 3rd central moment of the trait distribution at time $t$ |
| $D(t)$ | Distance of the mean phenotype from the optimum at time $t$ |
| $D_L(t)$ | Lande's approximation for $D(t)$ |
| $t_1$ | The time of the end of the rapid phase |
| $x_t(a, x_0)$ | Expected frequency of an allele with effect $a$ and initial MAF $x_0$, at time $t$ |
| $\Delta x_t^*(a, x_0)$ | Expected frequency difference between opposite alleles at time $t$ ($\equiv x_t(a, x_0) - x_t(-a, x_0)$) |
| $\Delta z_t^*(a, x_0)$ | Expected contribution to phenotypic change of a pair of opposite alleles at time $t$ |
| $v^*(a, x)$ | An allele's contribution to phenotypic variance ($= 2a^2 x(1-x)$) |
| $\Delta x_d^*(a, x_0)$ | Expected frequency difference between opposite alleles due to an instantaneous pulse of directional selection ($\equiv x_d(a, x_0) - x_d(-a, x_0)$) |
| $\Delta z_d^*(a, x_0)$ | Expected contribution to phenotypic change of a pair of opposite alleles after an instantaneous pulse of directional selection |
| $\Delta z_t(a, x_0), \Delta z_t(a)$ | Expected contribution to phenotypic change per unit mutational input of opposite alleles with magnitude $a$, or with magnitude $a$ and initial MAF $x_0$, at time $t$ |
| $v(a, x_0), v(a)$ | Equilibrium density of phenotypic variance per unit mutational input of alleles with magnitude $a$, or with magnitude $a$ and initial MAF $x_0$ |
| $\pi(a, x)$ | Fixation probability of an allele with magnitude $a$ and initial frequency $x$ under stationary stabilizing selection and genetic drift |
| $f(a)$ | Relative long-term contribution to phenotypic change |
| $A$ | Amplification of the long-term contribution to phenotypic change from standing variation |
| $C$ | Amplification of the long-term contribution to phenotypic change $= \int v(a)g(a)da / \int f(a)g(a)da - 1$ |

**Appendix 1—table 2.** Summary of assumptions on parameters.

| Assumption | Interpretation |
|---|---|
| $\sqrt{2NU} \gg 1$ | The trait is highly polygenic |
| $U = Lu \leq 0.02$ | The mutation rate per gamete is sufficiently low such that $V_A(0) \ll V_S$ |
| $a/\sqrt{V_S} \ll 1$ | Directional selection coefficients of alleles satisfy $s_d \ll 1$ |
| $g(a)$ with a substantial portion satisfying $a^2 \gtrapprox 1$ | A substantial proportion of incoming mutations are not effectively neutral |
| $g(a) = g(-a)$ | Mutational input is symmetric |
| $\Lambda > \delta$ | The shift is not negligible (i.e. larger than the typical equilibrium fluctuations in mean phenotype) |
| $\Lambda \lessapprox \sqrt{V_S}$ | Maximal directional selection coefficients satisfy $s_d \ll 1$ |
| $\Lambda/\sqrt{V_A(0)} \lessapprox 1/2 \cdot \sqrt{2NU}$ | Moving the trait mean to the new optimum requires only small average frequency changes per segregating site |

## Appendix 2

### Additional figures

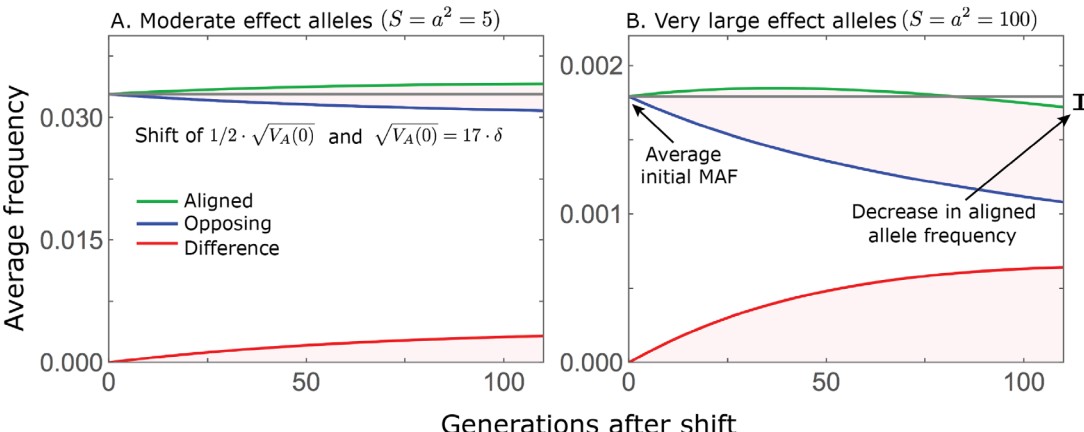

**Appendix 2—figure 1.** While directional selection during the rapid phase increases the frequency of aligned alleles relative to opposing ones, the frequency of aligned alleles does not necessarily increase. Here, we show an example of the trajectories of (**A**) moderate and (**B**) large effect alleles in response to a relatively small shift in optimum; the trajectories were calculated using *Equations A3-39*, *A3-40* and *A3-48* in *Appendix 3*. When directional selection is sufficiently weak (the shift is small), the frequency of aligned alleles with sufficiently large effects will decrease (**B**). However, the frequency of opposing alleles decreases *more*, and the frequency difference (in red) contributes to the change in mean phenotype.

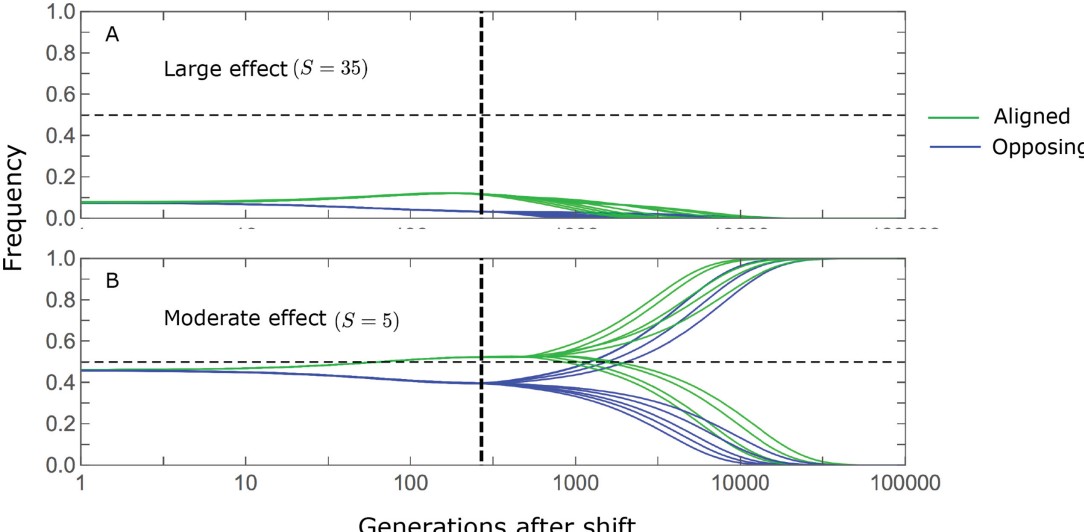

**Appendix 2—figure 2.** Stabilizing selection during the equilibration phase causes turnover in the genetic basis of adaptation. The cartoons depict the trajectories of alleles with opposing effects of given magnitudes and initial MAFs. For the purpose of illustration, we focus on alleles with large (**A**) and moderate (**B**) effects, with initial MAFs in the tail of the corresponding equilibrium MAF distribution (the 99.5th percentile), and a shift size of $\Lambda = 2 \cdot \sqrt{V_A(0)}$ with $\sqrt{V_A(0)} = 17\delta$. Directional selection during the rapid phase increases the frequency of aligned alleles relative to those with opposing effects, and these frequency differences underlie short-term phenotypic adaptation. (**A**) The initial MAF of large effect alleles, even those in the 99.5th percentile, is sufficiently low such that both aligned and opposing alleles still have low MAFs at the end of the rapid phase. Consequently, they are both strongly selected against during the equilibration phase and almost certainly go extinct, thereby erasing their short-term contribution to phenotypic adaptation. (**B**) Moderate effect alleles start at much higher initial MAFs. In the extreme, this initial frequency is sufficiently high for directional selection during the rapid phase to push aligned alleles above frequency 1/2, thereby reversing the direction of (under-dominant) selection on

*Appendix 2—figure 2 continued*

them, but not on the opposing alleles, during the equilibration phase. Consequently, the expected contribution of moderate effect alleles with sufficiently high initial MAF to phenotypic adaptation is amplified during the equilibration phase.

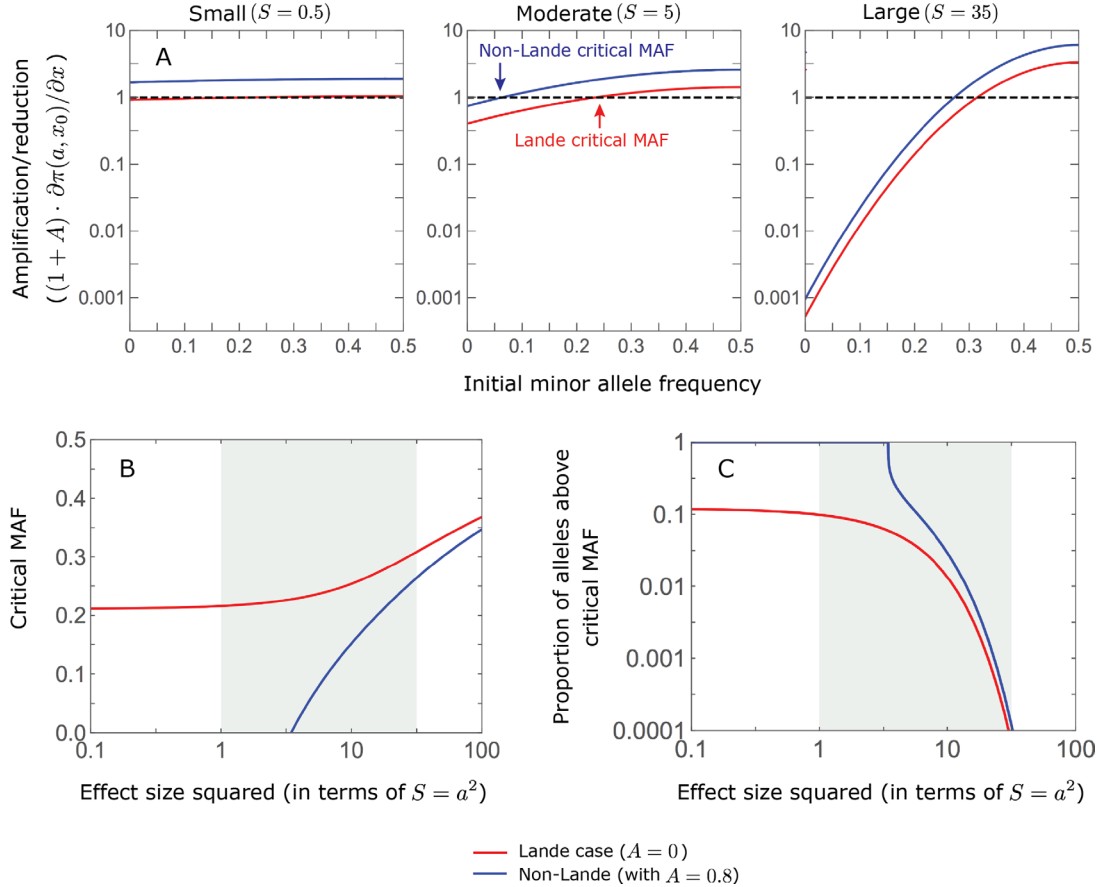

Appendix 2—figure 3. The long-term phenotypic contribution of minor alleles is amplified if they start above a critical initial frequency, which depends on their magnitude, and it is diminished if they start below this critical MAF. In the linear approximation, we found that

$$\Delta z_\infty \left(a, x_0\right) \approx (1 + A) \cdot \Delta z_\infty^L \left(a, x_0\right) = (1 + A) \cdot \partial\pi/\partial x(a, x_0) \cdot \Delta z_d^L \left(a, x_0\right)$$

(*Equation 24* in the main text). Further assuming that $\Delta z_{t_1} \left(a, x_0\right) \approx \Delta z_d^L \left(a, x_0\right)$ (which holds if the shift is not miniscule, that is, $\Lambda \gg \delta$), we find that the (multiplicative) amplification/reduction of the phenotypic contribution is approximated by $(1 + A) \cdot \partial\pi/\partial x(a, x_0)$. In this approximation, the critical MAF for alleles with magnitude satisfies

$$(1 + A) \cdot \partial\pi/\partial x(a, \boldsymbol{x_c}) = 1.$$

(**A**) The amplification/reduction for alleles with small (left), moderate (middle), and large (right) effects as a function of initial MAF, in the Lande (red) and non-Lande (blue) case. The curves and critical MAFs are calculated from the above equations. Given a factor $A > 0$, the contribution of alleles with sufficiently small effect sizes are amplified for any initial MAF $x_0$ (*Figure 6*), because $\partial\pi/\partial x(a, x_0) \approx 1$ and thus $\Delta z_\infty^L \left(a, x_0\right) \approx \Delta z_d^L \left(a, x_0\right)$ and $\Delta z_\infty^v \left(a, x_0\right) > \Delta z_{t_1} \left(a, x_0\right)$. In turn, for sufficiently large effect sizes, the curves for $\Delta z_\infty^v \left(a, x_0\right)$ and $\Delta z_d^L \left(a, x_0\right)$ ($\approx \Delta z_{t_1} \left(a, x_0\right)$) intersect (*Figure 6B*) and thus a critical MAF exists (i.e., $0 < x_c < 1/2$). These considerations explain why, for sufficiently large effect sizes, the long-term contribution of alleles with low initial MAFs is diminished relative to their short-term contribution. (**B**) The critical MAF as a function of effect size in the Lande and non-Lande case (based on $(1 + A) \cdot \partial\pi/\partial x(a, \boldsymbol{x_c}) = 1$). The critical MAF is lower in the non-Lande case, because $A > 0$. This proportion declined with increasing allele magnitude both because initial MAFs at equilibrium decrease and because the critical MAF increases (panel **B**). It is greater in the non-Lande case because the critical MAF is lower (panel **B**). (**C**) The proportion of segregating sites at equilibrium with MAF exceeding the critical MAF.

## Appendix 3

## Supplementary analyses

## Table of contents for Appendix 3

## 1. The basic equations

Our analysis relies on two basic equations:

- **The allelic equation:** describing the expected change in frequency per generation of an allele with effect size $\pm a$ and frequency $x$ (**Equation 7** in the main text):

$$E\left(\Delta x\right) \approx \left(\pm a \cdot D(t)/V_S\right) \cdot x(1-x) - \left(a^2/V_S\right) \cdot (1 - D^2(t)/V_S) \cdot x(1-x)(1/2 - x). \tag{A3-1}$$

- **The phenotypic equation:** describing the expected change in the mean distance from the optimum, $D(t)$, per generation (**Equation 3** in the main text):

$$E\left(\Delta D(t)\right) \approx -\left(V_A(t)/V_S\right) \cdot D(t) + (1 - D^2(t)/V_S) \cdot \mu_3(t)/(2V_S), \tag{A3-2}$$

where $V_A(t)$ and $\mu_3(t)$ denote the 2ⁿᵈ and 3ʳᵈ central moments of the phenotypic distribution.

Previous work relied on similar allelic and phenotypic equations. Assuming a fitness landscape with a single optimum, the expected change in allele frequency per generation has commonly been approximated by

$$E\left(\Delta x\right) \approx \left(a \cdot D/V_S\right) \cdot x(1-x) - \left(a^2/V_S\right) \cdot x(1-x)(1/2 - x) \tag{A3-3}$$

(**Barton, 1986a**; **Charlesworth, 2013**; **de Vladar and Barton, 2014**). This equation was used to derive phenotypic equations similar to ours (see below and **Barton and Turelli, 1986b**; **Bürger, 1991**). The allelic equation has been justified for a Gaussian fitness function, i.e., $W(z) = \text{Exp}\left[-z^2/(2V_S)\right]$ (our

**Equation 2** in the main text), further assuming that: (i) phenotypes are normally distributed in the population; (ii) the width of the distribution is far smaller than the width of the fitness function (i.e. $V_A \ll V_S$); and (iii) the mean distance from the optimum is very small (i.e., $D^2 \ll V_S$) (**Barton and Turelli, 1986b**; **Simons et al., 2018**). As we show in the main text, the assumption that the phenotype distribution is normally distributed can be violated in ways that affect the course of adaptation qualitatively, giving rise to the non-Lande case. Also, requiring that the mean distance from the optimum is very small can be restrictive.

We derive our basic equations without making the assumptions of a Normal phenotypic distribution or that the distance to the optimum is very small. Similar to previous work, we do not provide a rigorous treatment of the errors introduced by our approximations (see **Hayward, 2020**, for such treatment). We therefore validate our main results against simulations of the full model (Section 2.2).

## 1.1. Derivation of the allelic equation

The expected change in frequency of an allele with phenotypic effect $a$ at frequency $x$ in a single generation derives from the standard form (**Gillespie, 2004**):

$$E\left(\Delta x\right) = E\left(x'\right) - x = \frac{x^2 \cdot \bar{W}_2 + x(1-x) \cdot \bar{W}_1}{\bar{W}} - x,$$  (A3-4)

where $\bar{W}_i$ denotes the average fitness of an individual with $i = 0, 1$ or $2$ copies of the allele, where the averaging is over the distribution of contributions to the phenotype from other sites, and

$$\bar{W} \equiv x^2 \cdot \bar{W}_2 + 2x(1-x) \cdot \bar{W}_1 + (1-x)^2 \cdot \bar{W}_0$$  (A3-5)

is the average fitness in the population. An individual with $\iota$ copies of the allele and background contribution $R$ has fitness

$$W_i(R) = W(R + i \cdot a),$$  (A3-6)

where $W$ is the Gaussian fitness function (**Equation 2** in the main text), and the average fitness of individuals with $\iota$ copies at the focal site is

$$\bar{W}_i = \int_{-\infty}^{\infty} W(R + i \cdot a) \cdot p(R) dR,$$  (A3-7)

where $p(R)$ denotes the distribution of background contributions (in continuous form), which is independent of the genotype at the focal site assuming linkage equilibrium.

We apply two successive approximations to get from **Equation A3-4** to our allelic equation. First, we approximate the background contribution by its mean. The mean background contribution is $\mu - 2ax$, where $\mu$ is the mean trait value in the population and $2ax$ is the mean contribution of the focal site. In this approximation, the mean fitness of individuals with $\iota$ copies of the allele at the focal site is

$$\bar{W}_i = W(\mu + a \cdot (i - 2x)).$$  (A3-8)

Substituting these expressions into **Equations A3-4** and **A3-5** yields

$$E\left(\Delta x\right) = E\left(x'\right) - x \approx \frac{x^2 \cdot W\left(\mu + 2a(1-x)\right) + x(1-x) \cdot W\left(\mu + a(1-2x)\right)}{\bar{W}} - x,$$  (A3-9)

with $\bar{W} = x^2 \cdot W\left(\mu + 2a(1-x)\right) + 2x(1-x) \cdot W\left(\mu + a(1-2x)\right) + (1-x)^2 \cdot W\left(\mu - 2ax\right)$. Neglecting variation in background contributions seems reasonable given our assumption that the phenotypic standard deviation is small relative to the width of the fitness function (i.e. $\sqrt{V_A} \ll \sqrt{V_S}$). We use this approximation in our *all alleles* and *single alleles* simulations (see the *Model* section in the main text and Section 2).

Second, we obtain our allelic equation (**Equation A3-1**) from the 2nd order Taylor expansion of **Equation A3-9** in $a/\sqrt{V_S}$ around $a/\sqrt{V_S} = 0$. Neglecting terms of order $\left(a/\sqrt{V_S}\right)^3$ and higher seems reasonable given our assumption that $a/\sqrt{V_S} \ll 1$. However, the 2nd order expansion of a Gaussian becomes inaccurate far from its peak. In our case, this manifests in having the $\iota^{\text{th}}$ term of the expansion include expressions of the form $\left(a/\sqrt{V_S}\right)^i \cdot \left(D/\sqrt{V_S}\right)^k$ for $k \leq i$, which may not be negligible when

$|D| \gg \sqrt{V_S}$. We therefore also require that $|D| \lesssim \sqrt{V_S}$ for our allelic equation to be accurate (see **Appendix 1—table 2**).

## 1.2. Derivation of the phenotypic equation

Next, we rely on our allelic equation (**Equation 7** in the main text and **Equation A3-1**) and on our conditions on parameters to derive our phenotypic equation. To this end, we rewrite the allelic equation in terms of the change in an allele's contribution to the mean phenotype per generation:

$$E\left(2a \cdot \Delta x\right) \approx \left(v^*(a,x)/V_S\right) \cdot D - \left(1 - D^2/V_S\right) \cdot \left(v_3^*(a,x)/(2V_S)\right), \tag{A3-10}$$

where $v^*\left(a,x\right) = 2a^2 x(1-x)$ and $v_3^*\left(a,x\right) = a^3 x(1-x)(1-2x)$ are the allele's contributions to the 2nd and 3rd central moments of the phenotypic distribution, respectively. Our phenotypic equation then follows from expressing the expected change in the mean distance to the optimum per generation as a sum of these allelic contributions:

$$
\begin{aligned}
E\left(\Delta D\right) &= -\sum_i E\left(2a_i \cdot \Delta x_i\right) \\
&\approx -\left(\sum_i v^*\left(a_i,x_i\right)/V_S\right) \cdot D + \left(1 - D^2/V_S\right) \cdot \sum_i v_3^*\left(a_i,x_i\right)/\left(2V_S\right) \\
&= -\left(V_A/V_S\right) \cdot D + \left(1 - D^2/V_S\right) \cdot \mu_3/\left(2V_S\right),
\end{aligned}
\tag{A3-11}
$$

where $V_A$ and $\mu_3$ are the 2nd and 3rd central moments of the phenotype distribution, which equal the corresponding sum over alleles under the assumption of linkage equilibrium (because the first three central moments of a sum of independent random variables equal the sum of these variables' central moments).

The terms on the right hand side of the phenotypic equation can be interpreted as follows. The first term captures the effect of directional selection driving the mean phenotype towards the optimum at a rate that is proportional to the additive genetic variance (**Lande, 1976**). The 2nd part of the second term, $\mu_3/\left(2V_S\right)$, captures the effect of stabilizing selection on an asymmetric (skewed) phenotypic distribution. Namely, when the mean is near the optimum, stabilizing selection pushes the mean phenotype in the direction opposite to the thicker tail of the phenotype distribution, because, given that fitness is quadratic near the optimum, reducing the distance of extreme phenotypes (more of which lie in the thicker tail) from the optimum increases mean fitness. The 1st part of second term, $\left(1 - D^2/V_S\right)$, reflects the fact that stabilizing selection, which tends to reduce phenotypic variance, becomes weaker when the phenotypic mean is far from the optimum (provided $|D| < \sqrt{V_S}$). (It does so because the magnitude of the second derivative of the Gaussian fitness function decreases as the distance from the optimum increases.) When $D^2 \ll V_S$, our phenotypic equation is well approximated by

$$E\left(\Delta D\right) \approx -\left(V_A/V_S\right) \cdot D + \mu_3/\left(2V_S\right), \tag{A3-12}$$

which was derived previously under the rare-alleles approximation (**Barton and Turelli, 1986b**) and assuming a parabolic fitness function (**Bürger, 1991**).

## 2. Simulations

We compared our analytic results with results from three kinds of simulations. The first kind implements the full model. The second traces *all alleles* rather than individuals. Specifically, the frequency of an allele in the next generation follows a binomial distribution $x' \sim B\left(2N, x + E\left(\Delta x\right)\right)$ with $E\left(\Delta x\right)$ approximated by **Equation A3-9**, and the number of new mutations introduced per generation is Poisson distributed with mean $2NU$. Tracing alleles rather than individuals in this way entails two potential sources of error. First, our approximation for $E\left(\Delta x\right)$ replaces the distribution of background phenotypic contributions by its mean (Section 1.1), thus neglecting the effects of variation in this distribution. Second, we assume linkage equilibrium rather than free recombination, thus neglecting the effects of short-term linkage disequilibrium (**Robertson, 1956a**).

The 3rd kind of simulations traces a *single allele* that was segregating at the time of the shift. We sample the initial MAFs from the approximate closed form, equilibrium distribution (**Equation A3-27**), using importance sampling based on the density of variance at different MAFs (**Equation A3-28**). Changes in frequency are modeled as we described for the *all alleles* simulations, but here we use

the mean distance to the optimum, $\bar{D}(t)$, which is given from the outset. In the Lande case, we take the mean distance to equal the expectation under Lande's approximation (*Equation 5* in the main text), which depends only on the initial phenotypic variance $V_A(0)$; $V_A(0)$ follows from the distribution of magnitudes and extent of polygenicity as described in *Equation A3-30*. In the non-Lande case, we estimate the mean distance by averaging over 2500 *all alleles* simulations with the same model parameters. During the equilibration phase, we average over the quasi-static approximation for the distance (*Equation 6* in the main text) rather than the distance itself, because the former is considerably less variable among simulations than the latter (Section 7.1 and *Appendix 3—figure 22A and B*). *Single allele* simulations share sources of error with the *all alleles* simulations, in addition to potential errors arising from relying on the analytic approximation for the initial MAF and on the mean rather than on the actual (fluctuating) distance $D(t)$.

We use the *all alleles* and *single allele* simulations throughout because they are considerably more computationally tractable, allowing us to obtain larger sample sizes and thus more precise comparisons with our analytic predictions. For example, running a single simulation, with the parameter values of the non-Lande case used in the main text (see section on *Simulations and resources*), takes ~340 hours with the *full model* simulations and ~10 with the *all allele* simulations (for a burn-in period of $10N$ generations before the shift and a period of $12N$ generations after). Unfortunately, precise estimates of fixation probabilities of alleles with a particular effect size may require hundreds or thousands (or even orders of magnitude more, if the effect size is extremely rare) of *full* or *all alleles* simulations. This problem is overcome by the *single allele* simulations. For example, with the same parameters, running a *single allele* simulation for 1000 aligned and 1000 opposing alleles segregating at the time of the shift, with each of 13 different effect sizes (those used in many of the figures, e.g. *Appendix 3—figure 8*) takes ~10 hours on the same computational platform.

## 2.1. Choice of simulation parameters

The simulation results in the appendix are based on a wider range of model parameter values than those shown in the main text (see section on *Simulations and resources*). Specifically:

**All simulation types:** In all simulations, we use the same population size and shifts as in the main text, i.e. a populations size of $N = 10^4$ and a shift size of $\Lambda = 2\sqrt{V_A(0)}$ or $4\sqrt{V_A(0)}$. Since we work in units of $\delta$, the typical deviation of the population mean from the optimum at equilibrium, we take $V_S = 2N$ (see section on *Choice of units*).

***All alleles* simulations:** We run 2500 simulations for each of the following combinations of parameters:

- An exponential distribution of effect sizes squared (with the trait measured in units of $\delta$), with mean $E(a^2) = E(S) = 1$ in the **Lande case**, and $E(a^2) = E(S) = 16$ in the **non-Lande case**.
- Three mutation rates (per gamete per generation), which correspond to different degrees of polygenicity:
  **Low polygenicity:** $U = 0.005$,
  **Medium polygenicity:** $U = 0.01$, and
  **High polygenicity:** $U = 0.02$.

***Single allele* simulations:** We run 250,000 simulations for each effect size used, where:

- In the **Lande case**, we assume the Lande phenotypic response, which is determined by the initial genetic variance (*Equation 5* in the main text). The variances are calculated from *Equation A3-30* assuming an exponential distribution of effect sizes squared (with $E(a^2) = 1$) and the same three mutation rates used in the all alleles simulations yielding:
  **Low polygenicity:** $\sqrt{V_A(0)} = 12 \cdot \delta$
  **Medium polygenicity:** $\sqrt{V_A(0)} = 17 \cdot \delta$, and
  **High polygenicity:** $\sqrt{V_A(0)} = 24 \cdot \delta$.
- In the **non-Lande case**, $\bar{D}(t)$ is the average over the corresponding *all alleles* simulations.

These parameter values were primarily chosen to span as wide a range as possible given our conditions on them (*Appendix 1—table 2*). The one exception is our choice of a population size of $N = 10000$, which was motivated by being on the order of the effective size in human populations (*Schiffels and Durbin, 2014*), though on the lower side in order to allow for greater computational tractability. The choice of the effect size distribution of new mutations is constrained by the requirement that a substantial proportion of them are not effectively neutral, i.e., $a^2 \gtrsim 1$ (*Appendix 1—table 2*). In simulations representative of the Lande case, we further require that most new mutations satisfy

$a^2 \lessgtr 4$ (see **Equation A3-66**). Our choice of an exponential distribution of effect size squared with $E(a^2) = 1$ satisfies both requirements, with proportion 0.63 of mutations with $a^2 \leq 1$, 0.35 with $1 < a^2 \leq 4$, and 0.02 with $a^2 > 4$ (**Appendix 3—figure 1A**). In simulations representative of the non-Lande case, we require that a substantial proportion of mutations have effect sizes $a^2 > 4$ while being much smaller than the width of the fitness function, i.e., $a/\sqrt{V_S} \ll 1$ (**Appendix 1—table 2**). Our choice of an exponential distribution of effect size squared with $E(a^2) = 16$ satisfies these requirements, with proportion 0.78 of mutations with $a^2 > 4$ and only $4 \times 10^{-6}$ with $a^2 > V_S/100$.

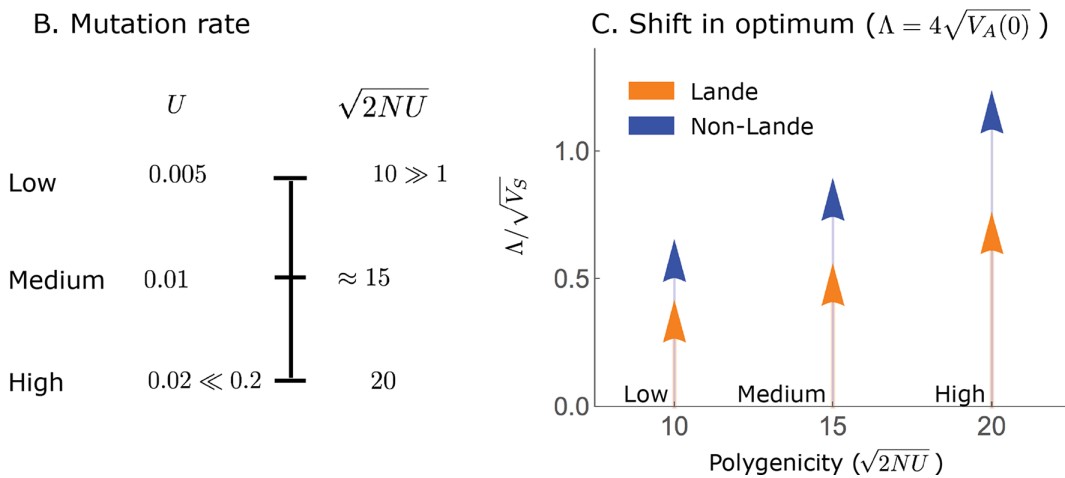

**Appendix 3—figure 1.** Our simulation parameter values are chosen to span as wide a range as possible given our conditions on them.

Given the population size, the choice of mutation rate per haploid genome per generation, $U$, is bounded from below by the assumption that the trait is highly polygenic, i.e., $\sqrt{2NU} \gg 1$ (**Appendix 1—table 2**). Consequently, the lowest mutation rate that we used was $U = 0.005$, corresponding to our low end of polygenicity, i.e., $\sqrt{2NU} = 10 \gg 1$. The choice of mutation rate is also bounded from above by the assumption that $U \leq 0.02$ (**Appendix 1—table 2**). The highest mutation rate that we used was therefore $U = 0.02$, corresponding to our high end of polygenicity, i.e., $\sqrt{2NU} = 20$. As an intermediate value, we used $U = 0.01$, corresponding to $\sqrt{2NU} \approx 15$ (**Appendix 3—figure 1B**).

Lastly, we require the shift in optimum to be smaller than or on the order of the width of the fitness function, i.e., $\Lambda \lessgtr \sqrt{V_S}$ (**Appendix 1—table 2** and **Appendix 3—figure 1C**). As we detail

in Section 3.2, this requirement constrains how large the shift can be relative to the phenotypic standard deviation, $\sqrt{V_A(0)}$, in a manner that depends on the mutation rate and distribution of allele magnitudes: $\Lambda/\sqrt{V_A(0)} \leq (1/\sqrt{U})/\sqrt{\int_0^\infty v(a)\,g(a)da}$ (*Equation A3-31* and *Appendix 3—figure 5*). As our largest shift, we chose $\Lambda = 4\sqrt{V_A(0)}$, which, for our highest mutation rate ($U = 0.02$), is near the upper bound given our choices of distribution of allele magnitudes for the Lande and non-Lande cases (with $\Lambda/\sqrt{V_S} = 0.7$ and 1.2, respectively). As a smaller shift, we used $\Lambda = 2\sqrt{V_A(0)}$.

## 2.2. Validation of our main results with full model simulations

In *Appendix 3—figure 2*, we compare our main results about the phenotypic dynamics with those from the *full model* simulation. Because of the computational intensity of the *full model* simulation, we use only one set of parameters, the non-Lande case used in the main text (see section on *Simulations and resources*) with a shift of $\Lambda = 2\sqrt{V_A(0)}$, and run a sample of only 500 simulations, which is why our standard errors are sizable. The phenotypic variance before the shift is greater in the *full model* simulation than in our analytic approximation and *all alleles* simulations (*Appendix 3—figure 2A*), plausibly because only the *full model* simulation incorporates the effect of the variance of the phenotypic background of a focal site (see Section 1.1). Specifically, the variance of the phenotypic background reduces the efficacy of stabilizing selection—akin to how environmental contributions to phenotypic variance effectively increase $V_S$ (*Turelli, 1984*; *Bürger, 2000*)—which increases alleles' initial MAFs and contributions to phenotypic variance (a similar slight underestimation of [equilibrium] genetic variance is seen in Fig. A6(a) of *Simons et al., 2018*).

The same effect plausibly accounts for most of the other small phenotypic differences between the *full model* and *all alleles* simulations. The short-term term phenotypic response is slightly faster in the *full* than in the *all alleles* simulations (*Appendix 3—figure 2E*), because of the greater initial phenotypic variance in the full model. The increase in the 3rd phenotypic moment is smaller in the *full* than in the *all alleles* simulations (*Appendix 3—figure 2B*), possibly because the greater initial phenotypic variance allows the same shift in mean phenotype to be achieved by smaller relative changes in allele frequency (e.g. *Equation 10* in the main text). Lastly, the greater variance and smaller 3rd central moment in the *full model* simulations plausibly explain the smaller distance from the optimum during the equilibration phase in the *full model* relative to the *all alleles* simulations, seen both directly and in the quasi-static approximation (*Appendix 3—figure 2D and C*). Small differences notwithstanding, the results of the *full model* simulations support all of our main results about the phenotypic dynamic. The same is true of our main analytic results about the allele dynamics (*Appendix 3—figure 3*).

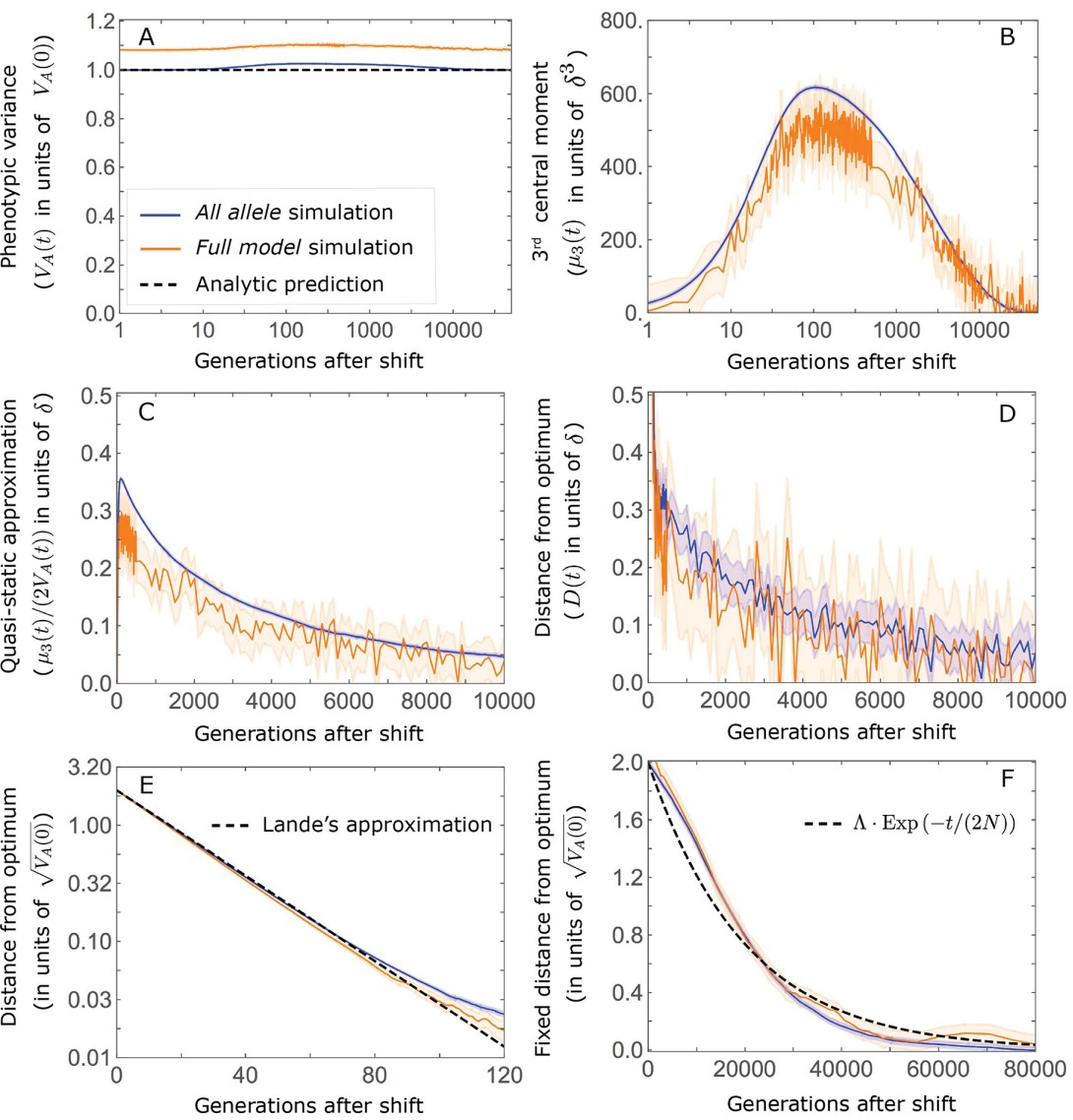

**Appendix 3—figure 2.** Comparison of our main results about the phenotypic dynamics with corresponding results from the *full model* simulations. The results shown are based on 2500 *all alleles* and 500 *full model* simulations with the parameters of the non-Lande case used in the main text (see section on *Simulations and resources*), i.e. $N = 10^4$, $U = 0.01$, an exponential distribution of effect sizes squared with $E(a^2) = 16$, which together imply $\sqrt{V_A(0)} = 29 \cdot \delta$, and a shift size of $\Lambda = 2 \cdot \sqrt{V_A(0)}$. (**A**) Phenotypic variance. (**B**) 3rd phenotypic moment. (**C**) The quasi-static approximation for the mean phenotypic distance from optimum ($\mu_3(t)/(2V_A(t))$; *Equation A3-6* in the main text). (**D**) The mean phenotypic distance from optimum. (**E**) The mean phenotypic distance from optimum shortly after the shift (during the rapid phenotypic response). (**F**) The fixed distance from the optimum (defined in the section on *Other properties of the equilibration process*).

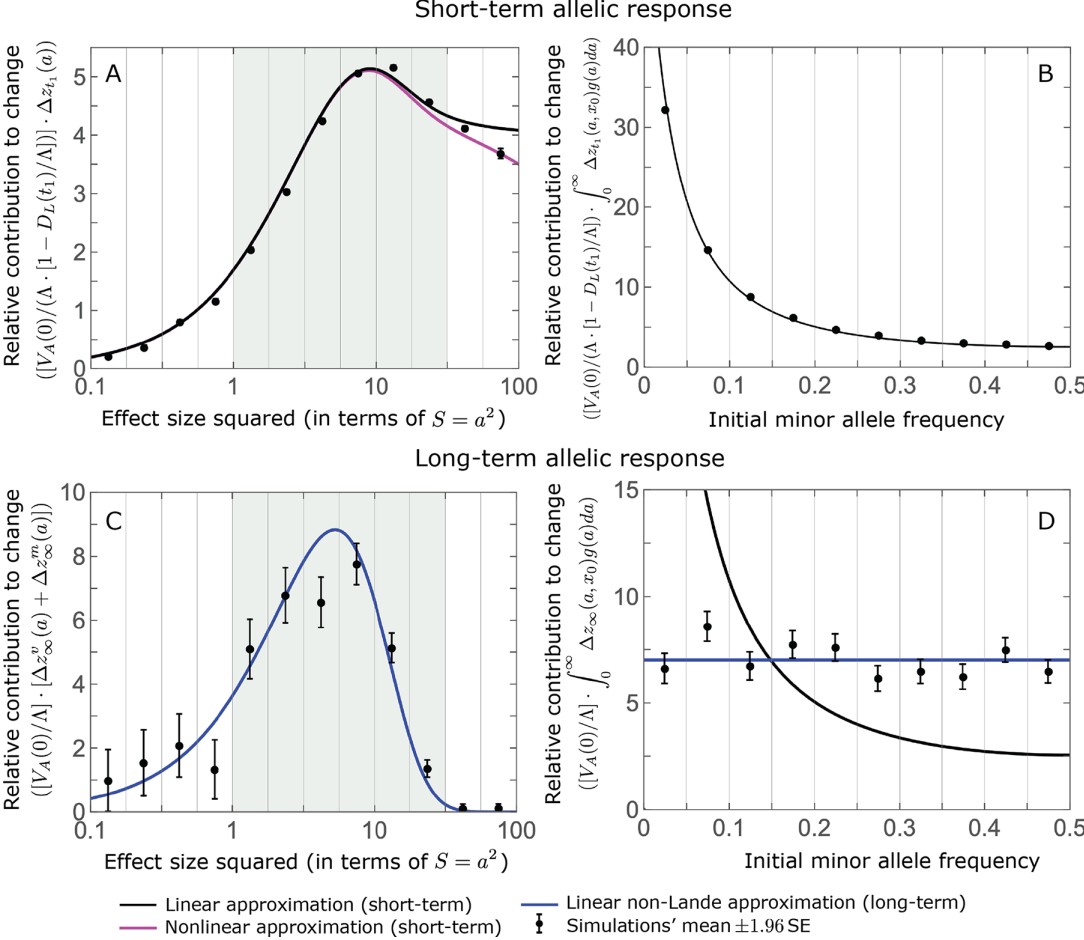

**Appendix 3—figure 3.** Comparison of our main results about the allele dynamics with the corresponding results of the *full model* simulations. The results shown are based on 500 *full model* simulations with the parameters of the non-Lande case used in the main text (see section on *Simulations and resources*). We calculate the relative contribution from each effect size bin (between the gray gridlines) by dividing the contribution of all alleles in the bin by the mutation rate per generation corresponding to that bin. We show the relative phenotypic contribution per unit mutational input of alleles as a function of effect size squared (**A** and **C**) and initial MAF (**B** and **D**) in the rapid (top) and equilibration (bottom) phases.

## 3. Allele dynamics under stabilizing selection

Understanding how stabilizing selection on the trait affects allele dynamics is crucial to characterizing the allelic response after a shift in the optimal trait value. Notably, the equilibrium allele dynamic under stabilizing selection shapes the genetic architecture of the trait (i.e., the joint distribution of phenotypic effect sizes and frequencies of alleles affecting the trait) prior to the shift and this genetic architecture shapes the short term allelic and phenotypic response to the shift. Also, after the rapid, initial response, when the mean phenotype is close to the new optimum but the genetic architecture is out of equilibrium, the longer-term allele dynamics are largely determined by the effects of stabilizing selection. Here we therefore describe the allele dynamics under stabilizing selection, both at and out of equilibrium.

The analysis of the effects of stabilizing selection is greatly aided by it being stationary. In particular, the first two moments of change in the frequency of an allele with effect size $\pm a$ and frequency $x$ in a single generation do not depend on time:

$$
\begin{aligned}
E\left(\Delta x\right) &\approx -a^2/V_S \cdot x(1-x)(1/2-x) \\
V(\Delta x) &\approx x(1-x)/(2N).
\end{aligned}
\tag{A3-13}
$$

Consequently, we can use the diffusion approximation to calculate the sojourn time: the density of time a derived allele spends at any given frequency $x'$ until fixation or loss, having been introduced at some initial frequency $p$ (**Ewens, 2004**, page 141). Namely,

$$\tau(a, x'|p) = \begin{cases} 2Nh(a,x')\left[h_+\left(a,p\right)/h_+\left(a,x'\right)\right] & 0 \le x' \le p \\ 2Nh(a,x')\left[h_-\left(a,p\right)/h_-\left(a,x'\right)\right] & p < x' \le 1 \end{cases}, \tag{A3-14}$$

where

$$\begin{aligned} h(a,z) &\equiv \left(\sqrt{\pi}/2a\right) \cdot \mathrm{Exp}\left[a^2/4 \cdot \left(1-2z\right)^2\right] / \left(\mathrm{Erf}\left[a/2\right] \cdot z(1-z)\right) \cdot h_-(a,z)h_+(a,z), \\ h_+(a,z) &\equiv \mathrm{Erf}\left[a/2\right] + \mathrm{Erf}\left[a/2 \cdot \left(1-2z\right)\right], \\ h_-(a,z) &\equiv \mathrm{Erf}\left[a/2\right] - \mathrm{Erf}\left[a/2 \cdot \left(1-2z\right)\right] \end{aligned} \tag{A3-15}$$

and Erf is the Gaussian error function. We can also use the diffusion approximation to calculate the probability that an allele fixes, $\pi$, or goes extinct, $1 - \pi$, and the sojourn time conditional on these outcomes. Namely,

$$\pi(a,x) = h_-(a,x)/\left(2\mathrm{Erf}\left[a/2\right]\right) \quad \text{and} \quad 1 - \pi(a,x) = h_+(a,x)/\left(2\mathrm{Erf}\left[a/2\right]\right). \tag{A3-16}$$

In our analysis of allele dynamics during the equilibration phase, we rely on the derivative of the fixation probability with respect to the initial frequency (**Equation 18** in the main text), which is

$$\frac{\partial \pi(a,x)}{\partial x} = \frac{2f(a)}{v(a,x)} \quad \text{where} \quad f(a) \equiv 2a^3 \cdot \frac{\mathrm{Exp}\left[-a^2/4\right]}{\sqrt{\pi} \cdot \mathrm{Erf}\left[a/2\right]}, \tag{A3-17}$$

and $v(a,x) \equiv 4a^2\mathrm{Exp}\left[-a^2x(1-x)\right]$, which is the density of variance per unit mutational input at equilibrium from alleles with magnitude $a$ and MAF $x$ (Section 3.2).

The sojourn time takes a much simpler form when expressed in terms of the minor rather than derived allele frequency, because the strength of stabilizing selection depends on the former. This *folded sojourn time* with initial frequency $p = 1/(2N)$, corresponding to new mutations, is defined

$$\tau_M(a,x) \equiv \tau\left(a, x|1/(2N)\right) + \tau(a, 1 - x|1/(2N)). \tag{A3-18}$$

Substituting the expressions for the standard sojourn time, $\tau$, and noting that $h_\pm(a,z) = h_\mp(a, 1 - z)$ and that $h(a,z) = h(a, 1 - z)$, we find that

$$\tau_M(a,x) = \begin{cases} \left(2N \cdot L(a,x)\right) \cdot 2 \cdot \mathrm{Exp}\left[-v^*(a,x)/2\right] /[x(1-x)] & 0 \le x \le 1/(2N) \\ \left(2N \cdot L\left(a, 1/(2N)\right)\right) \cdot 2 \cdot \mathrm{Exp}\left[-v^*(a,x)/2\right] /[x(1-x)] & 1/(2N) < x \le 1/2 \end{cases}, \tag{A3-19}$$

where

$$L(a,z) \equiv \frac{\sqrt{\pi}e^{\frac{a^2}{4}}}{2a}\left(\mathrm{Erf}\left[\frac{a}{2}\right] - \mathrm{Erf}\left[\frac{a}{2}\left(1-2z\right)\right]\right) \tag{A3-20}$$

and $v^*\left(a,x\right) \equiv 2a^2x(1-x)$ is the genetic variance contributed by an allele with phenotypic effect $a$ and frequency $x$. In both cases where $L(a,z)$ is used in **Equation A3-19** its 2nd argument $z \le 1/(2N)$; moreover, our condition on parameters that $1 \gg a/\sqrt{V_S} = a/\sqrt{2N}$ (**Appendix 1—table 2**) implies that $a^2z \le a^2/(2N) \ll 1$ for $z \le 1/(2N)$. Rewriting $L(a,z)$ in terms of its first-order Taylor expansion around $z = 0$ and the corresponding Lagrange error term:

$$L(a,z) = z\left(1 + a^2ze^{a^2z^*(1-z^*)}\left(\frac{1}{2} - z^*\right)\right) \quad \text{for some } 0 \le z^* \le z, \tag{A3-21}$$

where for $a^2z \ll 1$ this implies that $L(a,z) \approx z$. Substituting the approximation for $L(a,z)$ into **Equation A3-19** we find that the folded sojourn time is well approximated by

$$\tau_M(a,x) \approx \begin{cases} (2Nx) \cdot 2 \cdot \mathrm{Exp}\left[-v^*(a,x)/2\right] /[x(1-x)] & 0 \le x \le 1/(2N) \\ 2 \cdot \mathrm{Exp}\left[-v^*(a,x)/2\right] /[x(1-x)] & 1/(2N) < x \le 1/2 \end{cases}. \tag{A3-22}$$

This equation shows that the expected time a segregating site spends with a given MAF declines exponentially with the site's contribution to variance, $v^*(a,x)$ (relative to the neutral expectation); this result accords with the intuition that stabilizing selection acts to reduce phenotypic variance (**Sella and Barton, 2019**).

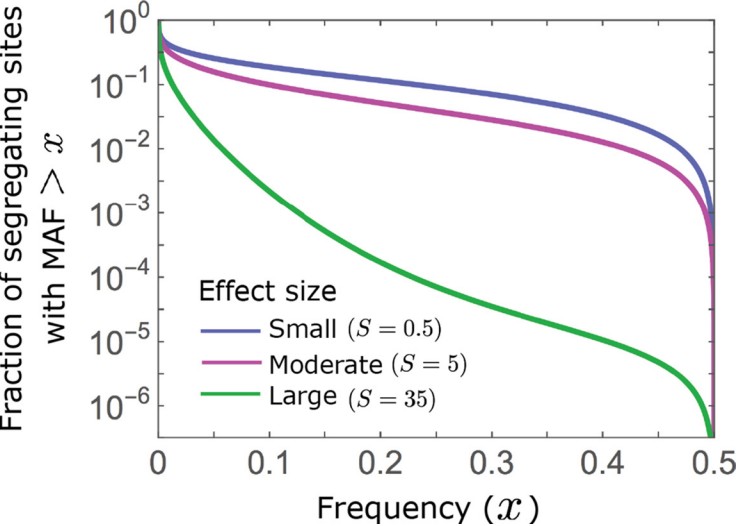

**Appendix 3—figure 4.** Allelic densities at equilibrium. We show the proportion of segregating sites with MAF > $x$ as a function of $x$. Minor alleles segregate at much lower frequencies at sites with larger effects than at sites with intermediate and small effects (see **Equations A3-22** and **A3-27**).

## 3.1. Summaries of architecture

Assuming that mutation is symmetric (i.e. with equal rates of positive and negative effects of any given magnitude), we can use the folded sojourn time to calculate summaries of the genetic architecture at equilibrium. Notably, the density of sites segregating with effects of size $\pm a$ and MAF $x$ per unit mutational input is

$$\rho(a,x) = \tau_M(a,x)/2, \tag{A3-23}$$

and the number of segregating sites is $2NU \cdot g(a) \cdot 2\rho(a,x)$, where $2NU \cdot g(a)$ is the expected number of new mutations per generation with magnitude $a$. The exponential decline of $\rho(a,x)$ with $v^*(a,x)$ implies that at equilibrium, alleles rarely segregate at MAFs much greater than $1/a^2$.

We can calculate the expectations of most summaries of interest using the density $\rho(a,x)$. Consider a summary $K$ that depends only on segregating sites, and to which a segregating site whose minor allele has effect size $\pm a$ and frequency $x$ contributes $k^*(\pm a,x)$. The density of contribution of such sites per unit mutational input is $k^*(\pm a,x) \cdot \rho(a,x)$, and the corresponding density for sites with magnitude $a$ is

$$k(a,x) = \left(k^*(a,x) + k^*(-a,x)\right) \cdot \rho(a,x). \tag{A3-24}$$

The marginal density for all sites with this magnitude (integrating over MAFs) is

$$k(a) = \int_0^{1/2} k(a,x)dx, \tag{A3-25}$$

and the expected value of the summary, weighted by the input of mutations with any given magnitude, $2NU \cdot g(a)$, is

$$K = 2NU \cdot \int k(a) \, g(a) \, da. \tag{A3-26}$$

In some cases, we are interested in the expected contribution to a summary *per segregating site* rather than *per unit mutational input*. To that end, we can replace $\rho(a, x)$ in *Equations A3-24* and *A3-25* by the probability density of segregating sites with magnitude $a$ having MAF $x$ (which is also the MAF distribution at equilibrium):

$$\rho_a\left(x\right) = \tau_M(a, x)\Big/ \int_0^{1/2} \tau_M(a, y)dy. \tag{A3-27}$$

The denominator of this expression is the expected time that a mutation with magnitude $a$ segregates before fixation or loss.

## 3.2. Phenotypic variance at equilibrium

One summary of genetic architecture that is central to our analysis is the distribution of phenotypic variance among alleles at equilibrium. An allele with effect size $\pm a$ and MAF $x$ contributes $v^*(a, x) = 2a^2 x(1 - x)$ to phenotypic variance. The density of variance per unit mutational input arising from minor alleles with magnitude and frequency is therefore

$$
\begin{aligned}
v\left(a\right) \quad &= v^*(a, x) \cdot 2\rho(a, x) \\
&\approx \begin{cases} (2Nx)4a^2 \cdot \mathrm{Exp}[-v^*(a, x)/2] & 0 \le x \le 1/(2N) \\ 4a^2 \cdot \mathrm{Exp}[-v^*(a, x)/2] & 1/(2N) \le x \le 1/2 \end{cases},
\end{aligned}
\tag{A3-28}
$$

and the marginal density of alleles with a given magnitude is

$$v\left(a\right) \equiv \int_0^{1/2} v\left(a, x\right) dx = 4a \cdot D_+\left(a/2\right), \tag{A3-29}$$

where $D_+$ is the Dawson function, $D_+\left(y\right) \equiv \frac{\sqrt{\pi}}{2} \cdot \mathrm{Exp}\left[y^2\right] \cdot \int_0^y \mathrm{Exp}\left[-u^2\right] du$.

These expressions clarify how the contributions of alleles to variance depend on their magnitudes (*Simons et al., 2018* and *Figure 4A* and *Appendix 3—figure 5*). Specifically, we see that the bulk of variance from sites with large effect sizes ($S = a^2 \gg 1$) arises from rare minor alleles ($x \lesssim 1/a^2$), whereas the variance from sites with small effect sizes ($S = a^2 \ll 1$) is approximately uniformly distributed across MAFs (since the exponent $v^*\left(a, x\right) \ll 1$) (*Figure 6B*). Other properties of $v(a)$ are detailed in the main text, as they shape the short-term allelic response to selection (see section on *The allelic response in the rapid phase*, Section 4, and *Figure 4A* and *Appendix 3—figure 5*).

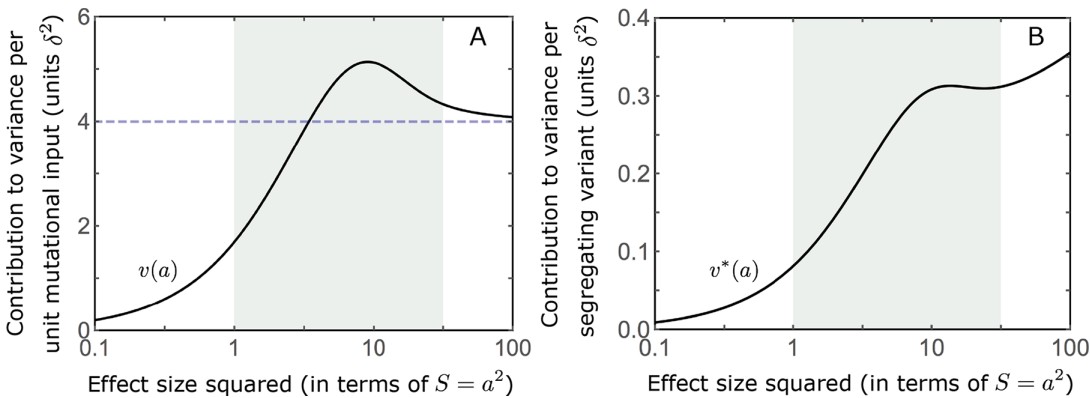

**Appendix 3—figure 5.** The distribution of variance at equilibrium. We show the contribution to variance per unit mutational input (**A**) (based on *Equation A3-29*) and per segregating variant (**B**) as a function of effect size squared.

The properties of $v(a)$ also allow us to translate our conditions on the shift size and phenotypic standard deviation (*Appendix 1—table 2*) into conditions on basic parameters. Notably, the standard deviation of the phenotypic distribution at equilibrium can be written as

$$\sqrt{V_A(0)} = \sqrt{2NU} \sqrt{\int_0^\infty v\left(a\right) g(a)da} \cdot \delta, \tag{A3-30}$$

where $\delta = \sqrt{V_S/(2N)}$ is the standard deviation of the distance between the mean and optimal phenotype at equilibrium (due to stochastic fluctuations). While we generally work in units of $\delta$, we have written $\delta$ explicitly here to emphasize the relationship between $\sqrt{V_A(0)}$ and $\delta$. Specifically, given our conditions that the trait is highly polygenic, that is, that $\sqrt{2NU} \gg 1$, and that a substantial portion of mutations are not effectively neutral, with effect sizes such that $v(a) \gtrsim 1$ (**Appendix 3—figure 5**), **Equation A3-30** implies that our assumption that $\sqrt{V_A(0)} \gg \delta$ holds. In addition, given that $v(a) \lesssim 5$ for any magnitude $a$, **Equation A3-30** also implies that $V_A(0)/V_S = U \cdot \int_0^\infty v(a)\, g(a) da \lesssim 5 \cdot U$, which alongside our condition that $U \leq 0.02$, (**Appendix 1—table 2**) implies that $V_A(0) \ll V_S$; this is why we require $U \leq 0.02$.

Given our condition that the shift size $\Lambda \lesssim \sqrt{V_S}$ (i.e. the shift does not drastically reduce mean fitness) and that $V_S = 2N$ in units of $\delta$, **Equation A3-30** also implies that

$$\frac{\Lambda}{\sqrt{V_A(0)}} \lesssim \frac{1}{\sqrt{U}} \cdot \frac{1}{\sqrt{\int_0^\infty v(a)\, g(a) da}}. \tag{A3-31}$$

This constraint on shift size is the strongest (i.e. the right-hand side is smallest) when most effect sizes are moderate and large. In this case, $v(a) \approx 4$ (**Equation A3-29** and **Figure 4A** and **Appendix 3—figure 5**) implying that $\Lambda/\sqrt{V_A(0)} \lesssim 1/(2\sqrt{U})$, i.e., that the mutation rate per gamete per generation, $U$, constrains how large a shift can be relative to the phenotypic standard deviation (**Appendix 3—figure 6**). If the mutation rate is sufficiently low, however, specifically if $U < 1/\sqrt{2N}$ (or, equivalently, $1/\sqrt{U} > \sqrt{2NU}$), **Equation A3-31** imposes only a weak constraint, allowing the shift size to be massive relative to the phenotypic standard deviation. In this case, a stronger constraint on the shift size is imposed by our condition that $\Lambda/\sqrt{V_A(0)} \lesssim 1/2 \cdot \sqrt{2NU}$ (**Appendix 1—table 2**; while similar in form, this condition originates from our analysis of the allele dynamic; see below)

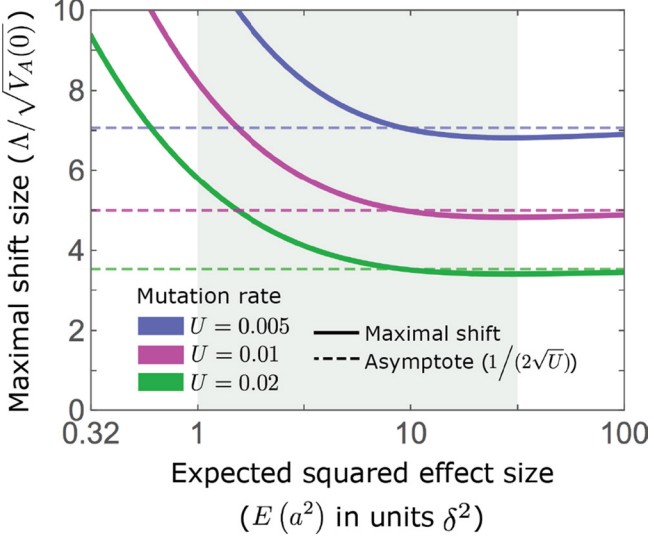

**Appendix 3—figure 6.** An upper bound on shift size measured in phenotypic standard deviations. As detailed in the text, our condition that $\Lambda \lesssim \sqrt{V_S}$ sets an upper bound on $\Lambda/\sqrt{V_A(0)}$, which in turn depends on the mutation rate $U$ and the distribution of effect sizes (**Equation A3-31**). As an illustration, we assume an exponential distribution of effect sizes squared, with the expectation $E(S) = E(a^2)$ shown on the x-axis; the x-axis starts at 0.32 because of our condition that a substantial portion of mutations are not effectively neutral (see **Appendix 1—table 2**).

Lastly, the properties of $v(a)$ allow us to derive bounds on the effect of directional selection on allele frequencies during the rapid phase. Specifically, the total change in allele frequency due to directional selection during the rapid phase is approximately proportional to $\int_0^{t_1} (D_L(t)/V_S)\, dt \approx \Lambda/V_A(0)$ (**Equation 10** in the main text), whereas $v(a) \lesssim 4$, **Equation A3-31** and our condition that the shift is not massive, that is, $\Lambda/\sqrt{V_A(0)} \lesssim 1/2 \cdot \sqrt{2NU}$ (**Appendix 1—table 2**), imply that

$$\begin{aligned}
\frac{\Lambda}{V_A(0)} &= \frac{\Lambda}{\sqrt{V_A(0)}} \cdot \frac{1}{\sqrt{V_A(0)}} = \frac{\Lambda}{\sqrt{V_A(0)}} \cdot \frac{1}{\sqrt{2NU}\,\sqrt{\int_0^\infty v(a)\, g(a) da}} \\
&\lesssim \frac{1}{2\sqrt{\int_0^\infty v(a)\, g(a) da}} \lesssim \frac{1}{2}.
\end{aligned} \tag{A3-32}$$

As expected, a smaller shift size ($\Lambda/\sqrt{V_A(0)}$) and greater extent of polygenicity ($\sqrt{2NU}$) reduce the integral effect of directional selection on allele frequencies. Additionally, our conditions on parameters impose a bound on this effect.

### 3.3. 3rd phenotypic moment

The 3rd central moment of the phenotypic distribution also plays a central role in the response to selection. Notably, when alleles with large effects contribute markedly to genetic variance, the 3rd central moment increases substantially from zero shortly after the shift in optimum, and the long-term phenotypic response takes what we refer to as the "non-Lande" form (see the *Phenotypic response* section and *Figure 2C*). Here, we propose an explanation for why alleles with large effects lead to a substantial increase in the 3rd central moment whereas alleles with small effects do not.

First consider the distribution of the allelic contributions to the 3rd central moment at equilibrium. An allele with effect size $\pm a$ and MAF $x$ contributes $v_3^*(\pm a, x) = \pm 2a^3 x(1-x)(1-2x)$. At equilibrium, the contributions of alleles with opposing effects cancel out, because for each allele magnitude and MAF $v_3^*(a, x) = -v_3^*(-a, x)$ and therefore the density per unit mutational input $v_3(a, x) = (v_3^*(a, x) + v_3^*(a, x)) \cdot \rho(a, x) = 0$. To learn about the genetic architecture of the 3rd central moment at equilibrium, we therefore consider the contributions of alleles with positive and negative effect sizes separately. The density of the 3rd central moment per unit mutational input of alleles with positive (+) or negative (–) effect sizes is

$$
\begin{aligned}
v_3^{\pm}(a, x) &= v_3^*(\pm a, x) \cdot \rho(a, x) \\
&\approx \pm \begin{cases} (2Nx) \cdot 4a^2 \cdot \text{Exp}[-v^*(a,x)/2] \cdot (1/2 - x) & 0 \leq x \leq 1/(2N) \\ 4a^2 \cdot \text{Exp}[-v^*(a,x)/2] & 1/(2N) \leq x \leq 1/2 \end{cases}
\end{aligned}
\tag{A3-33}
$$

and the corresponding marginal density for alleles with a given magnitude is

$$
v_3^{\pm}(a) = \int_0^{1/2} v_3^{\pm}(a, x)\, dx = \pm 2a \cdot \left(1 - \text{Exp}\left[-a^2/4\right]\right)
\tag{A3-34}
$$

(*Appendix 3—figure 7*). Alleles with smaller effects ($S = a^2 \ll 4$) therefore contribute negligibly to the 3rd central moment (because $\text{Exp}\left[-a^2/4\right] \approx 1$), whereas alleles with large effects ($S = a^2 \gg 4$) can have substantial contributions, which increase linearly with their magnitude (because $\text{Exp}\left[-a^2/4\right] \ll 1$).

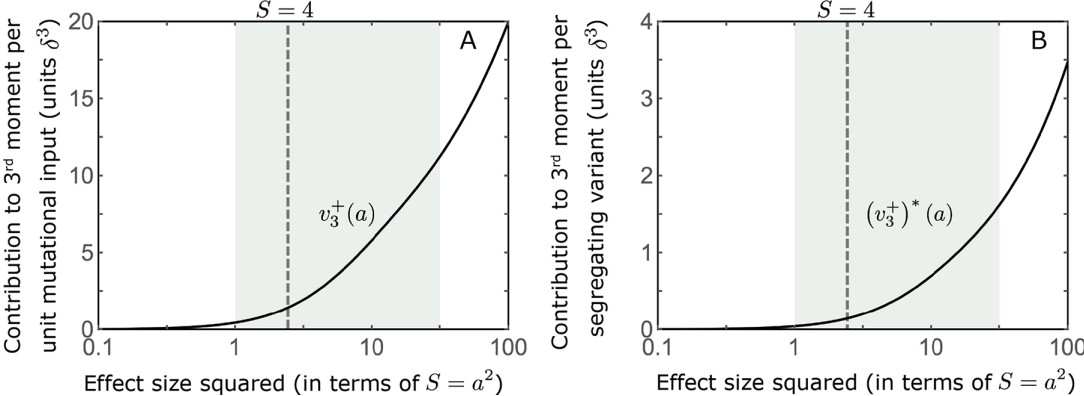

**Appendix 3—figure 7.** The distribution of the 3rd central moment at equilibrium. We show the positive/negative contribution to the 3rd central moment per unit mutational input (**A**) (based on *Equation A3-34*) and per segregating variant (**B**), as a function of effect size squared. The contributions become substantial for large effect sizes, i.e., when $S = a^2 \gg 4$.

After the shift, the frequency increase of aligned alleles relative to opposing ones introduces a non-zero 3rd central moment (*Figure 2C*). Large effect alleles contribute substantially more to this 3rd central moment, plausibly because their 'hidden' equilibrium contributions to the 3rd central moment are substantially greater and because they exhibit large changes in frequency relative to their frequency before the shift.

## 4. Allele dynamics during the rapid phase

Immediately after the shift in optimum, the population exhibits a rapid phenotypic and allelic adaptive response. Phenotypically, the population mean rapidly approaches the new optimum, as described by Lande's approximation (*Equation 5* in the main text). In the *Allele dynamics* section, we define the end of the rapid phase as the time at which the Lande's approximation for the distance $D_L(t_1)$ first equals $\delta = \sqrt{V_s/(2N)}$ (*Equation 9* in the main text and *Appendix 3—figure 25*); while somewhat arbitrary, the precise definition of this time is not fundamental for our analysis. The rapid phenotypic response arises from rapid changes to allele frequencies. Here we derive a *linear* and *non-linear* approximations for these frequency changes and the corresponding contributions to phenotypic change; the *linear* approximation was also described (in less detail) in the main text.

### 4.1. Approximate solutions for allele trajectories

Allele trajectories during the rapid phase are nearly deterministic, because it is too short for genetic drift to have a substantial effect. Further, deviations from Lande's approximation for the distance of the mean phenotype have negligible effects (see *Appendix 3—figure 8D–F*). We can therefore approximate these trajectories based on our expression for the first moment of change in allele frequency (*Equation A3-1*) and Lande's approximation for the mean phenotype (*Equation 5* in the main text). Rewriting this expression in continuous time and in the standard semi-dominant form, we find that

$$\frac{dx}{dt} = \frac{s(t)}{2}x(1-x),\tag{A3-35}$$

with selection coefficient

$$s(t) = \pm 2(a \cdot D_L(t)/V_S) - a^2/V_S \cdot (1 - D_L^2(t)/V_S) \cdot (1 - 2x).\tag{A3-36}$$

The first term in the selection coefficient reflects directional selection and the second term reflects stabilizing selection. The selection coefficient varies with time, because the strengths of directional and stabilizing selection depend on the changing distance to the new optimum, $D(t)$. It also depends on frequency, because stabilizing selection is stronger when the MAF is lower.

We can derive an implicit solution for the ordinary differential equation (ODE) for allele trajectories. Rewriting *Equation A3-35* as $1/[x(1-x)] \cdot dx = s(t)/2 \cdot dt$ and integrating both sides, we find that

$$\Delta x_t = x_t - x_0 = x_0(1-x_0)\left(\frac{\mathrm{Exp}\left[\int_0^t \frac{s(\tau)}{2}d\tau\right] - 1}{1 + x_0\left(\mathrm{Exp}\left[\int_0^t \frac{s(\tau)}{2}d\tau\right] - 1\right)}\right).\tag{A3-37}$$

In the standard semi-dominant case, with a constant selection coefficient, this expression with $\int_0^t s(\tau)d\tau = s \cdot t$ yields the standard explicit solution for allele trajectories. In our case, the selection coefficient and thus $\int_0^t s(\tau)d\tau$ depends on $x_t$ making *Equation A3-37* an implicit solution. Substituting our expression for the selection coefficient into this integral, we can express it as

$$\int_0^t \frac{s(\tau)}{2}d\tau = \left(\bar{s}_D(\pm a, t) + \bar{s}_S(a, x_0, t)\right) \cdot t,\tag{A3-38}$$

where $\bar{s}_D$ and $\bar{s}_S$ are the time-averaged directional and stabilizing selection coefficients respectively, and averages are taken from the time of the shift, $t = 0$, to time $0 \leq t \leq t_1$, that is,

$$\bar{s}_D\left(a,t\right) = \frac{a \cdot \overline{D}_L(t)}{V_S} \text{ with } \overline{D}_L(t) = \Lambda/V_A(0) \cdot \left(1 - D_L(t)/\Lambda\right) \cdot V_S/t \text{ and}$$
$$\bar{s}_S\left(a,x_0,t\right) = -\frac{a^2}{V_S} \cdot \overline{(1/2-x)(1-D_L^2/V_S)}. \tag{A3-39}$$

In developing our two explicit approximations, we rewrite the implicit solution as

$$\Delta x_t\left(\pm a,\ x_0\right) = x_t - x_0 = \pm a x_0 \left(1 - x_0\right) \cdot F_t(\pm a, x_0), \tag{A3-40}$$

where

$$F_t(a,x_0) \equiv \frac{1}{a} \cdot \left( \frac{\text{Exp}\left[\left(\bar{s}_D\left(a,t\right) + \bar{s}_S\left(a,x_0,t\right)\right) \cdot t\right] - 1}{1 + x_0 \cdot \left(\text{Exp}\left[\left(\bar{s}_D\left(a,t\right) + \bar{s}_S\left(a,x_0,t\right)\right) \cdot t\right] - 1\right)} \right). \tag{A3-41}$$

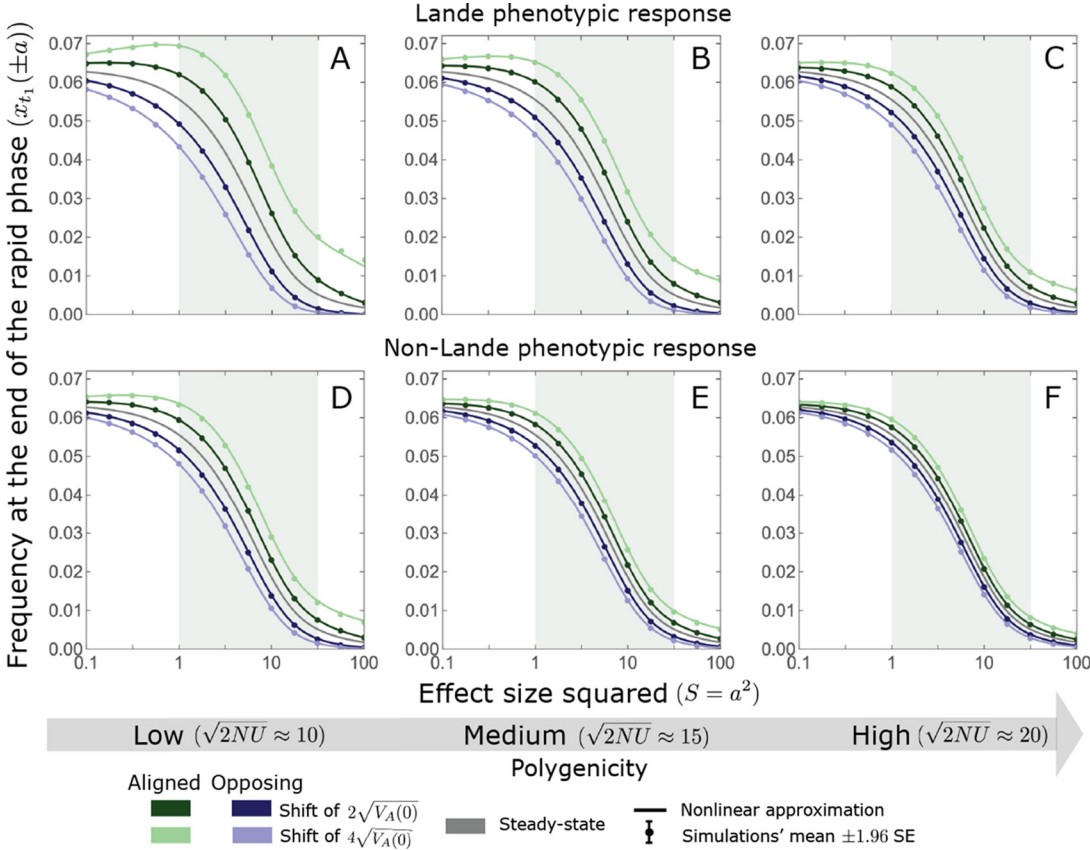

**Appendix 3—figure 8.** The nonlinear approximation accurately predicts average allele frequencies at the end of the rapid phase for a wide range of model parameters. The simulation results were generated using the *single allele* simulation, as described in Section 2.1; errors are not visible because they are smaller than the points. The analytic predictions were based on the nonlinear approximations (*Equations A3-40* and *A3-48*) averaged over the initial MAF distribution (*Equation A3-27*) with the corresponding parameters. As expected, when polygenicity is lower and shifts are larger the expected changes to allele frequencies are greater (compare **A** with **C** and **D** with **F**). Perhaps less obvious, despite the fact that our nonlinear approximation relies on Lande's approximation, it performs well even in the cases with non-Lande phenotypic dynamics (**D–F**).

The utility of this form, and specifically of the definition of $F_t$, will become apparent below, when we consider the *linear approximation* for the contribution of matched pairs of alleles to phenotypic change (*Equation A3-54* in Section 4.2).

### 4.1.1 The linear approximation

We derive the linear approximation for $F_t$ by assuming that changes to allele frequencies during the rapid phase are sufficiently small in two ways. First, they have negligible effect on the strength of stabilizing selection, such that

$$\bar{s}_S\left(a, x_0, t\right) = -\tfrac{a^2}{V_S} \cdot \overline{(1/2 - x.(a, x_0))(1 - D_L^2/V_S)} \approx \bar{s}_S^l(a, x_0, t),$$ (A3-42)

where the superscript $l$ denotes the linear approximation and

$$\bar{s}_S^l\left(a, x_0, t\right) \equiv -\tfrac{a^2}{V_S} \cdot \left(1 - \overline{D_L^2}(t)/V_S\right) \cdot (1/2 - x_0)$$ (A3-43)

with

$$\overline{D_L^2}(t)/V_S = 1/2 \cdot \tfrac{\Lambda^2}{V_A}\left(1 - D_L^2(t)/\Lambda^2\right)/t.$$ (A3-44)

Second, we assume that the integral of the effect of selection during the rapid phase is sufficiently small such that $\left(\bar{s}_D\left(a, t\right) + \bar{s}_S\left(a, x_0, t\right)\right) \cdot t \ll 1$ and therefore that $F_t$ is well approximated by the linear term in $\left(\bar{s}_D\left(a, t\right) + \bar{s}_S\left(a, x_0, t\right)\right) \cdot t$, which is why we call this the *linear approximation*. Under these assumptions

$$F_t\left(a, x_0\right) \approx \frac{\left(\bar{s}_D\left(a, t\right) + \bar{s}_S^l\left(a, x_0, t\right)\right) \cdot t}{a},$$ (A3-45)

where $\bar{s}_D$ and $\bar{s}_S^l$ are defined by *Equations A3-39* and *A3-43* respectively. This linear approximation captures the qualitative features of the allele dynamics during the rapid phase (*Figure 4A* and *Appendix 3—figures 9–11*).

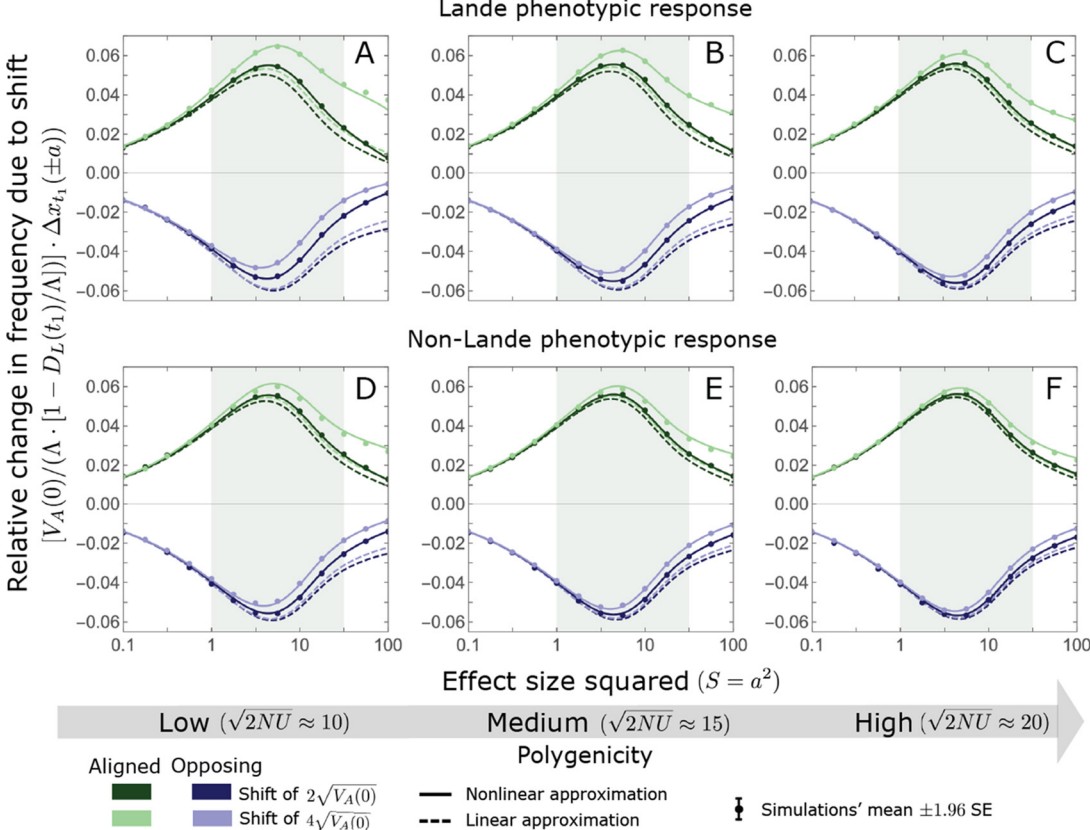

**Appendix 3—figure 9.** A comparison of the linear and nonlinear approximations with simulation results. The simulation results were generated using the *single allele* simulation, as described in Section 2.1; errors are not visible because they are smaller than the points. The analytic predictions were based on the linear (*Equations*

*Appendix 3—figure 9 continued on next page*

*Appendix 3—figure 9 continued*

**A3-40** and **A3-45**) and nonlinear (**Equations A3-40** and **A3-48**) approximations averaged over the initial MAF distribution (**Equation A3-27**) with the corresponding parameters. The nonlinear approximation is quite accurate throughout and substantially more accurate than the linear approximation for larger effect sizes and low polygenicity (see, e.g., **A**, **B**, and **D**). In the linear approximation, frequency changes with directional selection alone scale with $\Lambda/V_A(0) \cdot (1 - D_L(t_1)/\Lambda)$ (**Equations A3-39**, **A3-40** and **A3-45**). We therefore normalized the frequency changes by this factor to make the results for the two shift sizes and three extents of polygenicity comparable.

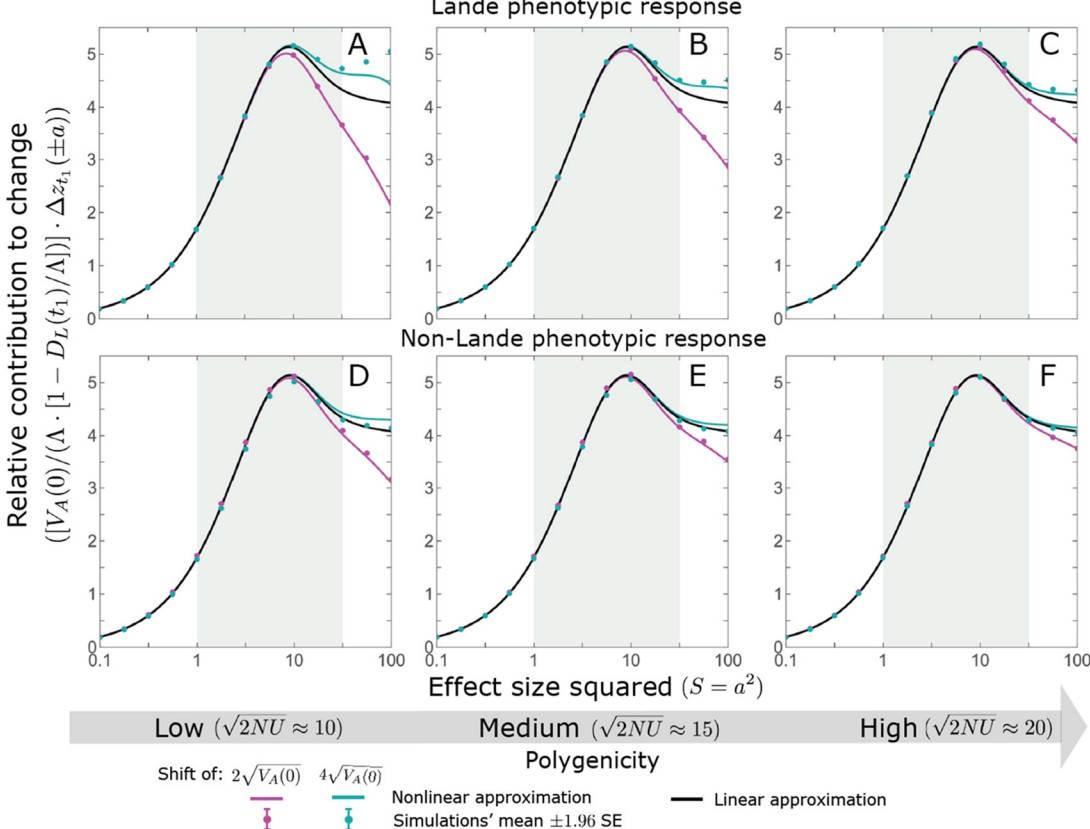

**Appendix 3—figure 10.** The allelic contribution to change in mean phenotype during the rapid phase. The simulation results were generated using the *single allele* simulation, as described in Section 2.1; errors are not visible because they are smaller than the points. The analytic predictions were based on the linear (**Equations A3-53** and **A3-54** or **Equation 13** in the main text) and nonlinear (**Equations A3-48**, **A3-53** and **A3-50**) approximations with the corresponding parameters. As for the frequency changes, the nonlinear approximation for the expected contribution to change is quite accurate throughout and substantially more accurate than the linear approximation for larger effect sizes and low polygenicity (see, e.g. **A** and **B**). In the linear approximation, the contribution to change in mean phenotype scales with $\Lambda/V_A(0) \cdot (1 - D_L(t_1)/\Lambda)$ (see **Equations A3-53** and **A3-54**). We therefore normalized the contributions by this factor to make them comparable for different initial variances and shift sizes.

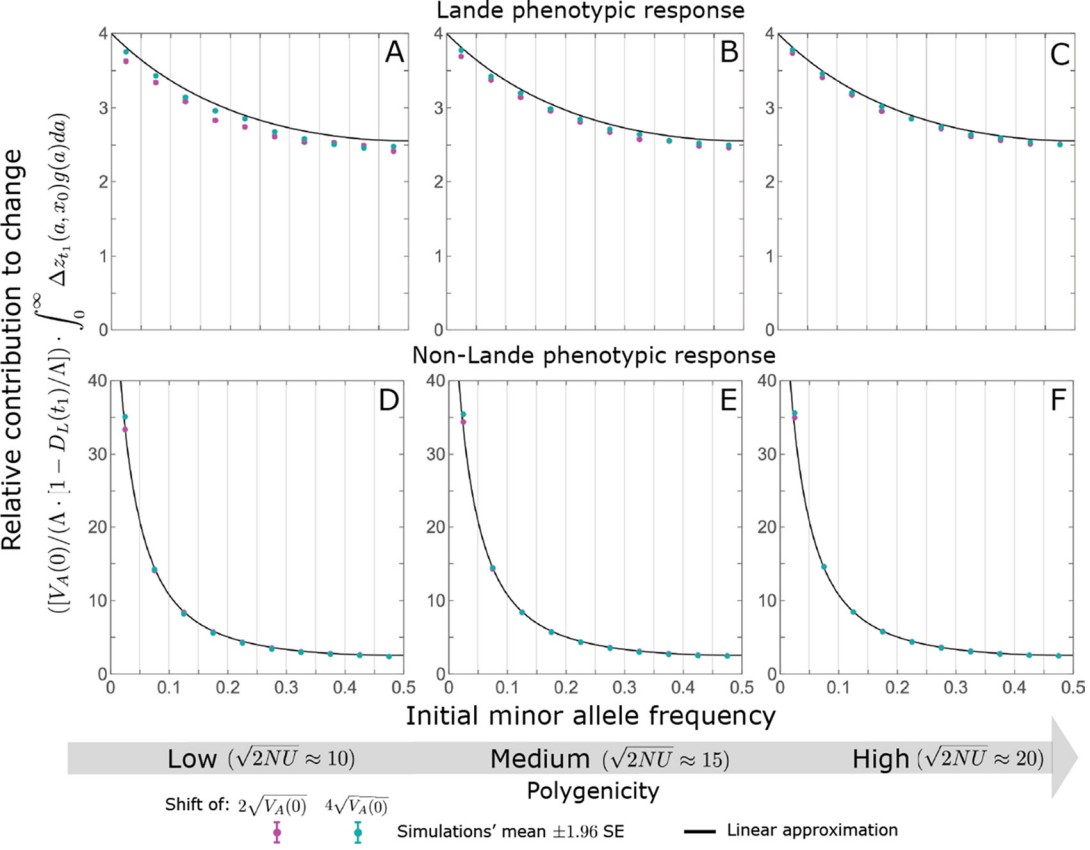

**Appendix 3—figure 11.** The relative contribution to change in mean phenotype during the rapid phase coming from different initial MAFs is well approximated by the relative contribution to phenotypic variance at equilibrium, as predicted by the linear approximation (*Equations A3-52* and *A3-54* or *Equation 12* in the main text). The simulation results were generated using the *all alleles* simulation, as described in Section 2.1. These results were binned by initial minor allele frequency (with boundaries between the bins marked by the light grey vertical lines) and the total contributions of alleles in frequency bin $(x_1, x_2)$ is divided by the total mutational input per generation times the size of the bin, $2NU(x_2 - x_1)$. As in *Appendix 3—figure 10*, we normalized the contributions by $\Lambda/V_A(0) \cdot (1 - D_L(t_1)/\Lambda)$ to make them comparable for different initial variances and shift sizes. Errors are often not visible because they are smaller than the points.

## 4.1.2 The nonlinear approximation

We build on the linear approximation to derive a nonlinear approximation that is more accurate, especially for large effect sizes, toward the end of the rapid phase. To this end, we substitute the frequency based on the linear approximation, $x_t^l$, for $x_t$ in the expression for the average stabilizing selection coefficient (*Equation A3-39*), to obtain:

$$
\begin{aligned}
\bar{s}_S^n (a, x_0, t) &\equiv -\frac{a^2}{V_S} \cdot \overline{(1/2 - x_t^l(a, x_0)) \cdot (1 - D_L^2/V_S)} \\
&= -\frac{a^2}{V_S} \cdot \left( (1/2 - x_0)(1 - \overline{D_L^2}(t)/V_S) - ax_0(1 - x_0)(\pm I_D(t) - a \cdot (1/2 - x_0) \cdot I_S(t)) \right),
\end{aligned}
$$
(A3-46)

where

$$
\begin{aligned}
I_D(t) &\equiv \frac{\Lambda}{V_A} \cdot \left( 1 - \overline{D_L}(t)/\Lambda \cdot \left( 1 + 1/6 \cdot \frac{\Lambda^2}{V_S} \cdot (1 - D_L(t)/\Lambda) \cdot (1 + 2 \cdot D_L(t)/\Lambda) \right) \right), \\
I_S(t) &\equiv \frac{1}{2} \cdot \frac{t}{2V_S} \left( 1 - \overline{D_L^2}(t)/V_S \cdot \left( 2 - \overline{D_L^2}(t)/V_S \right) \right),
\end{aligned}
$$
(A3-47)

and with $\overline{D_L^2}(t)$ given in *Equation A3-44*. We obtain an explicit, nonlinear approximation (which is nonlinear in $(\bar{s}_D(a, t) + \bar{s}_S^n(a, x_0, t)) \cdot t$) by substituting $\bar{s}_S^n(a, x_0, t)$ for $\bar{s}_S(a, x_0, t)$ into our implicit form solution (*Equation A3-37*), i.e.,

$$F_t(a, x_0) \approx \frac{1}{a} \cdot \left( \frac{\mathrm{Exp}\left[ \left( \bar{s}_D\left(a, t\right) + \bar{s}_S^n\left(a, x_0, t\right) \right) \cdot t \right] - 1}{1 + x_0 \left( \mathrm{Exp}\left[ \left( \bar{s}_D\left(a, t\right) + \bar{s}_S^n\left(a, x_0, t\right) \right) \cdot t \right] - 1 \right)} \right), \tag{A3-48}$$

with $\bar{s}_D$ given in *Equation A3-39*. The *nonlinear approximation* is quite accurate under a wide range of parameter values (*Appendix 3—figures 8* and *9*). It underestimates the increase in frequency of large effect, aligned alleles when the shift is large and polygenicity is low (*Appendix 3—figure 9C*), because $|\bar{s}_S^n|$ is greater than $|\bar{s}_S|$ when directional selection is strong and lasts longer.

## 4.2. Contribution to phenotypic change

Phenotypic adaptation after the shift in optimum arises from the increase in frequency of alleles whose effects align with the shift relative to those with opposite effects. Here, we generalize the steps that we took in the main text in order to approximate the allelic contributions to change in mean phenotype. We begin by considering a pair of alleles that are initially minor, with the same frequency, and that have opposite effects of the same magnitude. The frequency difference between them at time $t$ during the rapid phase is given by

$$\Delta x_t^*\left(a, x_0\right) \equiv \Delta x_t\left(a, x_0\right) - \Delta x_t\left(-a, x_0\right) = 2a x_0\left(1 - x_0\right) \cdot \overline{F}_t(a, x_0), \tag{A3-49}$$

where

$$\overline{F}_t(a, x_0) \equiv (F_t(a, x_0) + F_t(-a, x_0))/2. \tag{A3-50}$$

The pair's contribution to the change in mean phenotype is

$$\Delta z_t^*\left(a, x_0\right) = 2a \cdot \Delta x_t^*\left(a, x_0\right) = 2v^*\left(a, x_0\right) \cdot \overline{F}_t(a, x_0), \tag{A3-51}$$

where $v^*\left(a, x_0\right) = 2a^2 x_0(1 - x_0)$ is the initial contribution of each of the alleles to phenotypic variance. The expected contribution *per unit mutational input* of alleles with a given magnitude and initial MAF follows from multiplying the contribution of a pair by the density of pairs, $\rho(a, x_0)$, namely,

$$\Delta z_t\left(a, x_0\right) = \rho(a, x_0) \cdot \Delta z_t^*\left(a, x_0\right) = v\left(a, x_0\right) \cdot \overline{F}_t(a, x_0), \tag{A3-52}$$

where $v\left(a, x_0\right) = 2\rho(a, x_0) \cdot v^*\left(a, x_0\right)$ is the density of genetic variance at equilibrium (Section 3.2). The expected marginal contribution per unit mutational input of alleles with a given magnitude $a$ is given by

$$\Delta z_t\left(a\right) = \int_0^{1/2} \Delta z_t\left(a, x_0\right) dx_0 = \int_0^{1/2} v\left(a, x_0\right) \cdot \overline{F}_t\left(a, x_0\right) dx_0. \tag{A3-53}$$

Lastly, the expected total contribution of these alleles to phenotypic change follows from multiplying $\Delta z_t\left(a\right)$ by the mutational input per generation, $2NU \cdot g(a)$.

We can now use our approximations of $F_t\left(a, x_0\right)$ to obtain corresponding approximations for the allelic contributions to phenotypic change. Notably, in the linear approximation (*Equations A42– A45*), $\overline{F}_t$ takes the simple form

$$\overline{F}_t \approx \Lambda/V_A(0) \cdot \left(1 - D_L(t)/\Lambda\right), \tag{A3-54}$$

which is independent of magnitude and initial frequency (which is why we defined $F_t$ as we did). We use this approximation in our expressions for the phenotypic contribution in the main text. It captures the qualitative properties of the allelic contribution and is also fairly accurate for small and intermediate effect alleles.

When the rapid phase is longer, the contributions of large effect alleles substantially deviate from the linear approximation in one of two ways. First, when the shift in optimum is large, large effect alleles contribute more than predicted by the linear approximation, because the change in frequency of the aligned allele accelerates as its frequency increases. Second, when the shift is small, large effect alleles contribute less than predicted, because the linear approximation underestimates the reduction in the frequency of both aligned and opposite alleles caused by stabilizing selection. Both of these effects are captured by the nonlinear approximation (see *Appendix 3—figure 10*).

## 5. Allele dynamics during equilibration

In the long run, phenotypic adaptation transitions from being based on small frequency differences between alleles whose effects align with and oppose the shift in optimum to being based on small differences between the numbers of fixations of these opposite alleles (*Figure 3*). Here, we characterize the architecture of these long-term fixed differences. Specifically, we derive approximations for their relative contributions to phenotypic change as a function of allele magnitude and initial frequency (before the shift in optimum).

To this end, we approximate an allele's probability of fixation in two steps. First, we model the effect of directional selection on frequency as an instantaneous, deterministic pulse. We assume that the pulse occurs immediately after the shift for alleles already present in the population, and immediately after introduction for mutations that occur after the shift. Second, we apply the diffusion approximation for the fixation probability, assuming stationary stabilizing selection and genetic drift and given the allele frequency immediately after the instantaneous pulse of directional selection.

We consider four approximations of increasing complexity, which differ in two of their assumptions. The first is whether the phenotypic approach to the new optimum is well approximated by Lande's solution; we refer to the corresponding approximations as *Lande* and *non-Lande*. The second is whether the expected changes to allele frequency due to directional selection are sufficiently small that we can apply linear approximations in calculating these changes and their effects on fixation probabilities; we refer to the corresponding approximations as *linear* and *nonlinear* (the two *linear* approximations were also described in the main text). The derivations for changes to allele frequencies due to directional selection in the *linear* and *nonlinear* approximations are analogous to those of the corresponding approximations for the rapid phase. We consider each of the four approximations that follow from the combinations of these assumptions in turn.

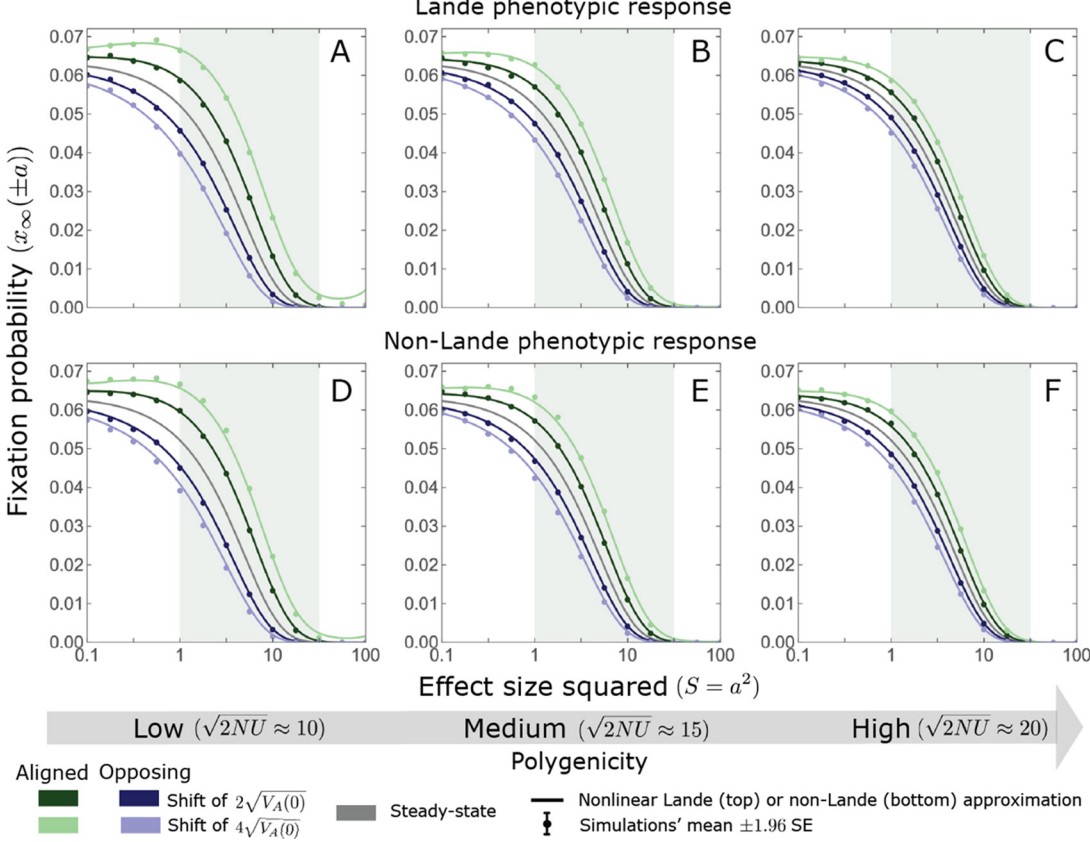

**Appendix 3—figure 12.** For small and intermediate alleles, changes in fixation probability are fairly small relative to the fixation probability at equilibrium. The simulation results were generated using the *single allele* simulation, as described in Section 2.1; errors are not visible because they are smaller than the points. The analytic predictions were based on the nonlinear Lande (*Equations A3-60*, *A3-84* and *A3-85*) and non-Lande (*Equations A3-60*, *A3-88* and *A3-89*) approximations, averaged over the initial MAF distribution (*Equation A3-27*) with the corresponding parameters.

## 5.1. The linear Lande approximation

The linear Lande approximation is the simplest and the one that we detail in the main text. Here, we briefly describe it for completeness and detail the conditions on model parameters under which we expect it to be accurate.

First, we approximate the change in allele frequency caused by directional selection. We use a deterministic approximation, which neglects the effects of drift, and in contrast to our approximations for the rapid phase, also neglects the effects of stabilizing selection. Under these simplifications, the change in allele frequency due to directional selection can be described by the ODE

$$\frac{dx}{dt} = \frac{s_d(t)}{2} x \left(1 - x\right) \text{ with } s_d\left(t\right) = \pm 2a \cdot D\left(t\right)/V_S \tag{A3-55}$$

(see *Equation A3-36*). Further assuming that relative changes to allele frequency due to directional selection are small, we can approximate this ODE with

$$\frac{dx}{dt} = \frac{s_d(t)}{2} x_0(1 - x_0), \tag{A3-56}$$

where $x_0$ is the initial frequency, before the shift in optimum. The total effect of directional selection is then given by integrating this ODE:

$$\Delta x_d\left(\pm a, x_0\right) \equiv x_d\left(\pm a, x_0\right) - x_0 \approx \pm a x_0\left(1 - x_0\right) \int_0^\infty \left(D\left(\tau\right)/V_S\right) d\tau. \tag{A3-57}$$

Here and throughout this section, we approximate the integral of the distance from the optimum by its expectation, where we assume that this expectation is finite and integrable; for brevity, we denote this expectation by $\int D(\tau)d\tau$. Further assuming Lande's approximation for the change in mean phenotype

$$\int_0^\infty D(\tau)d\tau = \int_0^\infty D_L\left(\tau\right) d\tau = \Lambda \cdot V_S/V_A(0), \tag{A3-58}$$

implying that

$$\Delta x_d\left(\pm a, x_0\right) \approx \pm a x_0\left(1 - x_0\right) \cdot \Lambda/V_A(0). \tag{A3-59}$$

We assume that this change in frequency occurs instantaneously, immediately after the shift in optimum.

Next, we approximate the fixation probability, using the diffusion approximation with stationary stabilizing selection and drift (*Equation A3-16*) and the initial allele frequency after the instantaneous pulse due to directional selection (*Equation A3-57*). We denote the expected allele frequency in the long run, which equals the fixation probability, by $x_\infty\left(\pm a, x_0\right)$ and the long-term change in frequency (or fixation probability) due to directional selection by

$$\Delta x_\infty\left(\pm a, x_0\right) \equiv x_\infty\left(\pm a, x_0\right) - \pi\left(\pm a, x_0\right) = \pi\left(a, x_d(\pm a, x_0)\right) - \pi\left(a, x_0\right). \tag{A3-60}$$

Assuming that changes in frequencies due to directional selection are small, we can approximate their long-term effect on frequency by

$$\begin{aligned}\Delta x_\infty\left(\pm a, x_0\right) &\approx \frac{\partial \pi}{\partial x}\left(a, x_0\right) \cdot \Delta x_d\left(\pm a, x_0\right) = \pm \frac{2f(a)}{v\left(a, x_0\right)} \cdot a x_0\left(1 - x_0\right) \cdot \Lambda/V_A(0) \\ &= \pm \frac{f(a)}{a} \cdot \frac{v^*\left(a, x_0\right)}{v\left(a, x_0\right)} \cdot \Lambda/V_A(0),\end{aligned} \tag{A3-61}$$

where, as shown in Section 3 (*Equation A3-17*), $\partial_x \pi\left(a, x\right) = 2f(a)/v(a, x)$ with $f\left(a\right) \equiv 2a^3 \cdot \text{Exp}\left[-a^2/4\right] / \left(\sqrt{\pi} \cdot \text{Erf}\left[a/2\right]\right)$. The expected long-term fixed contribution to phenotypic change of a pair of opposite minor alleles with effect sizes $\pm a$ and initial frequency $x_0$ is then approximated by

$$\begin{aligned}\Delta z_\infty^*\left(a, x_0\right) &= 2a\left(x_\infty\left(a, x_0\right) - x_\infty\left(-a, x_0\right)\right) \approx \Delta z_d^*\left(a, x_0\right) \cdot \frac{\partial \pi}{\partial x}\left(a, x_0\right) \\ &= 2 \cdot \Lambda/V_A(0) \cdot f(a) \cdot \frac{2v^*\left(a, x_0\right)}{v\left(a, x_0\right)},\end{aligned} \tag{A3-62}$$

where $\Delta z_d^*\left(a, x_0\right) \equiv 2a\left(x_d\left(a, x_0\right) - x_d\left(-a, x_0\right)\right)$. The contribution per unit mutational input of such pairs is approximated by

$$\Delta z_\infty \left(a, x_0\right) = \Delta z_\infty^* \left(a, x_0\right) \cdot \rho(a, x_0) = 2 \cdot \Lambda/V_A(0) \cdot \frac{v^*\left(a, x_0\right) \cdot 2\rho(a, x_0)}{v\left(a, x_0\right)} \cdot f(a)$$
$$= 2 \cdot \Lambda/V_A(0) \cdot f(a), \tag{A3-63}$$

which is independent of the initial allele frequency. The expected marginal contribution per unit mutational input of alleles with magnitude $a$ is approximated by

$$\Delta z_\infty \left(a\right) = \int_0^{1/2} \Delta z_\infty \left(a, x_0\right) dx_0 \approx \frac{\Lambda}{V_A(0)} \cdot f(a). \tag{A3-64}$$

Lastly, the approximation for the total long-term contribution to change in mean phenotype is

$$2NU \cdot \Delta z_\infty = 2NU \cdot \int_0^\infty \Delta z_\infty(a) \cdot g(a)da \approx \Lambda \cdot \frac{2NU \cdot \int_0^\infty f(a) \cdot g(a)da}{V_A(0)}$$
$$= \Lambda \cdot \frac{\int_0^\infty f(a) \cdot g(a)da}{\int_0^\infty v(a) \cdot g(a)da} = \frac{\Lambda}{1+C}, \tag{A3-65}$$

where $C \equiv \frac{\int_0^\infty v(a) \cdot g(a)da}{\int_0^\infty f(a) \cdot g(a)da} - 1.$

This leads to the first of two conditions under which we expect the linear Lande approximation to be accurate. In the long run, we expect the total change in mean phenotype to be equal to the shift in optimum. *Equation A3-65* implies that a necessary condition for this to be the case is that

$$C \equiv \frac{\int_0^\infty v(a) \cdot g(a)da}{\int_0^\infty f(a) \cdot g(a)da} - 1 \ll 1. \tag{A3-66}$$

This condition implies that the bulk of genetic variance before the shift arises from alleles with fairly small effects (because $f(a) \approx v(a)$ for $a^2 \lesssim 4$, but $f$ is substantially smaller than $v$ for larger effect sizes; *Figure 5*). The second condition ensures that our linear expansions are accurate. In Section 5.3, we show that it will be the case when

$$a \cdot \Lambda/V_A(0) \ll 1. \tag{A3-67}$$

Under our conditions on parameters $\Lambda/V_A(0) \lesssim 1/2$ (*Equation A3-32*), implying that condition *A3-67* necessarily holds for alleles with $a \ll 2$. For alleles with larger effects, this condition should hold when: (i) the shift in optimum is not too large compared to the phenotypic standard deviation (e.g. $\Lambda/\sqrt{V_A(0)} \sim 1$); (ii) the trait is sufficiently polygenic; specifically when $\sqrt{2NU} \gg a$ (because $\sqrt{V_A(0)} \sim \sqrt{2NU}$ under our assumption that a substantial proportion of mutations are not effectively neutral—see *Equation A3-30*). With regard to (ii), we already require that $\sqrt{2NU} \gg 1$ (*Appendix 1— table 2*), and when condition *A3-66* holds we also expect the bulk of the long-term contribution to phenotypic adaptation to come from alleles with $a \leq 2$. *Appendix 3—figures 13–15* illustrate that the linear Lande approximation is accurate for $a < 2$ when *A3-66* holds.

While we do not prove that our conditions are sufficient, we conjecture that they are, for the following reasons. First, in Sections 6 and 9 we illustrate by simulation that Lande's approximation is accurate when condition *A3-66* is met. Intuitively, when the bulk of variation has fairly small effect sizes, we expect little build-up of a $3^{rd}$ central moment during the rapid phase (also see Section 3.3). When Lande's approximation applies, directional selection has non-negligible effects for only a brief period after the shift in optimum (for $\sim 1/U$ generations; see section on *Allele dynamics*). Consequently, these effects should be well approximated as an instantaneous pulse that neglects drift and stabilizing selection, whose effects manifest over longer timescales. Additionally, newly arising mutations should contribute negligibly to phenotypic change. This is because fairly few new mutations arise while directional selection is effective (i.e. during the rapid phase), and because the fixation probabilities (let alone the difference in fixation probabilities between new mutations with opposing effects) of those that do are minute, given that they start from an initial frequency of $1/(2N)$.

Second, we require that our first order approximations for the change in allele frequency (*Equation A3-57*) and change in fixation probability (*Equation A3-61*) be accurate. With high polygenicity and a moderate shift, we expect the average contribution to phenotypic change per segregating allele to be small. Nevertheless, large effect alleles could experience changes in frequency that are large relative to their low initial frequencies, causing a substantial relative error in our linear instantaneous pulse approximation (*Equation A3-57*). This reasoning provides some intuition for why condition

*A3-67* combines requirements on the extent of polygenicity, shift size, and effect size. In Section 5.3, we show why this particular combination is required.

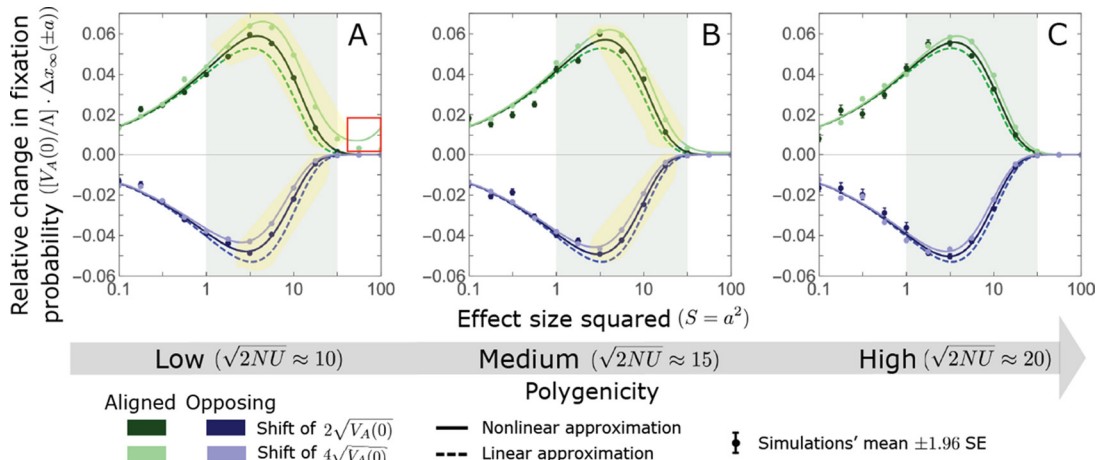

**Appendix 3—figure 13.** The Lande case: A comparison the linear and nonlinear approximations for the expected change in fixation probability of alleles already present at the time of the shift. The simulation results were generated using the *single allele* simulation, as described in Section 2.1; errors are often not visible because they are smaller than the points. The analytic predictions were based on the linear Lande (*Equations A3-60* and *A3-61*) and nonlinear Lande (*Equations A3-60*, *A3-84* and *A3-85*) approximations averaged over the initial MAF distribution (*Equation A3-27*) with the corresponding parameters. The nonlinear Lande approximation performs better than the linear Lande approximation for alleles with large and intermediate effects, especially when polygenicity is lower (see the golden shaded regions in **A** and **B**). In the linear approximation, the change in fixation probability scales with $\Lambda/V_A(0)$ (*Equation A3-61*). We therefore divided the changes in fixation probability by this factor to make them comparable for different initial variances and shift sizes. Note that with this normalization, the linear curves for the two shifts are identical, which is why only one is visible.

## 5.2. The linear non-Lande approximation

Next, we derive approximations for the case in which our linear expansions are accurate but Lande's approximation is not. Lande's approximation for the mean performs poorly when alleles with larger effect sizes ($a^2 \geq 4$) contribute substantially to genetic variance before the shift, or more precisely when $C$ is substantial (see *Equation A3-65*). As we describe in the main text, in this case, the phenotypic distribution develops a substantial 3rd central moment during the rapid phase, leading to a much slower approach to the new optimum afterwards. This results in prolonged weak directional selection that amplifies the difference in fixation probabilities between alleles with opposite effects present before the shift in optimum. In addition, despite it being weak, directional selection acts for so long (on the order of $N$ generations; see *Figure 2D* and *Appendix 3—figures 20* and *26*) that it can produce a substantial cumulative difference between the numbers of fixations of mutations with opposite effects that arise after the shift in optimum. In what follows, we denote the proportions of the long-term (fixed) contribution to phenotypic change that arise from standing variation and from new mutations by $\eta$ and $1 - \eta$ respectively.

## 5.2.1 Standing variation

First, we consider the contribution from standing variation. In this case, the integral effect of directional selection is greater than under Lande's approximation,

$$\int_0^\infty D(\tau)d\tau > \int_0^\infty D_L(\tau)\,d\tau = \Lambda \cdot V_S/V_A(0). \tag{A3-68}$$

Since directional selection acts on the order of $N$ generations after the shift, an allele present at the time of the shift may not experience the full integral effect of directional selection. We define $t_*$ as the effective number of generations over which an allele is subject to directional selection (which we assume to be independent of $a$ and $x_0$) and $A > 0$ such that

$$\int_0^{t_*} D(\tau)d\tau = (1 + A) \cdot \int_0^\infty D_L(\tau)\,d\tau = (1 + A) \cdot \Lambda \cdot V_S/V_A(0). \tag{A3-69}$$

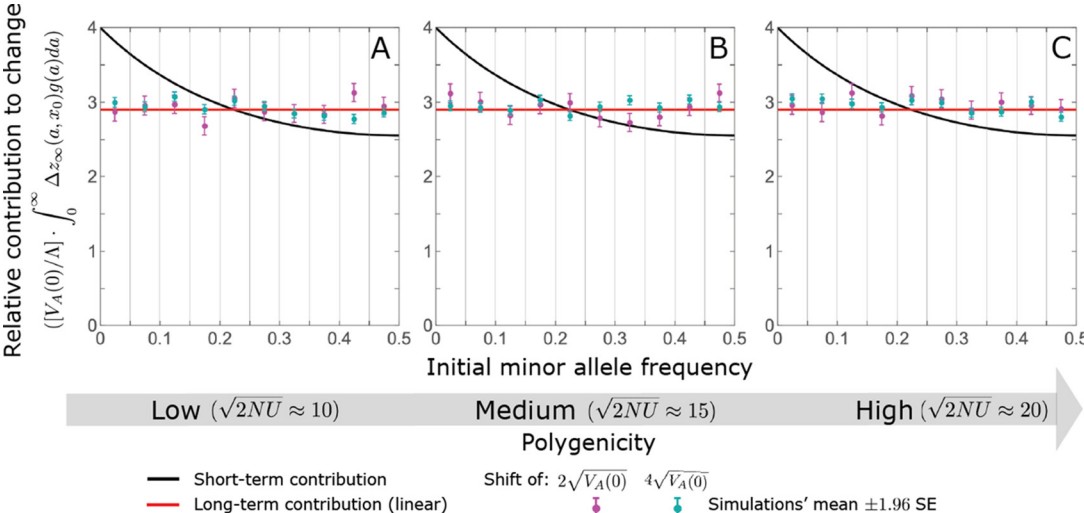

**Appendix 3—figure 14.** The Lande case: the long-term relative contribution to the change in mean phenotype coming from different initial MAFs is approximately constant. The simulation results were generated using the *all alleles* simulation, as described in *Choice of simulation parameters*. The results for a given case were binned by initial minor allele frequency (with boundaries between the bins marked by the light grey vertical lines). The total contributions of alleles in frequency bin $(x_1, x_2]$ is divided by the total mutational input per generation times the size of the bin, $2NU(x_2 - x_1)$. The analytic predictions were based on the linear Lande approximation (*Equation A3-63* or *Equation 19* in the main text).

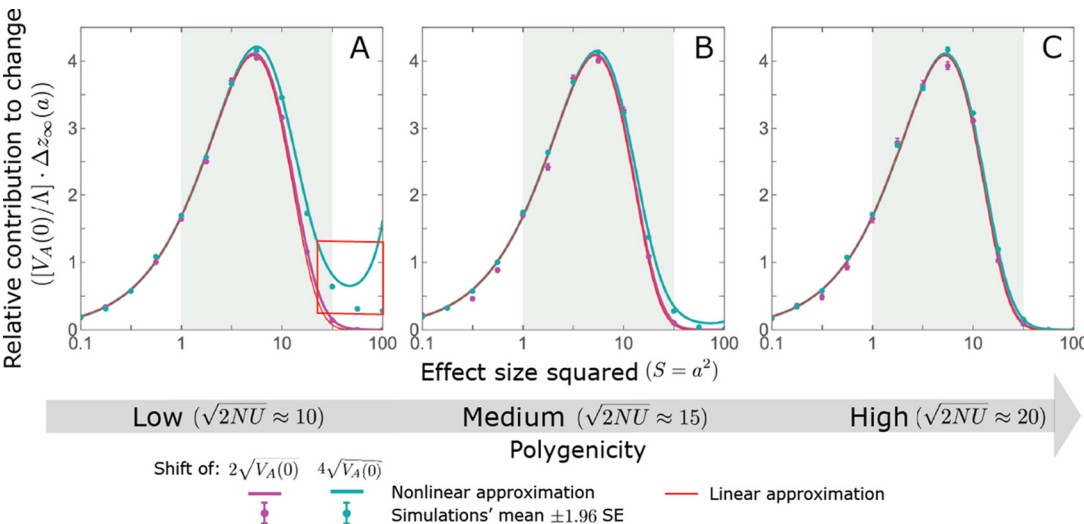

**Appendix 3—figure 15.** The Lande case: As predicted by the linear approximation, the long-term relative contribution to the change in mean phenotype coming from standing variation scales with $\Lambda/V_A(0)$. Both the linear and nonlinear Lande approximation are fairly accurate for alleles with small effects. When polygenicity is low, the nonlinear Lande approximation is more accurate for alleles with intermediate effects (see the teal line in **A** and **B**). Both approximations perform badly for large effect alleles when the polygenicity is low and the shift is large (the red box in **A**). However, in the Lande case, we expect very few alleles to have large effect sizes. The simulation results were generated using the *single allele* simulation, as described in Section 2.1; errors are often not visible because they are smaller than the points. The analytic predictions were based on the linear Lande (*Equation A3-64* or *Equation 20* in the main text) and nonlinear Lande (*Equations A3-84–A3-A88*) approximations.

By analogy with the derivations of the previous section, the instantaneous pulse approximation for the effect of directional selection is

$$\Delta x_d\left(a, x_0\right) \approx ax_0\left(1 - x_0\right) \cdot \int_0^{t^*}\left(D\left(\tau\right)/V_S\right)d\tau = ax_0\left(1 - x_0\right) \cdot (1 + A) \cdot \Lambda/V_A(0). \tag{A3-70}$$

The expressions for the fixed contribution are also analogous to those derived in the previous section, where here we replace $\Lambda/V_A(0)$ by $(1 + A) \cdot \Lambda/V_A(0)$. In particular, the expected long-term (fixed) contribution to phenotypic change per unit mutational input of pairs of opposite alleles with effect sizes $\pm a$ and initial MAF $x_0$ is approximated by

$$\Delta z_\infty\left(a, x_0\right) = \Delta z_d\left(a, x_0\right) \cdot \frac{2f\left(a\right)}{v\left(a, x_0\right)} \approx (1 + A) \cdot \frac{\Lambda}{V_A(0)} \cdot 2f(a), \tag{A3-71}$$

where $\Delta z_d\left(a, x_0\right) \equiv 2a(x_d\left(a, x_0\right) - x_d\left(-a, x_0\right)) \cdot \rho(a, x_0) \approx v\left(a, x_0\right) \cdot (1 + A) \cdot \Lambda/V_A(0)$, and $\rho(a, x_0)$ is the density of pairs of opposite alleles per unit mutational input segregating with magnitude $a$ and MAF $x_0$ at equilibrium (**Equation A3-23**). Similar to what we obtained for the linear Lande approximation, this contribution is independent of the initial allele frequency. The expected marginal contribution per unit mutational input of pairs of alleles with magnitude $a$ is approximated by

$$\Delta z_\infty^v\left(a\right) = \int_0^{1/2}\Delta z_\infty\left(a, x_0\right) dx_0 \approx (1 + A) \cdot \Lambda/V_A(0) \cdot f(a). \tag{A3-72}$$

Lastly, the total fixed contribution to phenotypic change from standing variation is approximated by

$$2NU \cdot \Delta z_\infty^v = 2NU \cdot \int_0^\infty \Delta z_\infty^v(a) \cdot g(a)da \approx \frac{2NU \cdot \int_0^\infty f(a) \cdot g(a)da}{V_A(0)} \cdot (1 + A) \cdot \Lambda = \frac{1 + A}{1 + C} \cdot \Lambda. \tag{A3-73}$$

Thus, we find that the proportional long-term contribution from standing variation is

$$\eta = \frac{1 + A}{1 + C}. \tag{A3-74}$$

In this case, the justification for the instantaneous pulse approximation is less obvious, because directional selection substantially affects allele frequencies over an extended period. Assuming that the pulse approximation is accurate nonetheless, the same reasoning that we employed for the linear Lande approximation suggests that our linear expansions are accurate when polygenicity is sufficiently high and the shift and allele magnitudes are not too large. However, given that directional selection acts for longer, we expect the condition on these quantities to be more stringent than for the linear Lande approximation. By analogy, we would conjecture that this condition becomes $a \cdot (1 + A) \cdot \Lambda/V_A(0) \ll 1$ (see Section 5.4).

To test the linear non-Lande approximation for standing variation, we estimate the proportional contribution of standing variation to phenotypic change, $\eta$, using our *all alleles* simulations, and then calculate $A$ using **Equation A3-74**. We find the approximation to be quite accurate provided high polygenicity and a moderate shift (**Appendix 3—figures 16–18**). However, when polygenicity is low and/or the shift is large, it underestimates the fixation probabilities of both aligned and opposing intermediate effect alleles (see the golden shaded region in **Appendix 3—figure 16**). The nonlinear non-Lande approximation performs better in some of these cases (see Section 5.4).

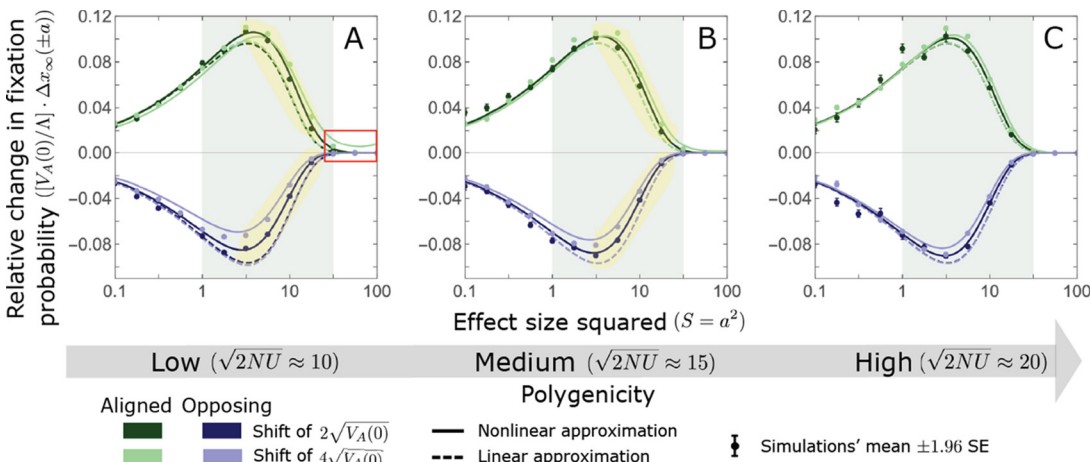

**Appendix 3—figure 16.** The non-Lande case: A comparison of the linear and nonlinear approximations for the expected change in fixation probability of alleles segregating at the time of the shift. The simulation results were generated using the *single* allele simulation, as described in Section 2.1; errors are often not visible because they are smaller than the points. The analytic predictions were based on the linear non-Lande (*Equations A3-60 and A3-70*) and nonlinear non-Lande (*Equations A3-60*, *A3-88* and *A3-89*) approximations averaged over the initial MAF distribution (*Equation A3-27*) with the corresponding parameters. The nonlinear approximation performs better than the linear one for intermediate effect alleles when polygenicity is low (see the gold shaded regions in **A** and **B**). As in *Appendix 3—figure 13*, we normalize the changes in fixation probability by $\Lambda/V_A(0)$ to make them comparable for different initial variances and shift sizes (*Equation A3-61*).

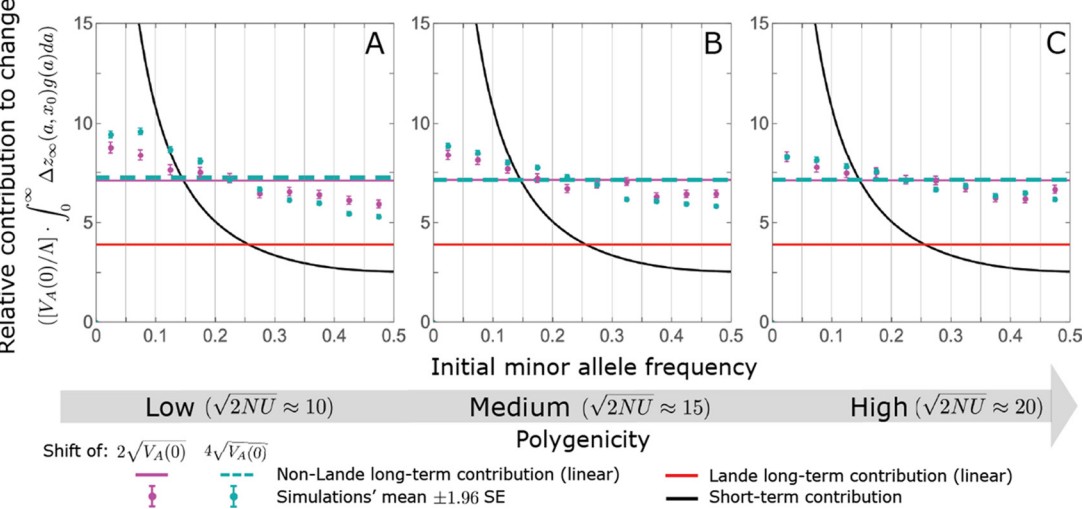

**Appendix 3—figure 17.** The non-Lande case: The long-term relative contribution to the change in mean phenotype coming from different initial MAFs is approximately constant, with low initial MAF alleles contributing slightly more than high initial MAF alleles. The slight excess in the contribution of alleles with low initial MAF is especially pronounced when the shift is large and the polygenicity is low. The simulation results were generated using the *all alleles* simulation, as described in Section 2.1. We calculate the relative contribution of alleles in each MAF bin $(x_1, x_2)$ (between the gray gridlines) by dividing the contribution of fixations in the bin by the total mutational input per generation times the size of the bin, $2NU(x_2 - x_1)$. The analytic predictions were based on the linear non-Lande approximation (*Equation A3-71* or *Equation 24* in the main text).

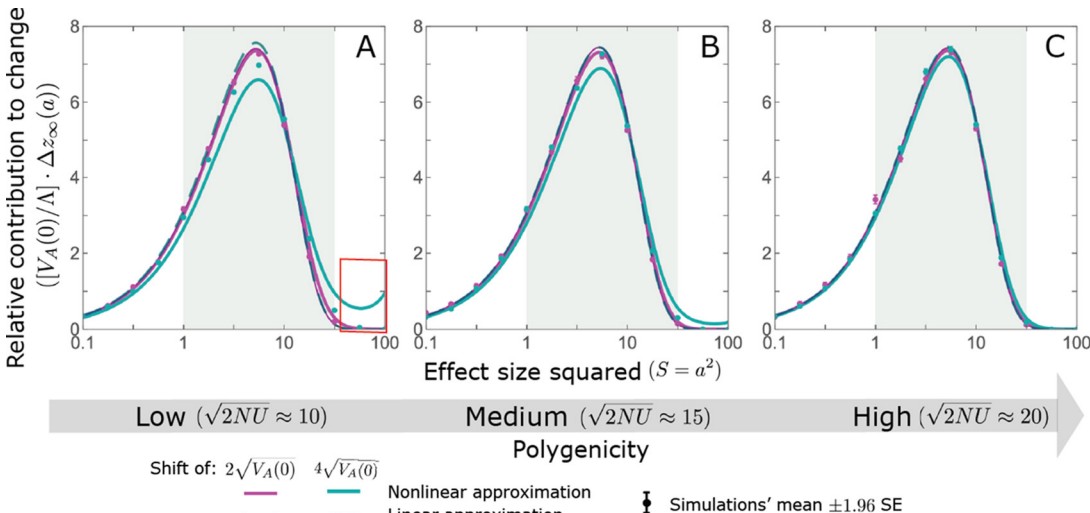

**Appendix 3—figure 18.** The non-Lande case: As predicted by the linear approximation, the long-term relative contribution to the change in mean phenotype coming from standing variation scales with $\Lambda/V_A(0)$. Both the linear and nonlinear approximations predict the long-term relative contribution to the change in mean phenotype fairly well. However, when polygenicity is low and the shift is large, the nonlinear approximation overestimates the proportion of the change in mean due to large effect alleles (see the red box in **A**). The simulation results were generated using the *single allele* simulation, as described in Section 2.1; errors are often not visible because they are smaller than the points. The analytic predictions were based on the linear non-Lande (**Equation A3-72** or **Equation 25** in the main text) and nonlinear non-Lande (**Equations A3-86–A3-89**) approximations.

## 5.2.2 New mutations

First, we consider a new mutation with effect size $\pm a$ that arose $t$ generations after the shift in optimum. The instantaneous pulse approximation for the effect of directional selection on such a mutation is

$$\Delta x_d\left(\pm a | t\right) \approx \pm a \frac{1}{2N}\left(1 - \frac{1}{2N}\right) \cdot \int_{t}^{t+\Delta_t}\left(D\left(\tau\right)/V_S\right)d\tau. \tag{A3-75}$$

where $\Delta_t$ is the effective number of generations over which an allele arising in generation $t$ is subject to directional selection (which we assume to be independent of $a$). The expected change in mean phenotype caused by a pair of such mutations with opposite effects $\pm a$ is

$$\Delta z_d^*\left(a | t\right) \approx 2v^*\left(a, 1/2N\right) \cdot \int_{t}^{t+\Delta_t}\left(D\left(\tau\right)/V_S\right)d\tau. \tag{A3-76}$$

Assuming that $a \cdot \Delta x_d\left(a | t\right) \ll 1$, we can approximate the expected long-term (fixed) effect of such a pair by

$$
\begin{aligned}
\Delta z_\infty^*\left(a | t\right) &\approx 2v^*\left(a, 1/(2N)\right) \cdot \int_{t}^{t+\Delta_t}\left(D\left(\tau\right)/V_S\right)d\tau \cdot \frac{2f\left(a\right)}{v\left(a, 1/(2N)\right)} \\
&= \frac{2f\left(a\right)}{\rho\left(a, 1/(2N)\right)} \cdot \int_{t}^{t+\Delta_t}\left(D\left(\tau\right)/V_S\right)d\tau \\
&= 2f\left(a\right) \cdot \int_{t}^{t+\Delta_t}\left(D\left(\tau\right)/V_S\right)d\tau/(2N),
\end{aligned} \tag{A3-77}
$$

where $2v^*\left(a, 1/(2N)\right)/v\left(a, 1/(2N)\right) = \rho^{-1}\left(a, 1/(2N)\right) = 1/(2N)$. Multiplying this contribution by the number of such pairs per unit mutational input per generation, $1/2$, we find that the expected contribution of pairs per unit mutational input is approximated by

$$\Delta z_\infty \left(a|t\right) \approx f\left(a\right) \left( \frac{\int_t^{t+\Delta_t} \left(D\left(\tau\right)/V_S\right) d\tau}{2N} \right).$$

(A3-78)

Next, considering all mutations that arise after the shift in optimum, we find that the expected long-term contribution of mutations with magnitude $a$ is approximated by

$$\Delta z_\infty^m \left(a\right) = \int_0^\infty \Delta z_\infty \left(a|t\right) dt \approx B \cdot f\left(a\right) \cdot \Lambda/V_A(0),$$

(A3-79)

where

$$B \equiv \frac{\int_0^\infty \int_t^{t+\Delta_t} \left(D\left(\tau\right)/V_S\right) d\tau dt/(2N)}{\Lambda/V_A(0)}.$$

(A3-80)

Notably, we find that the relative contribution of mutations of any given magnitude (and that arise at any time) is the same as for standing variation, that is, proportional to $f(a)$.

We can now relate our approximations for new mutations with their total long-term (fixed) contribution to phenotypic change. To this end, we express **Equation A3-79** as

$$\Delta z_\infty^m(a) \approx B \cdot \Delta z_\infty^{lL}(a) = B \cdot f(a) \cdot \Lambda/V_A(0),$$

(A3-81)

where $\Delta z^{lL}$ is the linear Lande approximation for the corresponding contribution from standing variation. In these terms, our approximation for the total (fixed) contribution to phenotypic change from new mutations is

$$2NU \cdot \Delta z_\infty^m = 2NU \cdot \int_0^\infty \Delta z_\infty^m(a) \cdot g(a) da \approx \frac{2NU \cdot \int_0^\infty f(a) \cdot g(a) da}{V_A(0)} \cdot B \cdot \Lambda = \frac{B}{1+C} \cdot \Lambda.$$

(A3-82)

Thus, we find that the proportional long-term contribution from new mutations is $1 - \eta = B/(1+C)$. Further recalling that $\eta = (1+A)/(1+C)$, we also find that $A + B = C$.

We expect the linear non-Lande approximation for new mutations to be accurate under more general conditions than it is for standing variation. First, consider our linear expansion of the fixation probability (e.g., **Equation A3-61**), which is accurate when $a \cdot \Delta x_d(a|t) \ll 1$. We expect this to be the case for most if not all new mutations and model parameter ranges that meet our conditions (**Appendix 1—table 2**), because new mutations have a tiny initial frequency of $1/(2N)$ and the vast majority of them arise during the equilibration phase, when $D$ is very small. Second, consider the condition under which our linear approximation for $\Delta x_d(a|t)$ (**Equation A3-75**) is accurate. As we already noted (and argue in Section 5.4), we expect the linear non-Lande approximation for standing variation to be accurate when $a \cdot (1+A) \cdot \Lambda/V_A(0) \ll 1$. An analogous argument suggests this is case for new mutations when $a \cdot \int_t^{t+\Delta_t} \left(D\left(\tau\right)/V_S\right) d\tau \ll 1$. However, $\int_t^{t+\Delta_t} \left(D\left(\tau\right)/V_S\right) d\tau \leq \int_0^{t_*} \left(D\left(\tau\right)/V_S\right) d\tau = (1+A) \cdot \Lambda/V_A(0)$, since $D\left(t\right)$ decreases after the shift and alleles segregating at the time of the shift tend to segregate longer than new mutations (i.e., $t_* > \Delta_t$). Thus, with the exception of a few mutations that arise shortly after the shift, we expect that $a \cdot \int_t^{t+\Delta_t} \left(D\left(\tau\right)/V_S\right) d\tau \ll a \cdot (1+A) \cdot \Lambda/V_A(0)$.

### 5.2.3 Standing variation and new mutations

To test the linear non-Lande approximation for both standing variation and new mutations, we note that their joint contribution per unit mutational input of alleles with magnitude $a$ is approximated by

$$\Delta z_\infty^v \left(a\right) + \Delta z_\infty^m \left(a\right) \approx \left[\left(1+A+B\right) \cdot \Lambda/V_A(0)\right] \cdot f\left(a\right) = \left(1+C\right) \cdot f\left(a\right) \cdot \Lambda/V_A(0),$$

(A3-83)

where we have closed form expression for all parts of this expression. As we illustrate in **Appendix 3—figure 19**, the linear non-Lande approximation for this joint contribution is quite accurate provided high polygenicity and a moderate shift. However, when polygenicity is low and/or the shift is large, the linear approximation overestimates the contribution of intermediate effect alleles and underestimates the contribution of large effect alleles.

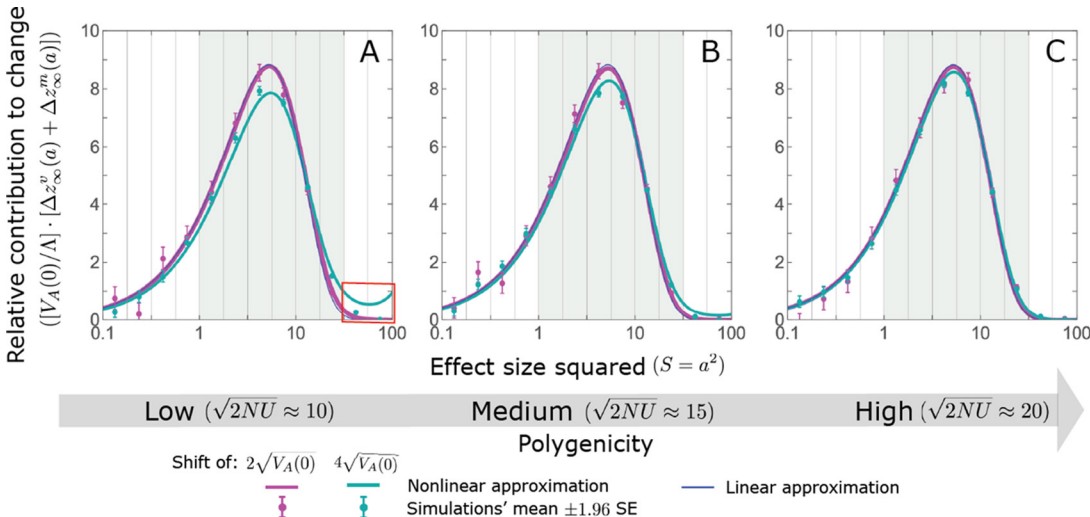

**Appendix 3—figure 19.** The non-Lande case: As predicted by the linear approximation, the long-term contribution, per unit mutational input, to the change in mean phenotype coming from standing variation and new mutations scales with $(1 + C) \cdot \Lambda/V_A(0)$ (**Equation A3-83**). Both the linear and nonlinear approximations perform fairly well. However, when the polygenicity is low and the shift is large, the nonlinear approximation overestimates the proportion of the change in mean due to large effect standing variation (**Appendix 3—figure 18**) and therefore overestimates the proportion contribution coming from *all* large effect alleles (see the red box in **A**). The simulation results were generated using the *all allele* simulation, as described in Section 2.1. We calculate the relative contribution of alleles in each effect size squared bin $(a_1^2, a_2^2]$ (between the gray gridlines) by dividing the contribution of fixations in the bin by the total mutational input per generation times the size of the bin, $2NU \int_{a_1}^{a_2} g(a)da$. The analytic predictions were based on the linear non-Lande (**Equation A3-83** or **Equation 28** in the main text) and nonlinear non-Lande (Section 5.4.3) approximations with the corresponding parameters.

## 5.3. The nonlinear Lande approximation

Our linear approximations become less accurate when directional selection causes large changes in allele frequencies, as expected when trait polygenicity is low, the shift is large (relative to the initial phenotypic standard deviation) or the magnitude of effect size is large (relative to $\delta$). Here, we develop a nonlinear Lande approximation that is more accurate than the linear one and we derive conditions under which the improvement is substantial.

Our derivation follows the same steps that we took in previous sections but without relying on linear expansions. First, we derive a nonlinear approximation for the effects of directional selection, once again assuming that they occur as an instantaneous pulse. To this end, we apply the same kind of nonlinear approximation that we used for the rapid phase (Section 4.1), but in this case we ignore the effects of stabilizing selection. By analogy to our derivation in Section 5.1, we express the change in frequency due to directional selection as

$$\Delta x_d \left( \pm a, x_0 \right) = x_d \left( \pm a, x_0 \right) - x_0 = \pm a x_0 \left( 1 - x_0 \right) \cdot F_d \left( \pm a, x_0 \right), \tag{A3-84}$$

where

$$
\begin{aligned}
F_d \left( a, x_0 \right) &\equiv \frac{1}{a} \left( \frac{\mathrm{Exp} \left[ a \cdot \int_0^\infty \left( D_L \left( \tau \right) / V_S \right) d\tau \right] - 1}{1 + x_0 \cdot \left( \mathrm{Exp} \left[ a \cdot \int_0^\infty \left( D_L \left( \tau \right) / V_S \right) d\tau \right] - 1 \right)} \right) \\
&= \frac{1}{a} \left( \frac{\mathrm{Exp} \left[ a \cdot \Lambda/V_A(0) \right] - 1}{1 + x_0 \cdot \left( \mathrm{Exp} \left[ a \cdot \Lambda/V_A(0) \right] - 1 \right)} \right)
\end{aligned}
\tag{A3-85}
$$

(here, $F_d$ is defined along the same lines as $F_t$ in Section 4.1).

Similar to our previous derivations, we approximate fixation probabilities using the frequencies after the instantaneous pulse as an initial condition, but in this case, we do not use a linear expansion of the fixation probability in $\Delta x_d$. In particular, we approximate the expected long-term (fixed) contribution to phenotypic change of a pair of opposite alleles with effect sizes $\pm a$ and initial MAF $x_0$ by

$$\Delta z_\infty^* \left(a, x_0\right) = 2a \cdot \Delta x_\infty^* \left(a, x_0\right) \approx 2a \left(x_\infty \left(a, x_0\right) - x_\infty \left(-a, x_0\right)\right) \tag{A3-86}$$

and the contribution of such pairs per unit mutational input by

$$\Delta z_\infty \left(a, x_0\right) \approx \Delta z_\infty^* \left(a, x_0\right) \cdot \rho(a, x_0). \tag{A3-87}$$

We can also write down integral forms for the marginal contribution of alleles with a given magnitude and the total contribution across all magnitudes, but in this case these integrals do not simplify in an obvious way.

We can now compare the results of the linear and nonlinear Lande approximations in order to ascertain when we should expect the linear approximation to be accurate. Notably, from *Equation A3-85*, we see that when $a \cdot \Lambda/V_A(0) \ll 1$ then $F_d \left(a, x_0\right) \approx \Lambda/V_A(0)$ and thus $x_d^{nL} \left(a, x_0\right) \approx x_d^{lL} \left(a, x_0\right)$, where superscripts $nL$ and $lL$ correspond to the nonlinear and linear Lande approximations, respectively. We therefore expect the linear approximation to be accurate when $a \cdot \Lambda/V_A(0) \ll 1$.

Our simulation results are consistent with this condition (*Appendix 3—figure 13*): when it is met, the linear and nonlinear approximations and simulations produce similar results. In cases with intermediate polygenicity and large shifts or low polygenicity and intermediate to large shifts, we see that the nonlinear approximation becomes substantially more accurate for intermediate effect size alleles (when $a \cdot \Lambda/V_A(0) \gtrsim 1/2$; shaded gold region in *Appendix 3—figure 13B and C*). When polygenicity is low and both the shift and effects sizes are large, the nonlinear approximation can substantially overestimate the contribution to phenotypic adaptation (see the red box in *Appendix 3—figure 15A*), because many large effect alleles do not segregate long enough to feel the full integral effect of directional selection. However, when Lande's approximation applies, we expect such large effect alleles to be scarce and contribute negligibly to polygenic adaptation.

## 5.4. The nonlinear non-Lande approximation

Lastly, we consider the case where both the linear expansions and Lande's approximation are inaccurate. We expect Lande's approximation to be inaccurate when alleles with large effect sizes contribute substantially to genetic variance before the shift, or more precisely when $C$ is not negligibly small (see Section 5.1). We expect the linear expansions to become inaccurate when polygenicity is low, the shift is large or the allele magnitude is large. More precisely, by analogy with our condition in the Lande case and as will become apparent below, we expect the linear approximation to become inaccurate when $(1 + A) \cdot a \cdot \Lambda/V_A(0)$ is not negligibly small. We derive an approximation for this case by combining the approaches that we used in the linear non-Lande and nonlinear Lande approximations. As the contribution of new mutations in the non-Lande case can be substantial, we consider both new mutations and standing variation.

### 5.4.1 Standing variation

The nonlinear non-Lande approximation for standing variation is similar to the nonlinear Lande approximation. The only difference is that here the integral effect of directional selection is $\int_0^{t_*} D\left(\tau\right) d\tau = (1 + A) \cdot \int_0^\infty D_L\left(\tau\right) d\tau = (1 + A) \cdot \Lambda \cdot V_S/V_A(0)$ rather than $\Lambda \cdot V_S/V_A(0)$. By analogy, the instantaneous pulse approximation for the effect of directional selection is

$$\Delta x_d \left(a, x_0\right) = a x_0 \left(1 - x_0\right) \cdot F_d(a, x_0), \tag{A3-88}$$

where

$$
\begin{aligned}
F_d \left(a, x_0\right) &\equiv \frac{1}{a} \left( \frac{\mathrm{Exp}\left[a \cdot \int_0^{t_*} \left(D\left(\tau\right)/V_S\right) d\tau\right] - 1}{1 + x_0 \left(\mathrm{Exp}\left[a \cdot \int_0^{t_*} \left(D\left(\tau\right)/V_S\right) d\tau\right] - 1\right)} \right) \\
&= \frac{1}{a} \left( \frac{\mathrm{Exp}\left[(1 + A) \cdot a \cdot \Lambda/V_A(0)\right] - 1}{1 + x_0 \left(\mathrm{Exp}\left[(1 + A) \cdot a \cdot \Lambda/V_A(0)\right] - 1\right)} \right).
\end{aligned}
\tag{A3-89}
$$

Based on *Equation A3-89* and the same reasoning that we applied to the Lande approximations, we conclude that when $(1 + A) \cdot a \cdot \Lambda/V_A(0) \ll 1$ then $\Delta x_d^{nN} \left(a, x_0\right) \approx \Delta x_d^{lN} \left(a, x_0\right)$ (where superscripts $nN$ and $lN$ correspond to the nonlinear and linear non-Lande approximations respectively), and we expect the linear non-Lande approximation to be accurate. The nonlinear non-Lande approximations

for other quantities take the same form as in the corresponding nonlinear Lande approximations (e.g., as in *Equations A3-86* and *A3-87*).

To test the nonlinear non-Lande approximation for standing variation, we first estimate the proportional contribution of standing variation to phenotypic change, $\eta$, using our *all alleles* simulations. We then solve for $A$ numerically by requiring that

$$2NU \cdot \Delta z_\infty^v \left(A\right) = 2NU \cdot \int \Delta z_\infty^v \left(a|A\right) \cdot g(a) da = \eta \cdot \Lambda, \tag{A3-90}$$

where $\Delta z_\infty^v \left(A\right)$ and $\Delta z_\infty^v \left(a|A\right)$ denote the nonlinear non-Lande approximations with a given value of $A$.

When we compare the nonlinear and linear non-Lande approximations for standing variation with simulation results (*Appendix 3—figures 16* and *18*) we find the non-linear approximation can provide better estimates of fixation probability when $a \cdot (1 + A) \cdot \Lambda / V_A(0) \gtrsim 1/2$ (golden shaded regions in *Appendix 3—figure 16*); this accords with our expectation given the condition for the accuracy of the linear approximation ($a \cdot (1 + A) \cdot \Lambda / V_A(0) \ll 1$). However, the linear and nonlinear non-Lande approximation perform comparably well for calculating the contribution to the change in mean phenotype (*Appendix 3—figure 18*), since the error in the linear approximation cancels out somewhat in calculating the *difference* in fixation probabilities of opposite alleles (which is what generates the change in mean). For large shifts and low polygenicity, the nonlinear approximation can even perform worse than the linear one, because it overestimates the proportion contribution coming from large effect alleles (see the red box in *Appendix 3—figure 18*); and, consequently, also underestimates the proportion coming from small and intermediate effect alleles. This is because the approximation does not account for the fact that large effect alleles tend to segregate for fewer generations than those with small and moderate effects, and therefore experience fewer effective generations of directional selection.

### 5.4.2 New mutations

We derive the nonlinear approximation for new mutations by combining the approach that we used for new mutations in the linear non-Lande case with the one we used for standing variation in the nonlinear non-Lande case. The nonlinear instantaneous pulse approximation for the effect of directional selection on the frequency of a new mutation with effect size $\pm a$ that arose $t$ generations after the shift in optimum is

$$\Delta x_d \left(\pm a|t\right) = \pm a \frac{1}{2N} \left(1 - \frac{1}{2N}\right) \cdot F_d(\pm a|t), \tag{A3-91}$$

where

$$\begin{aligned} F_d \left(\pm a|t\right) &\equiv \frac{1}{a} \left(\frac{\text{Exp}\left[a \cdot \int_t^{t+\Delta_t} \left(D\left(\tau\right)/V_S\right) d\tau\right] - 1}{1 + 1/(2N) \cdot \left(\text{Exp}\left[a \cdot \int_t^{t+\Delta_t} \left(D\left(\tau\right)/V_S\right) d\tau\right] - 1\right)}\right) \\ &\approx \frac{1}{a} \left(\text{Exp}\left[a \cdot \int_t^{t+\Delta_t} \left(D\left(\tau\right)/V_S\right) d\tau\right] - 1\right). \end{aligned} \tag{A3-92}$$

As we noted in Section 5.2.2, we expect that $a \cdot \Delta x_d(a|t) \ll 1$ for most if not all new mutations. In this case, the first order Taylor expansion of the fixation probability in $\Delta x_d \left(a|t\right)$ remains accurate and we can approximate the expected long-term contribution of a pair of opposite mutations with effect sizes $\pm a$ by

$$\begin{aligned} \Delta z_\infty^* \left(a|t\right) &= 2a \left(x_\infty \left(a|t\right) - x_\infty \left(-a|t\right)\right) \approx \frac{\partial \pi}{\partial x} \left(a, 1/(2N)\right) \cdot \Delta z_d^* \left(a|t\right) \\ &\approx \frac{2v^* \left(a, 1/(2N)\right)}{v \left(a, 1/(2N)\right)} \cdot 2f(a) \cdot \overline{F}_d \left(a|t\right) \\ &\approx 1/(2N) \cdot 2f\left(a\right) \cdot \overline{F}_d(a|t), \end{aligned} \tag{A3-93}$$

where

$$\overline{F}_d\left(a|t\right) \equiv \left(F_d\left(a|t\right) + F_d\left(-a|t\right)\right)/2 \; = \frac{1}{a}\mathrm{Sinh}\left[a \cdot \int_t^{t+\Delta_t}\left(D\left(\tau\right)/V_S\right)d\tau\right]. \tag{A3-94}$$

The expected contribution of such pairs per unit mutational input is then approximated by

$$\Delta z_\infty\left(a|t\right) \approx f\left(a\right)\left(\frac{1}{a}\mathrm{Sinh}\left[a \cdot \int_t^{t+\Delta_t}\left(D\left(\tau\right)/V_S\right)d\tau\right]/(2N)\right). \tag{A3-95}$$

We can also write down integral forms for the marginal contribution of alleles with a given magnitude and the total contribution across all magnitudes, but these integrals do not simplify in an obvious way.

We provided this approximation for completeness, but as we noted in Section 5.2.2, we expect the linear non-Lande approximation for new mutations to be accurate for considerably wider parameter ranges than it is for standing variation, possibly for the entire range permitted by our conditions (*Appendix 1—table 2*).

### 5.4.3 Standing variation and new mutations

We test the combination of the nonlinear non-Lande approximations for standing variation and linear non-Lande approximation for new mutations against simulation results, approximating their joint contribution by $\Delta z_\infty^v\left(a\right) + \Delta z_\infty^m\left(a\right)$ (*Equation A3-83*). As we illustrate in *Appendix 3—figure 19*, the linear and nonlinear non-Lande approximations perform similarly well in most cases. However, as discussed in Section 5.4.1, when polygenicity is low or the shift is large, the nonlinear approximation for standing variation ($\Delta z_\infty^v\left(a\right)$) can overestimate the contribution coming from large effect alleles. Since the joint contribution is the sum of the contribution from standing variation and new mutations, the nonlinear non-Lande approximation for the joint contribution also overestimates the contribution coming from large effect alleles (see the red box in *Appendix 3—figure 19*).

## 6. Phenotypic deviations from Lande's approximation are determined by $C$

Here, we revisit the phenotypic response to selection to articulate three conjectures that extend the analysis in the main text. The phenotypic response affects the allele dynamics primarily through the mean distance from the optimum, $D(t)$, as described in our allelic equation (*Equation A3-1*). The allele dynamics are insensitive to rapid stochastic fluctuations in $D(t)$, because selection effects on allele frequencies are small in a single generation, such that, in effect, alleles 'feel' the average $D(t)$ over several generations. We therefore focus on integrals of the form $\int_0^t D(\tau)/V_S d\tau$, which is how $D(t)$ features in our allelic approximations, and on the quasi-static approximation $D(t) \approx \mu_3(t)/(2V_A(t))$ for the distance during equilibration, both of which are less noisy than $D(t)$ itself (*Appendix 3—figure 22A and B*). We continue to measure the trait value in units of $\delta$ but sometimes include $\delta$ in equations for clarity.

While variation in each of the model parameters affects the trajectories of the mean distance (or its integral), a few compound parameters appear most influential (*Appendix 3—figure 20*). During the rapid phase, the distance is well approximated by Lande's solution $D_L(t)$, as illustrated for a couple of model parameter choices in *Figure 2B* and for a dozen in *Appendix 3—figure 20B* (where the integral over the distance scaled by the integral obtained under Lande's approximation stacks up around 1 during the rapid phase). During the equilibration phase, the distance from the optimum declines very slowly in non-Lande cases, resulting in a prolonged increase in the integral over the distance (*Appendix 3—figure 20A and B*). Nevertheless, for a given amplification $C$, the integral over the distance scaled by the integral obtained under Lande's approximation appears to be insensitive to the values of other parameters (*Appendix 3—figure 20B*).

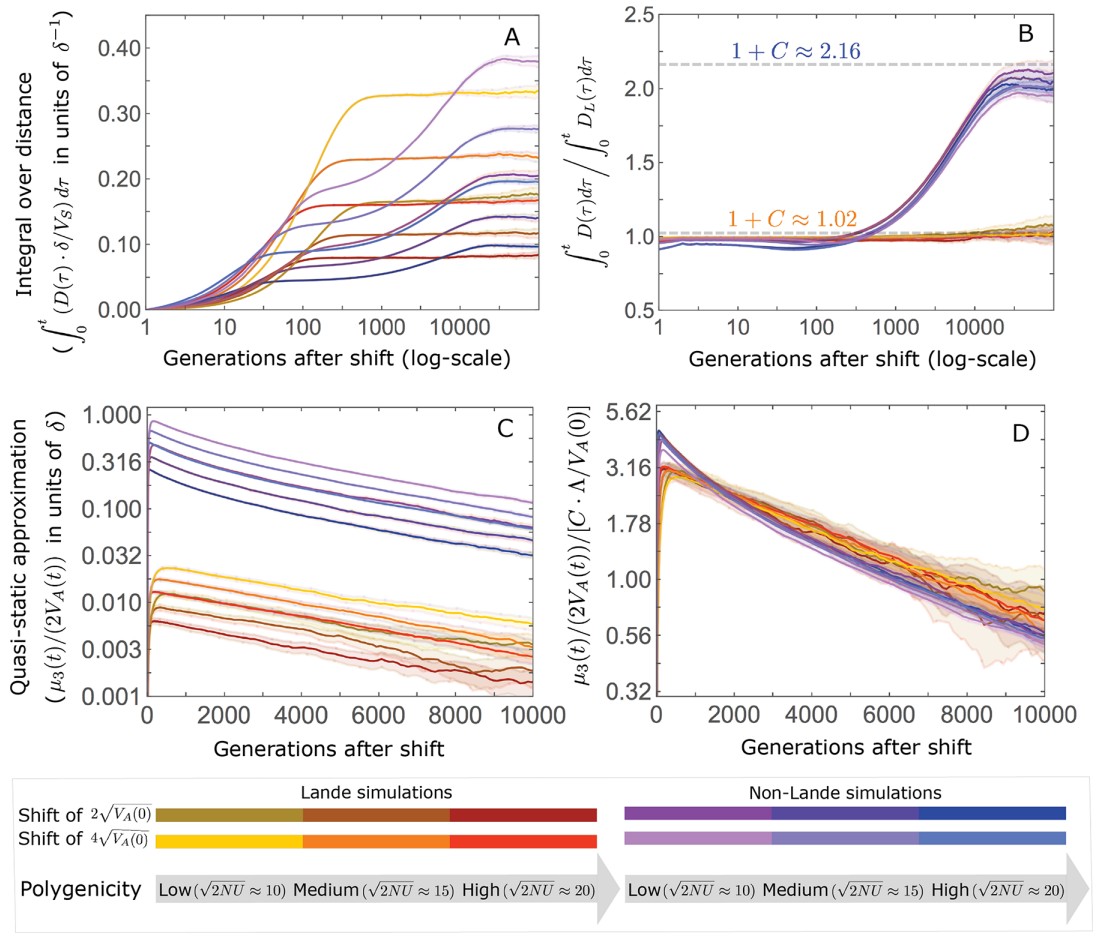

**Appendix 3—figure 20.** The trajectory of the mean integral over the phenotypic distance from the optimum is largely determined by Lande's approximation and by the amplification $C$. (**A**) The integral distance from the optimum as a function of time after the shift. In Lande cases, the integral nears the asymptote $\int_0^\infty \left( D_L(\tau) \cdot \delta / V_S \right) d\tau = (\Lambda \cdot \delta) / V_A(0)$ at the end of the rapid phase, whereas in non-Lande cases, the integral continues to increase substantially during equilibration. (**B**) The integral over the distance scaled by the integral over the distance obtained from Lande's approximation as a function of time. The phenotypic trajectories corresponding to the Lande and to the non-Lande cases with different parameter values stack up, illustrating that, to a good approximation, the impact of polygenicity and shift size are captured by Lande's approximation; the long-term impact of the effect size distribution, manifest in the non-Lande cases, is largely captured by $C$. (**C**) The quasi-static approximation of the distance as a function of time is affected by all of the parameters. (**D**) The quasi-static approximation of the distance scaled by $C \cdot \left( \Lambda \cdot \delta \right) / V_A(0)$ as a function of time. During the equilibration phase, the phenotypic trajectories corresponding to different parameter values stack up, illustrating that, to a good approximation, the effects of all of the parameters is captured by this compound parameter. The simulation results were generated using the *all allele* simulation, as described in Section 2.1.

The trajectories of the quasi-static approximation further support the notion that, after normalization based on Lande's approximation, $D(t)$ during equilibration depends primarily on $C$ (**Appendix 3—figure 20C and D**). In fact, simulations support our 1st conjecture that

$$D(t) \approx \mu_3(t)/V_A(t) \approx (k(t) \cdot \delta) \cdot C \cdot (\Lambda \cdot \delta)/V_A(0),$$ (A3-96)

where $(\Lambda \cdot \delta)/V_A(0) = \int_0^\infty \left( (D_L(\tau) \cdot \delta)/V_S \right) d\tau$ is the Lande-based normalization, and $k(t)$ is a proportionality coefficient that has no units. Simulations suggest that $k(t) \lesssim 5$ (see, e.g., **Appendix 3— figure 20D**), and our 2nd conjecture (**Equation A3-97** below) suggests that $\int_{t_1}^\infty k(\tau)d\tau \approx 2N$. One important implication of **Equation A3-96** is that at least under our assumptions on parameters, the distribution of effect sizes alone, as summarized by $C$, determines whether the phenotypic response deviates substantially from Lande's approximation. Specifically, under our conditions on parameters

$\Lambda/V_A(0) \lessgtr 1/2$ (**Equation A3-32**), and thus if $C \ll 1$ then even large shifts are insufficient to generate appreciable deviations from Lande's approximation during equilibration.

Our 2nd conjecture concerns the long-term integral distance $\int_0^\infty \left(D(\tau) \cdot \delta/V_S\right) d\tau$. We have seen that in the Lande case, when $C \ll 1$, then $\int_0^\infty \left(D(\tau) \cdot \delta/V_S\right) d\tau \approx \int_0^\infty \left(D_L(\tau) \cdot \delta/V_S\right) d\tau = (\Lambda \cdot \delta)/V_A(0)$. So far, we have not considered $\int_0^\infty \left(D(\tau) \cdot \delta/V_S\right) d\tau$ in the non-Lande case. (We explained our non-Lande allelic approximations in terms of $\int_0^{t_*} \left(D(\tau) \cdot \delta/V_S\right) d\tau$, where $t_*$ was the effective number of generations over which alleles are subject to directional selection; e.g., **Equation 23** in the main text.) Our simulations support our 2nd conjecture, that

$$\int_0^\infty \left(D(\tau) \cdot \delta/V_S\right) d\tau \approx (1 + C) \cdot \int_0^\infty \left(D_L(\tau) \cdot \delta/V_S\right) d\tau = (1 + C) \cdot (\Lambda \cdot \delta)/V_A(0) \qquad \text{(A3-97)}$$

(**Appendix 3—figures 20B** and **21B**). Assuming that $D(t)$ is well approximated by Lande's solution during the rapid phase, i.e., until $t_1$, and by **Equation A3-96** afterwards, we find that

$$\begin{aligned}
\int_0^\infty \left(D(\tau) \cdot \delta/V_S\right) d\tau &= \int_0^{t_1} \left(D_L(\tau) \cdot \delta/V_S\right) d\tau + \int_{t_1}^\infty (k(t) \cdot \delta) \cdot C \cdot (\Lambda \cdot \delta)/V_A(0) \cdot \delta/V_S d\tau \\
&= (\Lambda \cdot \delta)/V_A(0) \cdot (1 - \delta) + (\Lambda \cdot \delta)/V_A(0) \cdot \int_{t_1}^\infty (k(t) \cdot \delta) \cdot C \cdot \delta/V_S d\tau \quad \text{(A3-98)} \\
&\approx (\Lambda \cdot \delta)/V_A(0) \cdot \left(1 + C \cdot \int_{t_1}^\infty (k(t) \cdot \delta) \cdot \delta/V_S d\tau\right),
\end{aligned}$$

which given equation **Equation A3-97** suggests that $\int_{t_1}^\infty k(\tau) d\tau \approx V_S/\delta^2 = 2N$.

Our 3rd conjecture concerns the long-term phenotypic contribution of fixations of new mutations (i.e., that arise after the shift). The results of simulations suggest that the proportional contribution of new mutations, $1 - \eta$, depends primarily on the distribution of effect sizes, and specifically on $C$. In fact, it appears that under our assumptions on parameters, to a good approximation, the proportional contribution of new mutations.

$$1 - \eta \propto C, \qquad \text{(A3-99)}$$

and is thus insensitive to the values of other parameters, e.g., to the extent of polygenicity and shift size (**Appendix 3—figure 21B**). In summary, we posit that the parameter $C$, which depends only on the distribution of mutation magnitudes and arose from our analysis of the allelic response, largely determines the deviations of the phenotypic dynamics from Lande's approximation.

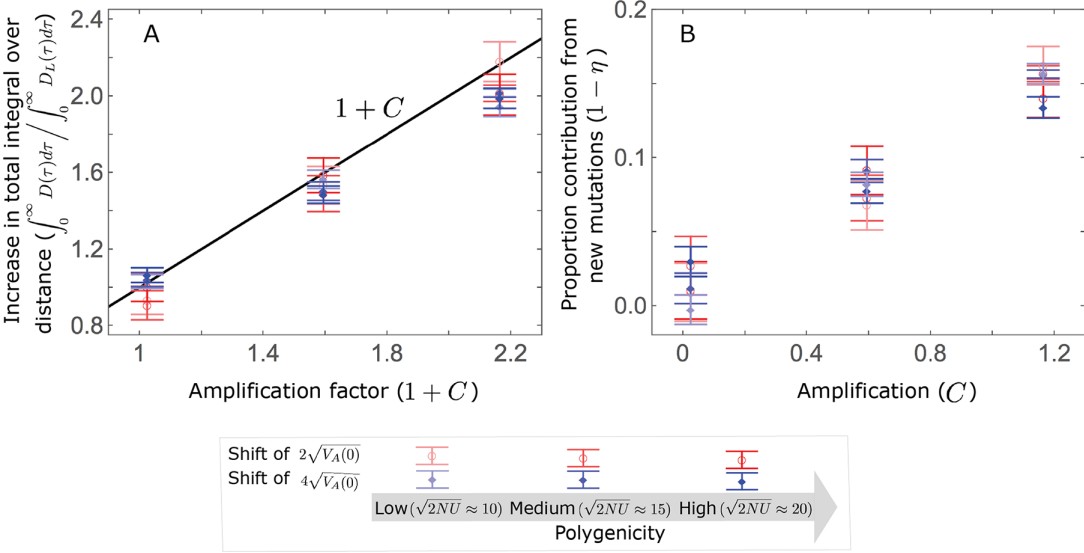

**Appendix 3—figure 21.** The amplification $C$ largely determines the increase in the total integral over distance, where $\int_0^\infty D(\tau)d\tau / \int_0^\infty D_L(\tau)/V_S d\tau \approx 1 + C$ (**A**); moreover, the proportion of long-term phenotypic change arising from new mutations $(1 - \eta)$, is approximately proportional to $C$ (**B**). As an illustration, we show the averages of these quantities over 500 *all alleles* simulations assuming $N = 10^4$, with the three extents of polygenicity and

*Appendix 3—figure 21 continued on next page*

*Appendix 3—figure 21 continued*
two shift sizes specified in the legend, and with three distributions of effect sizes: our two standard cases with an exponential distribution of effect sizes squared, with $E(a^2) = 1$ and $C \approx 0.02$ (our Lande case) and with $E(a^2) = 16$ and $C \approx 1.16$ (our non-Lande case), and an additional intermediate case with $a^2 \sim \text{Gamma}[2/9, 27]$ and $C \approx 1.16$.

## 7. Variation over realizations of the adaptive response

Throughout the manuscript we focused on the mean values of summaries of the phenotypic and allele dynamics, setting aside variation around the mean over different realizations of the adaptive response. In order to provide a sense of the magnitude and impact of this variation, we present simulation results for the main summaries we have considered, where in the allelic case we compare our results with a simple analytic approximation. Overall, these analyses do not reveal unexpected behaviors that alter our qualitative understanding of the adaptive response.

## 7.1. Variation in the phenotypic response

*Appendix 3—figure 22* shows the variation in our main summaries of the phenotypic distribution during the adaptive response. The variation in the distance from the optimum increases during the rapid phase, with the standard deviation (SD) exceeding its equilibrium value of $\delta$ (*Appendix 3—figure 22A*). This increase is greater for greater shifts and when allele magnitudes are greater (i.e., in the non-Lande case), plausibly because both factors increase the magnitude of perturbations to allele frequencies during the rapid phase. The variation then returns to the equilibrium level during the equilibration phase. Importantly, even during the rapid phase variation around the mean appears to be minor throughout the parameter range we consider.

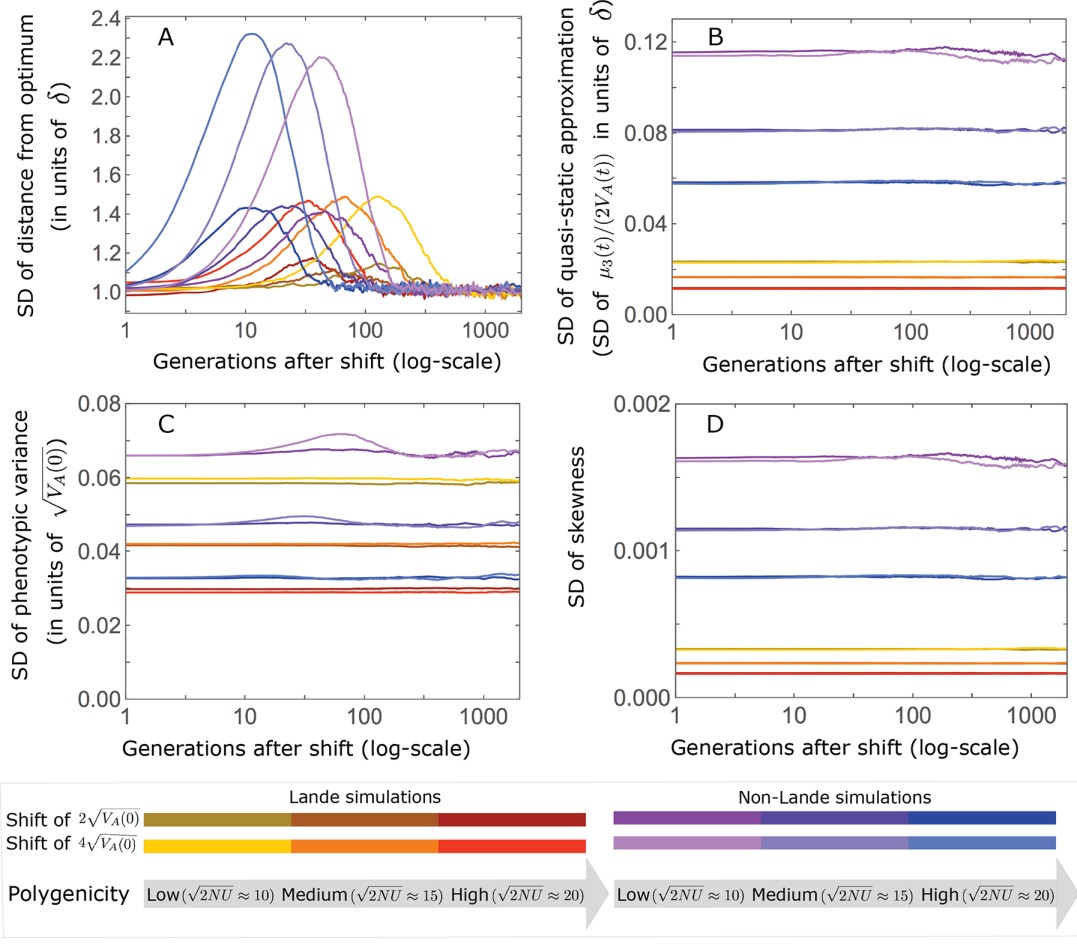

**Appendix 3—figure 22.** Variation in summaries of the phenotypic distribution as a function of time after the shift in optimum. (**A**) The standard deviation of the population's mean distance from the optimum. (**B**) The standard

*Appendix 3—figure 22 continued on next page*

*Appendix 3—figure 22 continued*

deviation of the quasi-static approximation for the population's mean phenotypic distance from the optimum, that is, the SD of $\mu_3(t)/(2V_A(t))$ (*Equation 6* in the main text). (**C**) The standard deviation of the phenotypic variance measured relative to the expected variance at equilibrium (*Equation A3-30*). (**D**) The standard deviation of the skewness of the phenotypic distribution (i.e., $\mu_3(t)/V_A^{3/2}(t)$). The simulation results were generated using the *all allele* simulation, as described in Section 2.1.

Variation in our quasi-static approximation for the distance from the optimum during the equilibration phase, i.e., in $\mu_3(t)/(2V_A(t))$, is stable throughout the adaptive response, and substantially smaller than the variation in the mean itself (*Appendix 3—figure 22B*); this observation lends justification for using the mean quasi-static approximation in the *single allele* simulations (see Section 2.1). Similarly, variation in both the phenotypic variance and skewness are stable throughout the adaptive response and similar to their magnitude at equilibrium (*Appendix 3—figure 22C and D*). Notably, the variation in phenotypic variance is negligibly small (*Appendix 3—figure 22C*), as we may have expected given that we are considering a highly polygenic trait. In the non-Lande cases in which the initial increase in skewness has a substantial impact on the dynamics, the variation in skewness is small relative to its peak values (e.g. in the case with the larger shift and medium polygenicity the peak in mean skewness is $\sim 0.01$ compared to a SD of $\sim 0.001$; *Figure 2C* and *Appendix 3—figure 22C*). Consequently, we do not expect this variation to have a substantial impact on the phenotypic dynamics.

## 7.2. Variation in the allelic response

Next, we consider variation in contributions to phenotypic adaptation that arise from sites with different magnitudes and initial MAFs. We focus on the contributions from standing variation at the time of the shift in optimum, setting aside contributions from new mutations (which per our analyses could be substantial only during the equilibration phase of the non-Lande case).

We model variation in allelic contributions from standing variation assuming that allele trajectories are independent of one another. Under this assumption (alongside our assumptions of infinite sites and that the number of newly arising mutations follows a Poisson distribution), we conjecture that the number of segregating sites before the shift is well approximated by a Poisson distribution (with expectation approximated by *Equation A3-26* with $k(a) = 1$), and that magnitudes and initial MAFs of segregating sites are sampled independently from a probability density function (which is proportional to $\rho(a,x) \cdot g(a)$ with $\rho(a,x)$ defined in *Equation A3-23*). Consequently, the number of segregating sites with any subset of magnitudes and initial MAFs also follows a Poisson distribution. This part of the model is akin to the 'Poisson Random Field' approximation (*Sawyer and Hartl, 1992*), and is supported by our simulations (specifically, estimates of the variance and mean of the number of segregating sites are approximately equal across bins of effect sizes and initial MAFs). We then assume that allele trajectories after the shift are also independent and follow the processes we have described for the rapid and equilibration phases. We derive analytic approximations along these lines below (Section 7.2.1).

These simple approximations do well at describing the variance of allelic contributions to polygenic adaptation (*Appendix 3—figures 23* and *24*). They suggest and simulations confirm that the variance of contributions during the rapid phase increases with the size of the shift in optimal phenotype, but is insensitive to the extent of polygenicity, or equivalently, to the mutational input (*Equation A3-111* and *Appendix 3—figure 23*). In contrast, the variance in contributions in the equilibration phase increases linearly with the mutational input, but is insensitive to the size of the shift (*Equation A3-119* and *Appendix 3—figure 24*). Some intuition for these dependencies is provided below, alongside deriving the analytic approximation.

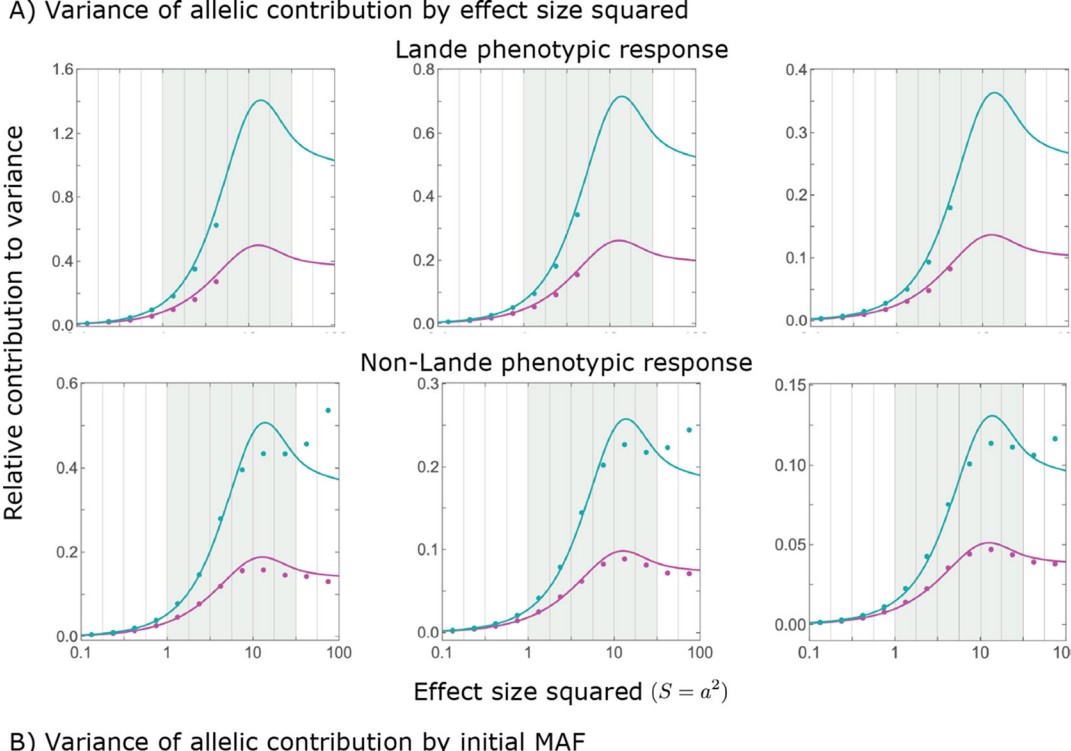

## A) Variance of allelic contribution by effect size squared

### Lande phenotypic response

### Non-Lande phenotypic response

Effect size squared $(S = a^2)$

## B) Variance of allelic contribution by initial MAF

### Lande phenotypic response

### Non-Lande phenotypic response

Initial minor allele frequency

Low $(\sqrt{2NU} \approx 10)$  Medium $(\sqrt{2NU} \approx 15)$  High $(\sqrt{2NU} \approx 20)$

Polygenicity

Shift of: $2\sqrt{V_A(0)}$  $4\sqrt{V_A(0)}$

Analytic approximation

Simulations

**Appendix 3—figure 23.** Variance of the allelic contribution to phenotypic adaptation in the rapid phase, as a function of effect size squared (**A**) and initial MAF (**B**). The results of our *all allele* simulation are compared to our corresponding analytic approximations (see text), with the parameters for each case detailed in Section 2.1. Simulation results are shown for bins (demarked by the gray vertical lines), where in (**A**) we divided the variance of the allelic contribution in bin $(a_1^2, a_2^2]$ by the corresponding expected mutational input $2NU \cdot \int_{a_1}^{a_2} g(a)da$, and in (**B**) we divided the variance in bin $(x_1, x_2]$ by $2NU \cdot (x_2 - x_1)$. In the Lande case, we only show the results for $a^2 < 10$, because fewer than $0.005\%$ of new mutations fall above that range.

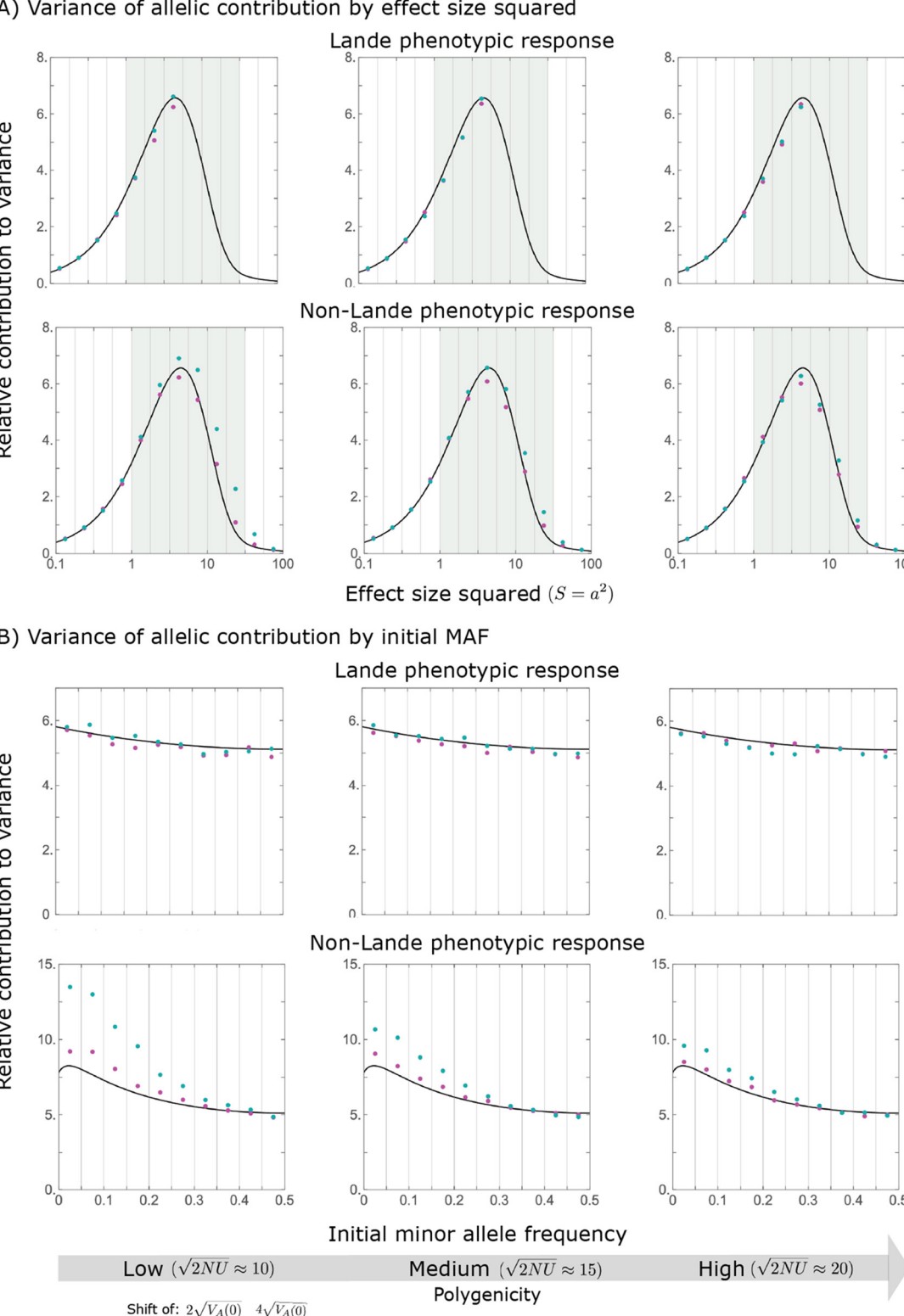

**Appendix 3—figure 24.** Variance of the allelic contribution to phenotypic adaptation in the equilibration phase, as a function of effect size squared (**A**) and initial MAF (**B**). The results of our *all allele* simulation are compared to our corresponding analytic approximations (see text), with the parameters for each case detailed in Section 2.1. Simulation results are shown for bins (demarked by the gray vertical lines), where in (**A**) we divided the variance of the allelic contribution in bin $(a_1^2, a_2^2]$ by the corresponding expected mutational input $2NU \cdot \int_{a_1}^{a_2} g(a)da$, and in (**B**) we divided the variance in bin $(x_1, x_2]$ by $2NU \cdot (x_2 - x_1)$. In the Lande case, we only show the results for $a^2 < 10$, because fewer than $0.005\%$ of new mutations fall above that range.

### 7.2.1 An independent alleles approximation

We model the variance of the contribution to change in mean phenotype from a bin, $b$, which corresponds to a subset of magnitudes and/or initial MAFs of segregating sites. We assume that the numbers of segregating sites in the bin with aligned or opposing minor alleles before the shift, $M_b^+$ and $M_b^-$ respectively, follow a Poisson distribution with the same mean, $E(M_b/2)$. We further assume that the contributions to phenotypic change of individual aligned or opposing alleles, $\Delta Z_b^+$ and $\Delta Z_b^-$ respectively, are independent and identically distributed. In these terms, the contribution of all the sites in the bin is

$$\Delta Z_b^T = \sum_{i=1}^{M_b^+} \Delta Z_b^{+,i} + \sum_{i=1}^{M_b^-} \Delta Z_b^{-,i};$$

(A3-100)

and the variance of this contribution is

$$
\begin{aligned}
V\left(\Delta Z_b^T\right) &= E(M_b/2) \cdot \left(V\left(\Delta Z_b^+\right) + V\left(\Delta Z_b^-\right)\right) + V(M_b/2) \cdot \left(E\left(\Delta Z_b^+\right)^2 + E\left(\Delta Z_b^-\right)^2\right) \\
&= E(M_b/2) \cdot \left(\left(V\left(\Delta Z_b^+\right) + V\left(\Delta Z_b^-\right)\right) + \left(E\left(\Delta Z_b^+\right)^2 + E\left(\Delta Z_b^-\right)^2\right)/2\right),
\end{aligned}
$$

(A3-101)

where the first line follows from the law of total variance, and $V(M_b) = E(M_b)$ because the number of sites in the bin follows a Poisson distribution. By the laws of total expectation and total variance

$$
\begin{aligned}
E\left(\Delta Z_b^\pm\right) &= E_b\left(E_d\left(\Delta Z^\pm | a, x_0\right)\right) \quad \text{and} \\
V\left(\Delta Z_b^\pm\right) &= V_b\left(E_d\left(\Delta Z^\pm | a, x_0\right)\right) + E_b\left(V_d\left(\Delta Z^\pm | a, x_0\right)\right),
\end{aligned}
$$

(A3-102)

where $\Delta Z^\pm | a, x_0$ is the contribution of a site with magnitude $a$ and initial MAF $x_0$, the subscript $d$ indicates that the expectation or variance is taken over the outcomes of genetic drift, and the subscript $b$ indicates that the expectation or variance is being taken over bin $b$. Substituting *Equation A3-102* into *Equation A3-101* yields

$$
\begin{aligned}
V\left(\Delta Z_b^T\right) = \quad & \underbrace{E(M_b) \cdot \left(E_b\left(E_d\left(\Delta Z^+ | a, x_0\right)\right)^2 + E_b\left(E_d\left(\Delta Z^- | a, x_0\right)\right)^2\right)/4}_{1^{\text{st}}\text{ term}} \\
& + \underbrace{E(M_b) \cdot \left(V_b\left(E_d\left(\Delta Z^+ | a, x_0\right)\right) + V_b\left(E_d\left(\Delta Z^- | a, x_0\right)\right)\right)/2}_{2^{\text{nd}}\text{ term}} \\
& + \underbrace{E(M_b) \cdot \left(E_b\left(V_d\left(\Delta Z^+ | a, x_0\right)\right) + E_b\left(V_d\left(\Delta Z^- | a, x_0\right)\right)\right)/2.}_{3^{\text{rd}}\text{ term}}
\end{aligned}
$$

(A3-103)

Each term in this equation reflects variance from a different source. The 1st is variance due to variation in the number of sites in the bin at the time of the shift. The 2nd is variance due to variation in effect sizes and initial MAFs of sites in the bin at the time of the shift. The 3rd reflects variance in the contribution of sites due to random genetic drift after the shift.

**The variance in the rapid phase.** We model the allelic contributions to phenotypic change in the rapid phase as follows. The contribution of a site with magnitude $a$ and initial MAF $x_0$ is

$$\Delta Z_{t_1}^\pm | a, x_0 = \pm 2a \cdot \Delta X_{t_1}^\pm | a, x_0,$$

(A3-104)

where the $\pm a$ correspond to aligned and opposing alleles, $t_1$ denotes the time at which the rapid phase ends, and $\Delta X_{t_1}^\pm | a, x_0$ denotes the change in frequency of the focal allele (the one that is initially minor) by time $t_1$. In approximating the contribution $\Delta Z_{t_1}^\pm | a, x_0$, we make the simplifying assumptions that its expectation is described by our *linear* approximation and that we can neglect the effects of stabilizing selection during the rapid phase. Under these assumptions, the expected contribution of a site equals half the contribution of a pair of sites with opposite minor alleles, $\Delta z_{t_1}^*(a, x_0)$, which in our linear approximation (*Equations A3-51* and *A3-54*) implies that

$$E_d\left(\Delta Z_{t_1}^\pm | a, x_0\right) = \Delta z_{t_1}^*(a, x_0)/2 \approx (\Lambda/V_A(0)) \cdot v^*(a, x_0),$$

(A3-105)

where, as before, $v^*(a, x_0) = 2a^2 x_0(1 - x_0)$ is the segregating site's initial contribution to phenotypic variance. We model the variation in a site's contribution using the Gaussian approximation for the effects of genetic drift on its frequency (*Cavalli-Sforza and Edwards, 1967*; *Nicholson et al., 2002*; *Coop et al., 2010*). Namely, given that the rapid phase is short relative to the time scale of random genetic drift, i.e., that $t_1 \equiv (V_S/V_A(0)) \cdot \mathrm{Ln}[\Lambda] \ll 2N$ (*Equation 9* in the main text), we approximate the variance in allele frequency by

$$V_d\left(\Delta X_{t_1}^\pm | a, x_0\right) \approx x_0(1 - x_0) \cdot t_1/(2N).$$

(A3-106)

In this approximation, the variance of a site's contribution to phenotypic change is

$$
\begin{aligned}
V_d\left(\Delta Z_{t_1}^\pm | a, x_0\right) &= V_d\left(\pm 2a \cdot \Delta X_{t_1}^\pm | a, x_0\right) \\
&= 4a^2 \cdot V_d\left(\Delta X_{t_1}^\pm | a, x_0\right) \\
&\approx 4a^2 x_0(1 - x_0) \cdot t_1/(2N) \cdot \\
&= 2 \cdot v^*(a, x_0)/V_A(0) \cdot \mathrm{Ln}[\Lambda],
\end{aligned}
$$

(A3-107)

where, given that we measure the trait in units of $\delta$, we have taken $V_S = 2N$ (see section on *Choice of units*) in the expression for $t_1$.

We rely on this model to calculate the three terms in our expression for the variance in allelic contribution in a bin $b$ (*Equation A3-103*). The 1st variance term, due to variation in the number of sites in the bin at the time of the shift, is

$$
\begin{aligned}
&E(M_b) \cdot \left(E_b\left(E_d\left(\Delta Z_{t_1}^+ | a, x_0\right)\right)^2 + E_b\left(E_d\left(\Delta Z_{t_1}^- | a, x_0\right)\right)^2\right)/4 \\
\approx\ & \left(\Lambda/V_A(0)\right)^2 \cdot E(M_b) \cdot E_b\left(v^*(a, x_0)\right)^2/2 \\
=\ & \left(E(M_b) \cdot E_b\left(v^*(a, x_0)/V_A(0)\right)\right) \cdot \left(\Lambda/\sqrt{V_A(0)}\right)^2 \cdot \left(E_b\left(v^*(a, x_0)\right)/2\right).
\end{aligned}
$$

(A3-108)

The 2nd variance term, due to variation in effect sizes and initial MAFs within the bin, is

$$
\begin{aligned}
&E(M_b) \cdot V_b\left(E_d\left(\Delta Z_{t_1}^+ | a, x_0\right) + E_d\left(\Delta Z_{t_1}^- | a, x_0\right)\right)/2 \\
\approx\ & E(M_b) \cdot \left(\Lambda/V_A(0)\right)^2 \cdot V_b\left(v^*(a, x_0)\right) \\
=\ & \left(E(M_b) \cdot E_b\left(v^*(a, x_0)/V_A(0)\right)\right) \cdot \left(\Lambda/\sqrt{V_A(0)}\right)^2 \\
& \cdot \left(E_b\left((v^*(a, x_0))^2\right)/E_b\left(v^*(a, x_0)\right) - E_b\left(v^*(a, x_0)\right)\right).
\end{aligned}
$$

(A3-109)

The 3rd variance term, due to random genetic drift, is

$$
\begin{aligned}
&E(M_b) \cdot E_b\left(\left(V_d\left(\Delta Z_{t_1}^+ | a, x_0\right) + V_d\left(\Delta Z_{t_1}^- | a, x_0\right)\right)/2\right) \\
\approx\ & 2 \cdot E(M_b) \cdot E_b\left(v^*(a, x_0)/V_A(0)\right) \cdot \mathrm{Ln}[\Lambda] \\
=\ & \left(E(M_b) \cdot E_b\left(v^*(a, x_0)/V_A(0)\right)\right) \cdot 2 \cdot \left(\mathrm{Ln}[\Lambda/\sqrt{V_A(0)}] - \mathrm{Ln}[\sqrt{V_A(0)}]\right).
\end{aligned}
$$

(A3-110)

Taken together, our approximation for the variance in allelic contributions is

$$
\begin{aligned}
V\left(\Delta Z_{b,t_1}^T\right) \approx\ & \left(E(M_b) \cdot E_b\left(v^*(a, x_0)\right)/V_A(0)\right) \\
& \cdot \left(\left(\Lambda/\sqrt{V_A(0)}\right)^2 \cdot Q_b + 2 \cdot \mathrm{Ln}[\Lambda/\sqrt{V_A(0)}] - 2 \cdot \mathrm{Ln}[\sqrt{V_A(0)}]\right),
\end{aligned}
$$

(A3-111)

where

$$Q_b \equiv E_b\left((v^*(a, x_0))^2\right)/E_b\left(v^*(a, x_0)\right) - E_b\left(v^*(a, x_0)\right)/2.$$

(A3-112)

The first part of the product, $E(M_b) \cdot E_b\left(v^*(a, x_0)\right)/V_A(0)$, is the expected initial proportion of phenotypic variance due to sites in the bin, which is a property of the equilibrium architecture that is independent of polygenicity. The first two terms in the second part of the product depend on the shift size measured relative to the initial phenotypic variance, $\Lambda/\sqrt{V_A(0)}$, and on $Q_b$, which is also a summary of the equilibrium architecture that is independent of polygenicity. The only explicit

dependence on polygenicity, in the last term, is logarithmic. Hence the insensitivity of the variance to the extent of polygenicity.

Some intuition for this insensitivity can be gleaned from considering the variance introduced by random genetic drift (the 3$^{rd}$ term in *Equation A3-103*). Doubling the expected number of segregating sites before the shift while keeping everything else equal (including $\Lambda/\sqrt{V_A(0)}$) halves the duration of the rapid phase, $t_1$, (by doubling $V_A(0)$) and thus the variance in frequency of individual alleles (*Equation A3-106*). Doubling the number of segregating sites, however, also doubles the number of sites contributing to phenotypic change, where the two effects cancel each other out. Coming up with simple intuitions for the other two variance terms (due to variation in the number of sites, and in their effect sizes and initial MAFs) is more challenging.

**The variance in the equilibration phase.** We model the variance of allelic contributions to phenotypic change during the equilibration phase assuming mutation-selection-drift equilibrium, thus neglecting the effect of the shift in optimum. The variance of allelic contributions during equilibration arises from the stochasticity of fixations. Our findings suggest that directional selection due to the shift in optimum typically introduces only small differences between the fixation probabilities of aligned and opposing alleles (*Figure 7* and Section 5). These small differences are important because they result in a nonzero expectation of fixed allelic contributions, which underlies long-term phenotypic adaptation. However, under most conditions the differences in fixation probabilities have a negligible effect on the variance of fixed contributions, which could be sizable at equilibrium. Exceptions might occur when the changes to fixation probabilities introduced by the shift in optimum are large relative to the fixation probabilities without the shift (at equilibrium). Specifically, the fixation probabilities of alleles with low initial frequencies exhibit marked changes in the non-Lande case with large shifts, due to prolonged, weak directional selection during the equilibration phase (*Appendix 3—figure 17*); this potentially explains the deviation from our approximation in the corresponding cases in *Appendix 3—figure 24B*. Setting these cases aside, we expect a model that neglects the effects of the shift to do well at approximating the variance of fixed allelic contributions.

We consider the variance that arises from sites that are segregating at the time of the shift in optimum. Neglecting the effects of directional selection, the fixed contribution of an allele with magnitude $a$ and initial MAF $x_0$ is a Bernoulli random variable

$$Z_\infty^{\pm,eq}|a, x_0 = \begin{cases} \pm 2a\left(1 - x_0\right) & \text{with probability } \pi(a, x_0) \\ \mp 2ax_0 & \text{with probability } 1 - \pi(a, x_0) \end{cases}, \tag{A3-113}$$

where the ± corresponds to aligned and opposing minor alleles, and $\pi(a, x_0)$ is the fixation probability at equilibrium, which is equal for aligned and opposing alleles. The expected contribution of a site is therefore

$$E_d\left(Z_\infty^{\pm,eq}|a, x_0\right) = \mp 2a\left(x_0 - \pi(a, x_0)\right) \tag{A3-114}$$

and the variance of this contribution is

$$V_d\left(\Delta Z_\infty^{\pm,eq}|a, x_0\right) = 4a^2 \cdot \pi\left(a, x_0\right) \cdot \left(1 - \pi\left(a, x_0\right)\right). \tag{A3-115}$$

We rely on this model to calculate the three terms in our expression for the variance in allelic contribution in a bin $b$ (*Equation A3-103*). The 1$^{st}$ variance term, due to variation in the number of sites in the bin at the time of the shift, is

$$
\begin{aligned}
& E\left(M_b\right) \cdot \left(E_b\left(E_d\left(\Delta Z_\infty^{+,eq}|a, x_0\right)\right)^2 + E_b\left(E_d\left(\Delta Z_\infty^{-,eq}|a, x_0\right)\right)^2\right)/4 \\
= & E\left(M_b\right)\left(E_b\left(2a\left(x_0 - \pi(a, x_0)\right)\right)^2 + E_b\left(-2a\left(x_0 - \pi(a, x_0)\right)\right)^2\right)/4 \\
= & E\left(M_b\right) \cdot E_b\left(2a\left(x_0 - \pi(a, x_0)\right)\right)^2/2.
\end{aligned} \tag{A3-116}
$$

The 2$^{nd}$ variance term, due to variation in effect sizes and initial MAFs within the bin, is

$$
\begin{aligned}
& E\left(M_b\right)\left(V_b\left(E_d\left(\Delta Z_\infty^{+,eq}|a, x_0\right)\right) + V_b\left(E_d\left(\Delta Z_\infty^{-,eq}|a, x_0\right)\right)\right)/2 \\
=\ & E\left(M_b\right) \cdot \left(V_b\left(-2a\left(x_0 - \pi(a, x_0)\right)\right) + V_b\left(2a\left(x_0 - \pi(a, x_0)\right)\right)\right)/2 \\
=\ & E\left(M_b\right) \cdot V_b\left(2a\left(x_0 - \pi(a, x_0)\right)\right).
\end{aligned}
\tag{A3-117}
$$

The 3$^{rd}$ variance term, due to random genetic drift, is

$$
\begin{aligned}
& E\left(M_b\right) \cdot E_b\left(\left(V_d\left(\Delta Z_\infty^{+,eq}|a, x_0\right) + V_d\left(\Delta Z_\infty^{-,eq}|a, x_0\right)\right)/2\right) \\
=\ & E\left(M_b\right) \cdot E_b\left(4a^2 \cdot \pi\left(a, x_0\right) \cdot \left(1 - \pi\left(a, x_0\right)\right)\right).
\end{aligned}
\tag{A3-118}
$$

Taken together, our approximation for the variance in the fixed allelic contributions is

$$
\begin{aligned}
V\left(\Delta Z_{b,\infty}^T\right) =\ & E\left(M_b\right) \cdot E_b\left(2a\left(x_0 - \pi(a, x_0)\right)\right)^2/2 + E\left(M_b\right) \cdot V_b\left(2a\left(x_0 - \pi(a, x_0)\right)\right) \\
& + E\left(M_b\right) \cdot E_b\left(4a^2 \cdot \pi\left(a, x_0\right) \cdot \left(1 - \pi\left(a, x_0\right)\right)\right) \\
=\ & E\left(M_b\right) \cdot \left(E_b\left(4a^2\left(x_0^2 + \pi(a, x_0) \cdot (1 - 2x_0)\right)\right) - E_b\left(2a\left(x_0 - \pi(a, x_0)\right)\right)^2/2\right).
\end{aligned}
\tag{A3-119}
$$

$E\left(M_b\right)$ is the expected number of segregating sites in the bin, which scales linearly with the mutational input, whereas the expression in parenthesis is a summary of the equilibrium architecture that is independent of the mutational input. Thus, we find that variance of the fixed allelic contribution is approximately linear in the mutational input. This makes intuitive sense: if the variance is dominated by the stochasticity of fixation at equilibrium, then it should scale linearly with the number of sites that are segregating before the shift, and hence with the mutational input.

**Variances corresponding to magnitude and initial MAF.** Rather than using some finite, arbitrary bin sizes, we consider how the density of variance depends on the magnitude $a$ or inital MAF $x_0$. First, we consider the variance of contributions in the rapid phase. Based on *Equations A3-111* and *A3-112*, we find that the density of variance for sites with magnitude $a$ is

$$
V\left(\Delta Z_{a,t_1}^T\right) = 2NU \cdot g(a) \cdot \nu_{t_1}(a),
\tag{A3-120}
$$

where

$$
\nu_{t_1}(a) = v(a)/V_A(0) \cdot \left(2 \cdot \mathrm{Ln}\left[\Lambda\right] + \left(\Lambda/\sqrt{V_A(0)}\right)^2 \cdot Q_a\right)
\tag{A3-121}
$$

is the density of variance per unit mutational input from sites with magnitude $a$, which is the analytic approximation shown in *Appendix 3—figure 23A*, $v(a)$ is the equilibrium density of genetic variance per unit mutational input from sites with magnitude $a$ (approximated in *Equation A3-29*),

$$
\begin{aligned}
Q_a &\equiv E_a\left(\left(v^*(a, x_0)\right)^2\right)/E_a\left(v^*(a, x_0)\right) - E_a\left(v^*(a, x_0)\right)/2 \\
&= (1 + a^2/2) - 2a^2/v(a) - v(a)/(2\rho(a)),
\end{aligned}
\tag{A3-122}
$$

and $\rho(a) \equiv 2 \cdot \int_0^{1/2} \rho(a, x)dx$ is the equilibrium density of segregating sites with magnitude $a$ per unit mutational input. Calculating the density of variance for sites with initial MAF $x_0$ requires us to integrate over the distribution of magnitudes for newly arising mutations. To this end, we assume an exponential distribution of effect sizes squared, with expectation $E(a^2)$. The corresponding density of variance for sites with initial MAF is

$$
V\left(\Delta Z_{x_0,t_1}^T\right) = 2NU \cdot \nu_{t_1}^M(x_0),
\tag{A3-123}
$$

where

$$
\nu_{t_1}^M(x_0) \approx v^M(x_0)/V_A(0) \cdot \left(2 \cdot \mathrm{Ln}\left[\Lambda\right] + \left(\Lambda/\sqrt{V_A(0)}\right)^2 \cdot Q_{x_0}\right)
\tag{A3-124}
$$

is the density of variance per unit mutational input from sites with initial MAF $x_0$ (hence the superscript $M$), which is the analytic approximation shown in *Appendix 3—figure 23B*,

$$v^M(x) \equiv \int_0^\infty v^*(a,x) \cdot 2\rho(a,x)g(a)da$$

$$= \begin{cases} (2Nx) \cdot 4 \cdot E\left(a^2\right) / \left(E\left(a^2\right) + x(1-x)\right)^2 & 0 \leq x \leq 1/(2N) \\ 4 \cdot E\left(a^2\right) / \left(E\left(a^2\right) + x(1-x)\right)^2 & 1/(2N) < x \leq 1/2 \end{cases} \tag{A3-125}$$

is the equilibrium density of genetic variance per unit mutational input from sites with MAF $x$,

$$\begin{aligned} Q_{x_0} &\equiv E_{x_0}\left(\left(v^*(a,x_0)\right)^2\right) / E_{x_0}\left(v^*(a,x_0)\right) - E_{x_0}\left(v^*(a,x_0)\right)/2 \\ &= 4\left(1 - 1/(1 + E\left(a^2\right) \cdot x_0(1-x_0))\right) - v^M(x_0)/(2\rho^M(x_0)), \end{aligned} \tag{A3-126}$$

and

$$\rho^M(x) \equiv 2 \cdot \int_0^\infty \rho(a,x)g(a)da$$

$$= \begin{cases} (2Nx) \cdot 2/\left(1 + E\left(a^2\right) \cdot x(1-x)\right) \cdot 1/(x(1-x)) & 0 \leq x \leq 1/(2N) \\ 2E\left(a^2\right) / \left(E\left(a^2\right) + x(1-x)\right) \cdot 1/(x(1-x)) & 1/(2N) < x \leq 1/2 \end{cases} \tag{A3-127}$$

is the equilibrium density of segregating sites with MAF $x$ per unit mutational input.

Next, we consider the variance of contributions after equilibration. Based on *Equation A3-119*, we find that the density of variance for sites with magnitude $a$ is

$$V\left(\Delta Z_{a,\infty}^T\right) \approx 2NU \cdot g(a) \cdot \nu_\infty(a), \tag{A3-128}$$

where

$$\begin{aligned} \nu_\infty(a) \approx & \int_0^{1/2} 4a^2\left(x^2 + \pi(a,x) \cdot (1-2x)\right) \cdot 2\rho(a,x)dx \\ & - \left(\int_0^{1/2} 2a\left(x - \pi(a,x)\right) \cdot 2\rho(a,x)dx\right)^2 / (2\rho(a)) \end{aligned} \tag{A3-129}$$

is the density of variance per unit mutational input from sites with magnitude $a$, which is the analytic approximation shown in *Appendix 3—figure 24A*. Similarly, assuming an exponential distribution of effect sizes squared, we find that the density of variance for sites with initial MAF $x_0$ is

$$V\left(\Delta Z_{x_0,\infty}^T\right) \approx 2NU \cdot \nu_\infty^M(x_0), \tag{A3-130}$$

where

$$\begin{aligned} \nu_\infty^M(x_0) \approx & \int_0^\infty 4a^2\left(x_0^2 + \pi(a,x_0) \cdot (1-2x_0)\right) \cdot 2\rho(a,x_0)g(a)da \\ & - \left(\int_0^\infty 2a\left(x_0 - \pi(a,x_0)\right) \cdot 2\rho(a,x_0)g(a)da\right)^2 / (2\rho^M(x_0)) \end{aligned} \tag{A3-131}$$

is the density of variance per unit mutational input from sites with initial MAF $x_0$, which is the analytic approximation shown in *Appendix 3—figure 24B*.

**Interpreting the density of variance per unit mutational input.** When we considered the expected allelic contributions to phenotypic adaptation, we expressed our results in terms of the "relative contributions" of sites of a given magnitude and/or initial MAF, which are densities per unit mutational input. The contribution from any subset of alleles, given any choice of model parameters, can be expressed as a simple integral over these relative contributions. For example, given the relative contribution from sites with magnitude $a$ in the rapid phase, $\Delta z_{t_1}(a)$ (*Equation 13* in the main text), the total contribution of sites with magnitudes $a \in (a_1, a_2)$ is $2NU \cdot \int_{a_1}^{a_2} \Delta z_{t_1}(a) g(a) da$.

Unfortunately, this property does not apply to the relative contributions to variance. Taking analogous integrals of the relative contribution to variance over a given range of

magnitudes and/or effect sizes would underestimate the variance in contributions arising from this range, because it would fail to account for the variation in magnitudes and initial MAFs of segregating sites in this range. This can be seen, for example, by calculating the variance associated with a bin of effect sizes, $(a_1, a_2)$, during the rapid phase based on *Equation A3-111*: specifically, there is no simple relationship between $Q_{(a_1,a_2)}$ and $Q_a$ (*Equation A3-122*), and $V\left(\Delta Z^T_{(a_1,a_2),t_1}\right) > 2NU \cdot \int_{a_1}^{a_2} \nu_{t_1}(a)\, g(a)\, da$. The same is true for the equilibration phase and for other choices of ranges. Consequently, in order to calculate the variance associated with any finite range of magnitudes and/or initial MAFs one must return to *Equations A3-111* and *A3-119*. Nonetheless, the relative contributions to variance shown in *Appendix 3—figures 23* and *24* provide a sense of the dependance of the variance on magnitudes and initial MAFs, and there is no other obvious better choice to do so.

## 8. The infinitesimal limit and Lande's approximation

In the section on the phenotypic response, we note that Lande's approximation corresponds to the infinitesimal limit (*Turelli, 2017*). Here we elaborate on this assertion by deriving the allelic and phenotypic equations in the infinitesimal limit, and by illustrating how to obtain this limit in our model.

To this end, we assume that in our model the phenotypic distribution becomes Normal in the infinitesimal limit—in which the number of segregating loci affecting a trait goes to infinity while their effects goes to zero such that genetic variance in the trait remains finite. This is a subtle point. Fisher argued that the central limit theorem implies that the phenotypic distribution should always be Normal in the infinitesimal limit (assuming that environmental contributions are normally distributed; *Fisher, 1918*). This argument is not always valid, however: even with free recombination, natural selection can introduce systematic associations between alleles at different loci, which could violate the assumption of independence in the central limit theorem (*Turelli, 2017*). Nonetheless, *Turelli and Barton, 1990* argue (and rigorously showed in the absence of drift) that in the special case of a Gaussian fitness function, the dependence among loci disappears in the infinitesimal limit, and thus, that the central limit theorem applies. Given that we assume a Gaussian fitness function we will therefore assume that the phenotypic distribution becomes Normal in the infinitesimal limit.

### 8.1. The phenotypic equation in the infinitesimal limit

We derive the phenotypic equation following the same steps as in Section 1. We begin with the allelic equation. Assuming that the phenotypic distribution is Normal allows us to calculate integrals over the phenotypic background of a focal site explicitly, without recourse to a Taylor expansion of the fitness function around the mean phenotype or the requirement that $V_A \ll V_S$. Specifically, *Equation A3-7* for the average fitness of individuals with $\iota$ copies of the allele of interest at the focal site becomes

$$\bar{W}_i = \lambda \cdot W_\lambda(\mu + a \cdot (i - 2x)), \tag{A3-132}$$

where $W_\lambda(z) \equiv \text{Exp}\left[-\lambda^2 z^2/(2V_S)\right]$ is the Gaussian fitness function, but with width $\sqrt{V_S}/\lambda$, and $\lambda \equiv \sqrt{V_S/(V_S + V_A)}$. Following the same reasoning as in Section 1.1 we find that the allelic equation is

$$E\left(\Delta x\right) \approx \lambda^2 \cdot (a \cdot D(t)/V_S) \cdot x(1-x) - \lambda^2 \cdot \left(a^2/V_S\right) \cdot (1 - D^2(t)/V_S) \cdot x(1-x)(1/2 - x). \tag{A3-133}$$

When $V_A \ll V_S$, $\lambda \approx 1$ and *Equation A3-133* reduces to *Equation A3-1* as expected.

Next, we derive the phenotypic equation from the allelic one, using the same reasoning as in Section 1.2. (Note that Lande's derivation of this approximation did not rely on an explicit model of the allele dynamics; *Lande, 1976*.) Given that the 3rd central moment of phenotypic distribution is zero (because it is Normal), we find that the phenotypic equation is

$$E\left(\Delta D\right) = -\lambda^2 \cdot \left(V_A/V_S\right) \cdot D = -\left(V_A/(V_A + V_S)\right) \cdot D, \tag{A3-134}$$

which is equivalent to equation (14) in *Lande, 1976* and reduces to *Equation 4* in the main text when $V_A \ll V_S$.

## 8.2. The infinitesimal limit in our model

Next, we consider the infinitesimal limit in our model. For mathematical convenience, we assume that the effect sizes squared of mutations are exponentially distributed with mean $E(a^2)$ (measuring the trait in units of $\delta$). We take the infinitesimal limit by taking $E(a^2) \to 0$ while holding the mutational variance (i.e. the expected input of variance per generation due to mutation) $V_M = 2U \cdot E(a^2)$ constant. As this limit is approached, the phenotype distribution approaches a Normal distribution and the expected, per generation change in allele frequency is well approximated by **Equation A3-133**. At equilibrium, the expected change in frequency of an allele with magnitude $a$ and frequency $x$ is therefore:

$$E\left(\Delta x\right) \approx -(\lambda \cdot a)^2/V_S \cdot x(1-x)(1/2 - x) \text{ with } \lambda \equiv \sqrt{V_S/(V_A + V_S)}. \tag{A3-135}$$

This equation is analogous to **Equation A3-13** with allele magnitude $\lambda \cdot a$ instead of $a$, where this substitution reflects the reduction in the efficacy of stabilizing selection due to the phenotypic variance (an equivalent substitution would be to replace $V_S$ with $V_S/\lambda^2$).

The analogy between **Equations A3-13** and **A3-135** allows us to translate the results of Section 3.1 and Section 3.2 to the case of the infinitesimal limit. Specifically, the equilibrium density of sites segregating with magnitude $a$ and MAF $x$ per unit mutational input is $2\rho\left(\lambda \cdot a, x\right)$ (**Equation A3-23**), the equilibrium density of variance is $v^*(a,x) \cdot 2\rho\left(\lambda \cdot a, x\right) = v(\lambda \cdot a, x)/\lambda^2$ (c.f. **Equation A3-28**), and the marginal density of sites with a given magnitude is therefore $v(\lambda \cdot a)/\lambda^2$ (see **Equation A3-29**). Consequently, the equilibrium phenotypic variance is

$$V_A = 2NU \int_0^\infty v\left(\lambda \cdot a\right) g(a) da = 2NU \cdot 2 \cdot E\left(a^2\right) \cdot H\left(\lambda \cdot E\left(a^2\right)\right), \tag{A3-136}$$

where

$$H(y) \equiv \frac{2 \cdot \left(y + 4 \cdot \sqrt{y/(4+y)} \cdot \text{arcsch}\left(2/\sqrt{y}\right)\right)}{y(4+y)}, \tag{A3-137}$$

and arcsch is the inverse hyperbolic cosecant function. (The closed form of the integral in **Equation A3-136** corresponds to the exponential distribution of effect sizes squared.) In the infinitesimal limit, where $E(a^2) \to 0$ while $V_M = 2U \cdot E(a^2)$ is held constant, the equilibrium phenotypic variance becomes:

$$\begin{aligned} V_A^{\text{inf}} &= \lim_{E(a^2) \to 0} 2N \cdot \left(V_M/\left(2E\left(a^2\right)\right)\right) \cdot 2 \cdot E\left(a^2\right) \cdot H\left(\lambda \cdot E\left(a^2\right)\right) = 2NV_M \cdot \lim_{E(a^2) \to 0} H\left(\lambda \cdot E\left(a^2\right)\right) \\ &= 2NV_M. \end{aligned} \tag{A3-138}$$

As expected, this coincides with the expected genetic variance under mutation-drift equilibrium (**Chakraborty and Nei, 1982**).

We note that the infinitesimal limit violates two of our model assumptions, but examining the rationale for these assumptions indicates that one of them can be relaxed and the other can be modified in a straightforward way. The first assumption that is violated is that the number of mutations per gamete per generation $U \leq 0.02$. This assumption guarantees that $\sqrt{V_A} \ll \sqrt{V_S}$, which was vital in the derivation of **Equation A3-1** for the expected per generation change in allele frequency. In the infinitesimal limit, we derived **Equation A3-133** in its stead, where to this end we assumed that the phenotype distribution is Normal but did not require the phenotypic variance to be small relative to the width of the fitness function. The second assumption that is violated is that a substantial proportion of incoming mutations are subject to effective selection at equilibrium, i.e., with $a^2 \gtrsim 1$. This assumption guarantees that the phenotypic variance is far greater than the typical fluctuations in mean phenotype at equilibrium, i.e., that $\sqrt{V_A} \gg \delta$. For this to hold in the infinitesimal limit we could require that $\sqrt{V_A^{\text{inf}}} = \sqrt{2NV_M} \gg \delta$.

## 9. Additional figures

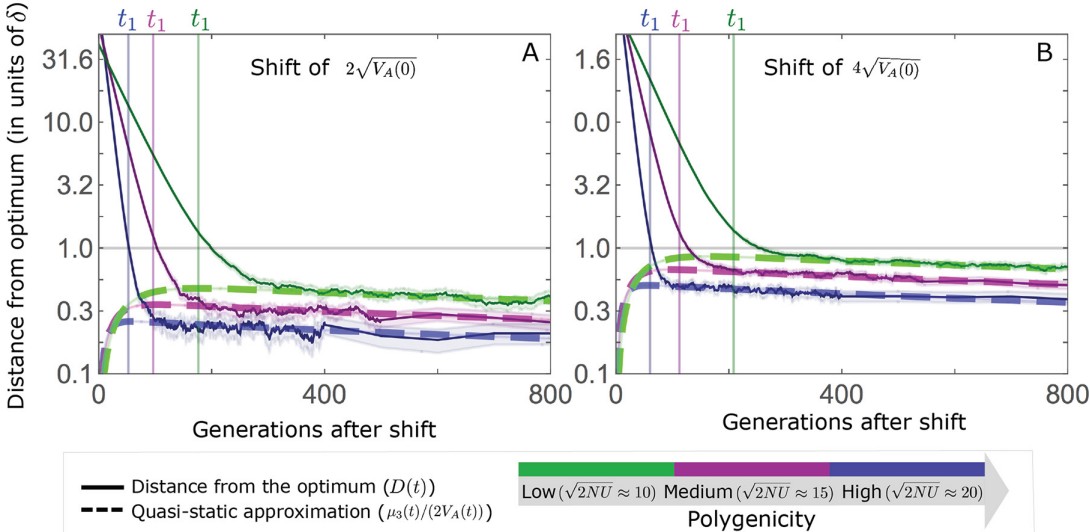

**Appendix 3—figure 25.** Our definition of the end of the rapid phase, $t_1$, roughly captures the change in phenotypic dynamic in the non-Lande case. At least in these non-Lande cases, $D_L(t_1)$ is quite close to $\delta$ and the quasi-static approximation (*Equation 6* in the main text) becomes accurate shortly after $t_1$. The simulation results were generated using the *all allele* simulation, as described in Section 2.1.

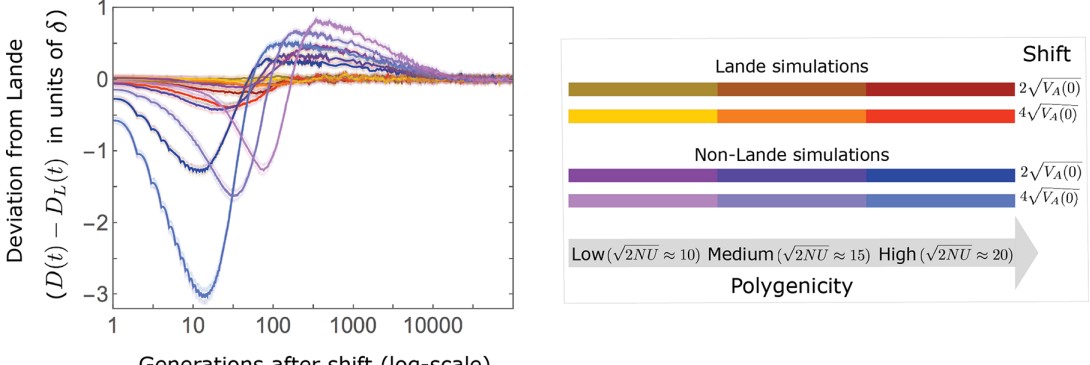

**Appendix 3—figure 26.** Deviations from Lande's approximation as a function of time after the shift. In the Lande case, i.e., when $C \ll 1$, the mean distance from the new optimum is well approximated by $D_L(t)$. Deviations increase with shift size and extent of polygenicity (as seen, e.g., in the red curve), primarily due to a moderate increase in phenotypic variance during the rapid phase. The increase in the 3rd phenotypic moment is minimal, which is why, in this case, there are no substantial long-term deviations. In the non-Lande case, the distance from the optimum decays faster than predicted by Lande's approximation ($D(t) - D_L(t) < 0$) during the rapid phase, because of the increase in phenotypic variance. The increase in the 3rd central moment during the rapid phase leads to a slower decay than predicted by Lande's approximation during the equilibration phase. Even in the non-Lande case, however, we always find the distance from the optimum during equilibration to be smaller than $\delta$. The simulation results were generated using the *all allele* simulation, as described in Section 2.1.

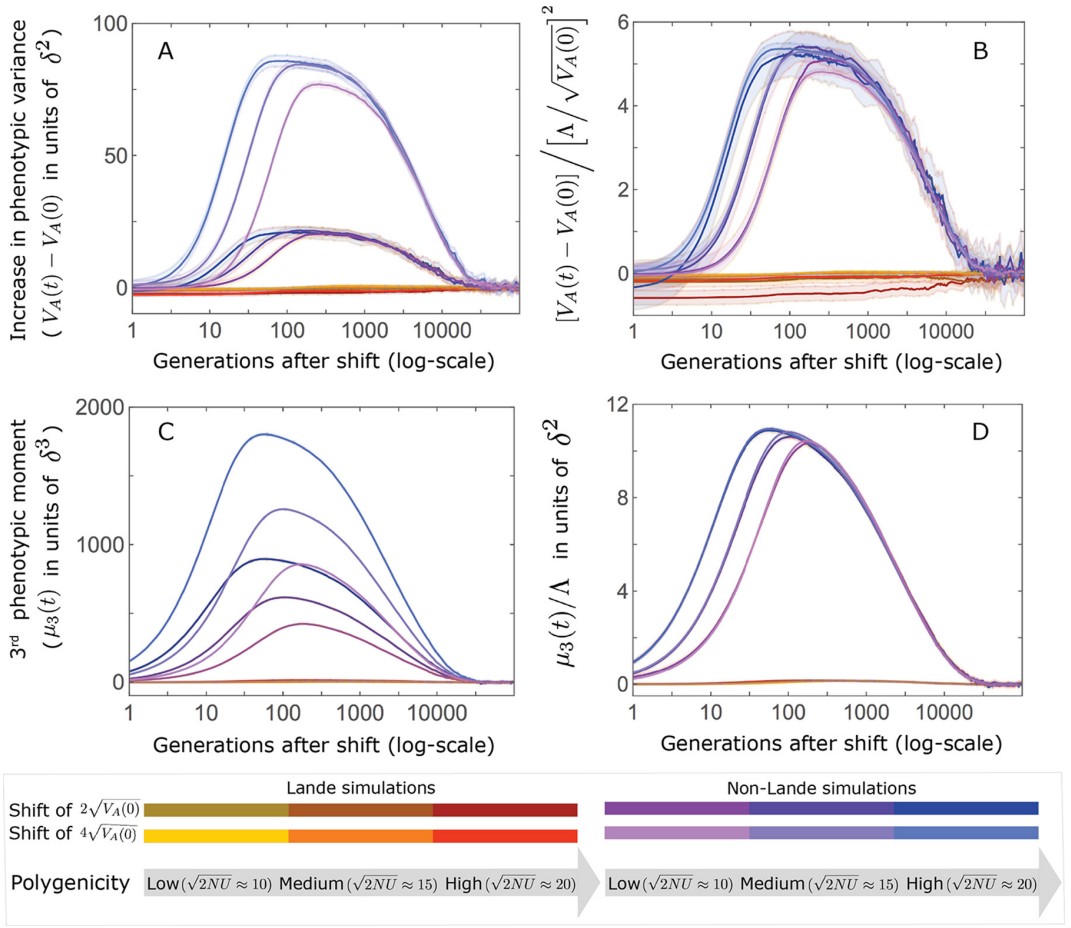

**Appendix 3—figure 27.** The increase in the 2nd and 3rd central moments of the phenotypic distribution. (**A** and **C**) The increase is largely driven by alleles with large effects. Specifically, there are substantial increases only in the non-Lande cases, that is, those with an abundance of new mutations with $a^2 > 4$. (**B** and **D**) We conjecture that the compound parameters that largely determine the increase in moments are: $(\Lambda/\sqrt{V_A(0)})^2$ for the 2nd central moment (**B**) and $\Lambda/\delta$ for the 3rd. Note that the slight differences among curves in the non-Lande case are short-lived (time is measured on a log-scale). The simulation results were generated using the *all allele* simulation, as described in Section 2.1.

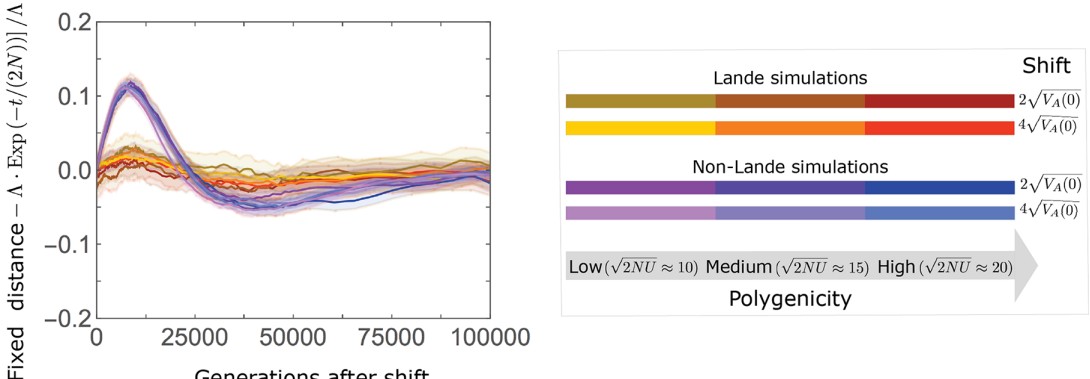

**Appendix 3—figure 28.** The approach of the fixed background to the new optimum is well approximated by $\Lambda e^{-t/(2N)}$, with deviations proportional to the shift size multiplied by a function that depends largely on $C$. We speculate that the (positive) deviations on the intermediate time-scale (e.g., around generation $1000$) in the non-Lande case are primarily due to the slower accumulation of fixations from new mutations, which contribute substantially in this case, whereas the (negative) deviations on the slightly longer time-scale (e.g. around generation $4000$) are due to the cumulative effect of prolonged weak directional selection, which has a substantial effect in this case. The simulation results were generated using the *all allele* simulation, as described in Section 2.1.

