## [Editor Report]

This paper is an impressive look at an important problem: understanding the genetic underpinnings of evolution acting on a quantitative trait. The authors analytically study the response to an abrupt shift in phenotypic optimum, in terms of both phenotype and genetic basis (how various alleles/loci contribute to this response). The basic assumptions are classic, but the methods and findings are new (especially the finite population effects) and well supported by clear analytical approximations and extensive simulation checks. The main finding is that the relative contribution of large vs moderate effect alleles changes substantially and predictably over a long-time period after the shift, even though the phenotypic changes are already undetectable over this period.

---

## [Decision Letter]

**Decision letter after peer review:**

Thank you for submitting your article "Polygenic adaptation after a sudden change in environment" for consideration by *eLife*. Your article has been reviewed by 2 peer reviewers, and the evaluation has been overseen by a Reviewing Editor and Patricia Wittkopp as the Senior Editor. The following individuals involved in review of your submission have agreed to reveal their identity: Peter L Ralph (Reviewer #1); Guillaume Martin (Reviewer #2).

Essential revisions:

The reviewers and I appreciated the impressive steps made in the analysis of this important model. The paper is a difficult read, but that mostly reflects the challenging area and the broad scope of the results. Most of our comments focus on how the clarity could be improved.

1) The reviewers have numerous suggestions on improving the clarity of the model. In particular, I think we all struggled with the parallel tracks throughout the paper.

There is enough new intuition from the Lande-style results that there's an argument for putting more of the technical results about the skewed distribution into the appendix. This would improve the readability of the paper, and I think would likely significantly broaden what the average *eLife* reader could take away from the paper. More technical readers will likely read the appendix anyway if it is clearly signposted from the main text. That said, I'm not going to force this move on you, and many of the suggested improvements to the manuscript could be made without this big change

2) In our post-review discussion one point that came through clearly is that we all struggled to keep track of which of the simulation frameworks was being used at various points in the paper. Please make sure this is clearer throughout the manuscript.

I apologize for the somewhat slow turn around I think everyone (especially myself) is currently struggling to get things done.

Finally, the Coop lab discussed the preprint paper as part of our journal club, everyone liked it and got a lot out of the paper. We needed some more background on why large effect mutations lead to a skewed distribution under selection. We also felt like the results about skew changing the rate of approach to the optimum could be explained more. One of the lab members came up with the following argument:

"When the phenotypic mean is far below the optimum (1-D^2/V_S < 0), the phenotypic distribution sits on a convex part of the Gaussian fitness function. There, the positive skew of the phenotypic distribution, by shifting variance away from the left tail (where it is inefficiently selected) towards the right tail (where it is efficiently selected), increases the average efficiency of selection and therefore the rate at which the mean moves towards the optimum.

The opposite is true when the phenotypic mean is nearer the optimum though still not very close (1 >> 1-D^2/V_S > 0), because then the phenotypic distribution sits on a concave part of the fitness function.

Finally, when the phenotypic mean is just below the optimum, so that the phenotypic distribution straddles the optimum, positive skew shifts variance away from the left tail (where it is relatively efficiently selected in the "correct" direction) towards the right tail (where it is relatively efficiently selected in the "incorrect" direction), and thus retards selection on the mean."

I am not sure if this argument is fully correct/useful, but I wanted to send it along in case it helped. I do think a little more intuition on this point would be useful.

*Reviewer #2 (Recommendations for the authors):*

This paper tackles an old but definitely unsolved issue (at least in my opinion, and this opinion is supported by the review of the literature in the intro).

One main limit in existing treatments of this model of gaussian fitness landscape has been either to tackle non equilibrium, or finite populations. The approach here proposes a methodology to fill this gap on both accounts, so I think it is a new and important contribution to theoretical quantitative genetics.

As a theoretician, I appreciated the appendix, which could be made a bit clearer there and then but is overall flowing and easy to follow, especially when it comes to understanding where we are going. I also appreciated the many illustrative figures in the appendix to check various steps in the approximations by simulations, and as a way to explore a wide parameter range. Overall, the accuracy of the approximations in the various figures is very good. These approximations are not rigorously derived as a limit for some parameter being small etc., some approximations are plugged into others (eg the linear or non linear lande and non lande cases), but we get an intuition of the idea behind the approx., and we get quite many simulations to back it up. Of course a rigorous treatment or a dedicated simulation paper might help identify the exact parameter range of validity, but these would be articles of their own. As such the paper is clearly dense enough!

I suggest that the authors clarify/emphasize two main points for the reader in main text

– First that the phenotypic dynamics are overall correctly approximated by the lande model in the parameter range that was analysed. Indeed, the deviations are much smaller than the main trend in Figure 2 for ex. For the main part of observable phenotypic adaptation (ie until t1). What is lande or non lande here is not this phenotypic mean trajectory but rather its consequences in the longer term dynamics (if I got this right). That should be emphasized and maybe cases where the lande model is not accurate to begin with should be shown too in the simulations.

– Second, therefore, the main contribution of the paper is on (i) obtaining the lande equation without the lande assumptions, (maybe, my knowledge of the vast literature is not large enough to knpw if the various derivations differ that much) and (ii) deriving the longer term dynamics after t1 and especially the genetic basis resuts (relative contribution of alleles as a function of their effect size). It would be nice to pull that string a bit further and illustrate a GWAS or QTL prediction between a pop staying in the original state and a pop having evolved in the new environment, either before t1, or after t1. This is to me the main impact that these results will have and it is somewhat not pushed to its final biological conclusion. Or I missed it and then it must be better put forward.

The paper is very long, both main text and the appendix. I found both flowing though, and that we were guided well into this elaborate modelling exercise. I don't know if this length will be a problematic criterion, that is for the editor to judge.

It was not always easy to know which type of simulations were used for each figure and check. The top check being the true individual based simulations of course. Maybe a table (sup info) dedicated to that would help clarify this without having to go back to methods to know which method is what.

In the intro maybe a link to other models mixing diffusion equation and deterministic pheno models would be nice, I am not sure I saw a reference to the stochastic house of cards approx. and its 'offspring' or to the zhang and hill methods I think.

---

## [Author Response]

Essential revisions:The reviewers and I appreciated the impressive steps made in the analysis of this important model. The paper is a difficult read, but that mostly reflects the challenging area and the broad scope of the results. Most of our comments focus on how the clarity could be improved.

We thank the reviewers for their many helpful comments.

1) The reviewers have numerous suggestions on improving the clarity of the model. In particular, I think we all struggled with the parallel tracks throughout the paper.There is enough new intuition from the Lande-style results that there's an argument for putting more of the technical results about the skewed distribution into the appendix. This would improve the readability of the paper, and I think would likely significantly broaden what the average eLife reader could take away from the paper. More technical readers will likely read the appendix anyway if it is clearly signposted from the main text. That said, I'm not going to force this move on you, and many of the suggested improvements to the manuscript could be made without this big change

We appreciate the importance of simplifying the presentation as much as possible, but have a good reason for keeping the deviations from Lande’s approximation in the main text. While these deviations typically have small effects on the phenotypic response (given our assumptions on model parameters), they have a qualitative impact on the allelic response, which is crucial to understanding the turnover in the genetic basis of adaptation over the longer term (during the equilibration phase). The impact of deviation from Lande’s approximation, notably of the third moment of the phenotypic distribution, on the long-term allelic response is quantified by the composite-parameter 
C
 (defined in Equation 22). C=0 *only in the infinitesimal limit* (see the new Appendix 3, Section 8). In any realistic case, with a finite number of segregating alleles with nonzero phenotypic effects, C>0; what we call Lande cases are just those in which C<<1. As the second reviewer suggested, we modified the text to make these points clearer. Specifically:

1) In the section on the *phenotypic response,* we now state that phenotypic deviations are typically small in the parameter ranges we consider. First, we say that (L 278):

“More generally, given our assumptions that polygenicity is high and that the shift is not too large (see Section 6 of Appendix 3) the deviations from Lande's approximation are usually small and their magnitude is determined by the distribution of allele effect sizes (see section on *the allelic response in the rapid phase* and Section 6 of Appendix 3)”,

and later in the same section, we add that (L 298):

“The same reasoning suggests that very large shifts in optima or low polygenicity could also lead to substantial changes to the 2nd and 3rd central moments of the phenotypic distribution (Section 6 of Appendix 3 and Barton and Turelli, 1987), but these cases violate our assumptions and are beyond the scope of this manuscript.”

2) After we define C (Equation 22 in *the allelic response in the equilibration phase*), we added a paragraph to explain its importance for the allele and phenotypic dynamics, and describe when the deviations from Lande’s approximation might become substantial (L 485):

“When alleles with larger effects contribute substantially to the variance at equilibrium (and 
C
 is appreciable), Lande's approximation becomes inaccurate. The prevalence of large effect alleles leads to a quasi-static decay of the mean distance from the new optimum, 
D
, during the equilibration phase (Figure 2D and the section on the *phenotypic response*). For the distance during the equilibration phase, and therefore for the deviations from Lande's phenotypic approximation to be substantial, would require that C>>1 (see Section 6 of Appendix 3), which implies that the vast majority (e.g., > 90%) of the genetic variance at equilibrium arises from alleles with large effect sizes, say with a^2^>>4 (see Figure 5A and Equation 22). Even a small distance 
D
 during the equilibration phase, however, would result in prolonged, weak directional selection that could markedly amplify the difference in fixation probabilities between opposite alleles. Our linear Lande approximation does not account for this amplification, and it could therefore greatly underestimate the total long-term allelic contribution.”

3) In Appendix 3 Section 6, we demonstrate that given our assumptions on model parameters, the deviations from Lande’s approximation will be substantial only if C>>1.

2) In our post-review discussion one point that came through clearly is that we all struggled to keep track of which of the simulation frameworks was being used at various points in the paper. Please make sure this is clearer throughout the manuscript.

We made several changes to clarify our use of simulations.

In the section on *simulations and resources* in the main text, we added the following to clarify when and why we use each type of simulation (L. 220): “In the main text, we use the simulations that afford the highest resolution in comparisons with analytical predictions, whereas in Section 2.2 of Appendix 3 we validate our main results against simulations that realize the full model (at a lower resolution). Specifically, we compare most of the predictions about the allele dynamics with the results of *single allele* simulations, running 250,000 replicas for any given allele effect size and optimum shift size (see parameter choices below). The *single allele* simulations do not describe phenotypic change or the trajectories of mutations that arise after the shift in optimum, however. We therefore compare the predictions for these processes with the results of the *all alleles* simulations; in these simulations, we run 2,500 replicas with any given set of parameters.”

We have also added a bullet-point description of the simulation parameters used in the main text (L. 229): “The simulations used in the main text correspond to the two qualitative phenotypic responses described below, which we refer to as the Lande and non-Lande (see *results*). Specifically, we use the following simulation parameter values:

In all simulations, we take a populations size of 
N=104
 and a shift size of 
Λ=2VA(0)
 or 
4VA(0)
. Since we work in units of 
δ
, the typical deviation of the population mean from the optimum at equilibrium, we take 
VS=2N
.The *single allele* simulations always assume the Lande phenotypic response, which is determined by the initial genetic variance; we take an initial variance such that 
VA(0)=17⋅δ
.The *all alleles* simulations are specified by the mutation rate, 
U
, and the distribution of allele effect sizes squared, for which we use an exponential distribution (measuring the trait in units of 
δ
). We use the following parameter values:Lande case: 
U=0.03
 and 
E(a2)=E(S)=1
.Non-Lande case: 
U=0.01
 and 
E(a2)=E(S)=16
.These parameter choices yield the same genetic variance at equilibrium (before the shift) in both cases; specifically, 
VA(0)=29∙δ
.”

In figure captions, we state which of the simulation types we used and refer the reader to the description in the *simulations and resources* section.

In Appendix 3, we moved up the section on simulations such that it appears early on (as Section 2) and before the figures in which simulations are used. We also added a bullet-point description (similar to the one in the main text) of the simulation parameters used in the Appendix (L. 942-964).

I apologize for the somewhat slow turn around I think everyone (especially myself) is currently struggling to get things done.Finally, the Coop lab discussed the preprint paper as part of our journal club, everyone liked it and got a lot out of the paper. We needed some more background on why large effect mutations lead to a skewed distribution under selection. We also felt like the results about skew changing the rate of approach to the optimum could be explained more. One of the lab members came up with the following argument:"When the phenotypic mean is far below the optimum (1-D^2/V_S < 0), the phenotypic distribution sits on a convex part of the Gaussian fitness function. There, the positive skew of the phenotypic distribution, by shifting variance away from the left tail (where it is inefficiently selected) towards the right tail (where it is efficiently selected), increases the average efficiency of selection and therefore the rate at which the mean moves towards the optimum.The opposite is true when the phenotypic mean is nearer the optimum though still not very close (1 >> 1-D^2/V_S > 0), because then the phenotypic distribution sits on a concave part of the fitness function.Finally, when the phenotypic mean is just below the optimum, so that the phenotypic distribution straddles the optimum, positive skew shifts variance away from the left tail (where it is relatively efficiently selected in the "correct" direction) towards the right tail (where it is relatively efficiently selected in the "incorrect" direction), and thus retards selection on the mean."I am not sure if this argument is fully correct/useful, but I wanted to send it along in case it helped. I do think a little more intuition on this point would be useful.

We agree with this explanation and have a similar one in Appendix 3 Section 1.2 (L 885-896), where we derive the phenotypic equation. We do not discuss the case in which 
1−D2/VS<0
 for a couple of reasons. First, given our assumption that 
Λ⪅VS
, we would expect this to hold, at most, for a brief period after the shift, and for stabilizing selection to be negligibly weak compared to directional selection during that time. Second, we rely on the assumption that 
Λ⪅VS
 in deriving our phenotypic equation, and do not expect the equation to hold if 
1−D2/VS
 is negative and substantial. We choose do not go into these details in the main text, as they are not important to the main points of the paper. We do, however, refer the reader to Appendix 3 Section 1.2 for more details on the interpretation of this and other terms in the allelic equation (L. 257).

We did, however, add some intuition about the main terms of the phenotypic equation after it first appears (L. 251): “The 1^st^ term on the right-hand side reflects selection to reduce the distance between the mean phenotype and the new optimum, which is proportional to this distance and to the additive genetic variance (Lande, 1976). The 2^nd^ term reflects the effect of stabilizing selection on an asymmetric (skewed) phenotypic distribution. In particular, when the mean phenotype is near the optimum (and this 2^nd^ term is approximately 
μ3(t)/(2VS)
), stabilizing selection pushes the mean phenotype in the direction opposite to the thicker tail of the phenotype distribution (see Section 1.2 of Appendix 3 for further discussion of Equation 3).”

Reviewer #2 (Recommendations for the authors):This paper tackles an old but definitely unsolved issue (at least in my opinion, and this opinion is supported by the review of the literature in the intro).One main limit in existing treatments of this model of gaussian fitness landscape has been either to tackle non equilibrium, or finite populations. The approach here proposes a methodology to fill this gap on both accounts, so I think it is a new and important contribution to theoretical quantitative genetics.As a theoretician, I appreciated the appendix, which could be made a bit clearer there and then but is overall flowing and easy to follow, especially when it comes to understanding where we are going. I also appreciated the many illustrative figures in the appendix to check various steps in the approximations by simulations, and as a way to explore a wide parameter range. Overall, the accuracy of the approximations in the various figures is very good. These approximations are not rigorously derived as a limit for some parameter being small etc., some approximations are plugged into others (eg the linear or non linear lande and non lande cases), but we get an intuition of the idea behind the approx., and we get quite many simulations to back it up. Of course, a rigorous treatment or a dedicated simulation paper might help identify the exact parameter range of validity, but these would be articles of their own. As such the paper is clearly dense enough!

Thank you kindly. We greatly appreciate the careful reading of the paper and helpful comments on the pdf, especially on the many technical results in Appendix 3!

I suggest that the authors clarify/emphasize two main points for the reader in main text– First that the phenotypic dynamics are overall correctly approximated by the lande model in the parameter range that was analysed. Indeed, the deviations are much smaller than the main trend in Figure 2 for ex. For the main part of observable phenotypic adaptation (ie until t1). What is lande or non lande here is not this phenotypic mean trajectory but rather its consequences in the longer term dynamics (if I got this right). That should be emphasized and maybe cases where the lande model is not accurate to begin with should be shown too in the simulations.

Great points! We heeded the reviewer’s advice.

– Second, therefore, the main contribution of the paper is on (i) obtaining the lande equation without the lande assumptions, (maybe, my knowledge of the vast literature is not large enough to knpw if the various derivations differ that much) and (ii) deriving the longer term dynamics after t1 and especially the genetic basis resuts (relative contribution of alleles as a function of their effect size). It would be nice to pull that string a bit further and illustrate a GWAS or QTL prediction between a pop staying in the original state and a pop having evolved in the new environment, either before t1, or after t1. This is to me the main impact that these results will have and it is somewhat not pushed to its final biological conclusion. Or I missed it and then it must be better put forward.

Scope of Lande’s approximation. As far as we know, this is the first time that Lande’s equation has been derived without making Lande’s assumptions (see new *Appendix 3* Section 8). When the number of loci and their effects are finite there is always some deviations from Lande’s approximation; however, as we discuss and illustrate in the section on the *phenotypic response* (L 272-284), Lande’s approximation works well for a wide range of parameters.

Relatedly, in the annotated pdf you asked us for a citation for our allelic and phenotypic equations (Equations 1 and 2 in Appendix 3); to the best of our knowledge, these equations are new. Specifically, in the case of a Gaussian fitness function, which we have considered, Barton and Turelli (1987) derive approximations based on the assumption that the distance of the mean from the optimum is far smaller than the width of the fitness function, i.e., that 
D2≪VS
. We relax this assumption, instead assuming that 
D≲VS
, which results in the additional factor of 
1−D2/VS
 in the stabilizing selection term (Equation 7). When 
D2≪VS
 our equations reduce to the equations derived by Barton and Turelli (1987) in the rare alleles approximation, and Bürger (1991) assuming a parabolic fitness function. We detail the relationship to previous work in Appendix 3 Section 1, L. 815-828.

Predictions for genetic differentiation between populations. We actually do provide predictions for population differentiation with and without changes in environment, although they are phrased in slightly different terms. Notably, Figure 7 shows these predictions over the long-term (also see L. 587-598 in the same section). In brief, we find that the shift would typically result in a slight excess of fixations of alleles that are aligned with the shift relative to opposing ones (Figure 7B), and a slight increase in the total number of fixations compared to the case without the change in environment (Figure 7C). In the *discussion*, we also consider the implications of this finding to the question of identifying adaptive changes (L. 724-765). The short-term behavior is summarized in Figure 4 and Appendix 3-figures 8-11, relating the changes in frequency with effect size and initial frequency. Doubtless, more can be done.

The paper is very long, both main text and the appendix. I found both flowing though, and that we were guided well into this elaborate modelling exercise. I don't know if this length will be a problematic criterion, that is for the editor to judge.

The comments and corrections in the annotated pdf were very helpful. We greatly appreciate the reviewer taking the time to check numerous equations; specifically, the spotting of a missing factor of 2 in Equation 16!

2 extra suggestions not in the annotated pdf:It was not always easy to know which type of simulations were used for each figure and check. The top check being the true individual based simulations of course. Maybe a table (sup info) dedicated to that would help clarify this without having to go back to methods to know which method is what.

We made several changes to clarify our use of simulations: please see p. 3 of this reply for details.

In the intro maybe a link to other models mixing diffusion equation and deterministic pheno models would be nice, I am not sure I saw a reference to the stochastic house of cards approx. and its 'offspring' or to the zhang and hill methods I think.

Having looked up the papers we thought the reviewer is alluding to, we remain unclear about what they mean by “other models mixing diffusion equation and deterministic pheno models” or more generally, how their methods and questions relate to our analysis. Specifically, we looked at Burger, Wagner and Stettinger (1989) and at Zhang and Hill (2002) Pleiotropic Model of Maintenance of Quantitative Genetic Variation at Mutation-selection Balance and (2003) Multivariate Stabilizing Selection and Pleiotropy in the Maintenance of Quantitative Variation; it is of course possible that the reviewer had other papers in mind.

Our reading of the house of cards approximations, for example, suggests that they are primarily concerned with the maintenance of quantitative genetic variation at equilibrium under explicit or apparent stabilizing selection. They typically rely on three assumptions: (1) A continuum of possible alleles at a *fixed* number of loci (in contrast, we use a bi-allelic, infinite sites model). (2) That selection at each locus dominates over mutation (we implicitly make the same assumption by using an infinite-sites model). (3) That all alleles have 
2Ns≫1
. We do not make this last assumption, which is important because weakly and moderately selected alleles are plausibly common, and they play a crucial role in the allelic adaptive response (and in genetic variance, actually).

Our reading of Burger, Wagner and Stettinger (1989) and Zhang and Hill (2003) suggests that stochastic house of cards approximations of phenotypic variance are typically derived using a diffusion approximation, under assumptions 1-3. Importantly, we didn’t see how the stochastic house of cards type papers mixed deterministic and stochastic methods. It is nonetheless possible that we missed something, as this literature is vast and complex!